# Treatment of peanut allergy and colitis in mice via the intestinal release of butyrate from polymeric micelles

Ruyi Wang[1,2,14], Shijie Cao[1,14], Mohamed Elfatih H. Bashir[1,14], Lauren A. Hesser[1], Yanlin Su[3,4], Sung Min Choi Hong[3,4], Andrew Thompson[3,4], Elliot Culleen[3,4], Matthew Sabados[3], Nicholas P. Dylla[5], Evelyn Campbell[3,6], Riyue Bao[7,8,9], Eric B. Nonnecke[10], Charles L. Bevins[10], D. Scott Wilson[1,11], Jeffrey A. Hubbell[1,12,13] ✉ & Cathryn R. Nagler[1,3,4,12] ✉

The microbiome modulates host immunity and aids the maintenance of tolerance in the gut, where microbial and food-derived antigens are abundant. Yet modern dietary factors and the excessive use of antibiotics have contributed to the rising incidence of food allergies, inflammatory bowel disease and other non-communicable chronic diseases associated with the depletion of beneficial taxa, including butyrate-producing Clostridia. Here we show that intragastrically delivered neutral and negatively charged polymeric micelles releasing butyrate in different regions of the intestinal tract restore barrier-protective responses in mouse models of colitis and of peanut allergy. Treatment with the butyrate-releasing micelles increased the abundance of butyrate-producing taxa in *Clostridium* cluster XIVa, protected mice from an anaphylactic reaction to a peanut challenge and reduced disease severity in a T-cell-transfer model of colitis. By restoring microbial and mucosal homoeostasis, butyrate-releasing micelles may function as an antigen-agnostic approach for the treatment of allergic and inflammatory diseases.

The gut microbiome has many effects on both mucosal and systemic health[1–3]. While many mechanisms by which bacteria regulate mucosal homoeostasis remain unknown, short-chain fatty acids (SCFAs), especially butyrate, have been well studied as immunoregulatory molecules[4,5]. Butyrate is produced by a subset of intestinal bacteria through the fermentation of dietary fibre[6]. Decreased abundance of butyrate-producing bacteria has been observed in human cohorts of food allergy, asthma, inflammatory bowel disease (IBD) and other non-communicable chronic diseases (NCCDs)[7–13]. However, oral delivery of butyrate to the small intestine (where food antigens are

[1]Pritzker School of Molecular Engineering, University of Chicago, Chicago, IL, USA. [2]Department of Chemistry, University of Chicago, Chicago, IL, USA. [3]Biological Sciences Division, University of Chicago, Chicago, IL, USA. [4]Department of Pathology, University of Chicago, Chicago, IL, USA. [5]Duchossois Family Institute, University of Chicago, Chicago, IL, USA. [6]Committee on Microbiology, University of Chicago, Chicago, IL, USA. [7]Department of Pediatrics, University of Chicago, Chicago, IL, USA. [8]UPMC Hillman Cancer Center, Pittsburgh, PA, USA. [9]Department of Medicine, University of Pittsburgh, Pittsburgh, PA, USA. [10]Department of Microbiology and Immunology, School of Medicine, University of California, Davis, CA, USA. [11]Department of Biomedical Engineering, Johns Hopkins School of Medicine, Baltimore, MD, USA. [12]Committee on Immunology, University of Chicago, Chicago, IL, USA. [13]Committee on Cancer Biology, University of Chicago, Chicago, IL, USA. [14]These authors contributed equally: Ruyi Wang, Shijie Cao, Mohamed Elfatih H. Bashir. ✉e-mail: jhubbell@uchicago.edu; cnagler1@uchicago.edu

absorbed) and large intestine (where most commensal bacteria reside) has been a challenge.

Butyrate, even with enteric coating or encapsulation, possesses a foul and lasting odour and taste. As a sodium salt, orally administered butyrate is not absorbed in the part of the gut where it can have a therapeutic effect and is metabolized too rapidly to maintain a pharmacologic effect[14]. Previous work in murine models that demonstrated therapeutic effects of butyrate relied on high concentration, ad libitum exposure to butyrate (mM quantities in drinking water for several weeks) or utilized butyrylated starches[15–21]. In human trials, intrarectal delivery of butyrate is moderately efficacious in treating colitis but is not a preferred route of administration[22]. A more controlled and practical delivery strategy is needed to exploit the potential therapeutic benefits of butyrate clinically to treat allergic and inflammatory diseases of the lower gastrointestinal (GI) tract. To address this problem, we designed block copolymers that can form water-suspensible micelles carrying a high content of butyrate in their core. These polymer formulations mask the smell and taste of butyrate and act as carriers to release the active ingredient (butyrate) over time as the micelles transit the GI tract. They can be formulated and administered as a suspension, allowing high dose without requiring a large number of pills. We developed two novel polymers: one with a neutral charge which predominantly releases butyrate in the ileum (NtL-ButM), and one with a negative charge which predominantly releases butyrate in the caecum (Neg-ButM). This micelle system allows us to deliver butyrate to its site of biological effect and overcome the existing limitations of oral administration. Butyrate is produced by certain members of the Clostridia class in the distal GI tract and is the preferred energy substrate for colonic epithelial cells[5]. It strengthens gut barrier function by stabilizing hypoxia-inducible factor and maintains epithelial tight junctions[23]. To mediate its immunomodulatory functions, butyrate acts via signalling through specific G protein coupled receptors or as an inhibitor of histone deacetylase activity (HDACs)[24]. HDAC inhibition by SCFAs promotes the differentiation of colonic regulatory T ($T_{reg}$) cells[17–19]. The ability of butyrate to regulate barrier immunity makes it an ideal drug candidate for inflammatory diseases of the gut such as food allergy and colitis.

We have previously shown a protective role for butyrate-producing Clostridia in the prevention of food allergies in both mouse models and human cohorts[7,8,10,25]. Neonatal administration of antibiotics reduced intestinal microbial diversity and impaired epithelial barrier function, resulting in increased access of food allergens to the systemic circulation[25]. Administration of a consortium of spore-forming bacteria in the Clostridia class restored the integrity of the epithelial barrier and prevented allergic sensitization to food[25]. We went on to demonstrate a causal role for bacteria present in healthy infant microbiota in protection against cow's milk allergy[8]. Transfer of the microbiota from healthy, but not cow's milk allergic (CMA), human infants into germ-free (GF) mice protected against an anaphylactic response to a cow's milk allergen. By integrating differences in the microbiome signatures present in the healthy and CMA microbiotas with the changes each induced in ileal gene expression upon colonization of GF mice, we identified a single butyrate-producing Clostridial species, *Anaerostipes caccae*, that mimicked the effects of the healthy microbiota upon monocolonization of GF mice[8]. Recent findings from a diverse cohort of twin children and adults concordant and discordant for food allergy validated the mouse model data with human microbiome samples. We found that most of the operational taxonomic units (OTUs) differentially abundant between healthy and allergic twins were in the Clostridia class; the broad age range of the twins studied indicated that an early-life depletion of allergy-protective Clostridia is maintained throughout life[10].

As with food allergy, the incidence of IBD, including ulcerative colitis and Crohn's disease, has been rapidly increasing and is dependent on the commensal microbiota[11,26]. Bacterial dysbiosis in patients with IBD has been well characterized and consistently shows decreased abundance of butyrate-producing species[12,13]. Clinical trials have attributed the efficacy of faecal microbiota transplant (FMT) for the treatment of *Clostridium difficile* infection and ulcerative colitis to some engraftment of butyrate-producing taxa including *Lachnospiraceae* and *Ruminococcaceae* (*Clostridium* clusters XIVa and IV)[27,28]. However, long-term engraftment of oxygen-sensitive anaerobic bacteria from FMT or oral delivery of consortia has proven challenging[29,30]. We therefore sought to explore butyrate itself as a candidate drug to maintain both microbial and mucosal homoeostasis in the dysbiotic gut and to treat the consequences of inappropriate responses to the contents of the gastrointestinal lumen, including both dietary antigens (food allergies) and bacterial products (IBD).

The butyrate-conjugated polymer formulations developed for this study release butyrate in distinct segments of the lower GI tract, in contrast to sodium butyrate (NaBut), which is predominantly absorbed in the stomach. When the two polymers are administered together (ButM), they modulate barrier integrity in antibiotic-treated mice and in mice treated with dextran sodium sulfate (DSS), a chemical perturbant that induces epithelial barrier dysfunction. Intragastric administration of our butyrate-prodrug micelles ameliorates an anaphylactic response to peanut challenge in a mouse model of peanut allergy. We hypothesize that this therapeutic effect may be elicited in part by modulating the microbiome, as treatment with ButM increases the abundance of bacteria in Clostridial clusters IV and XIVa, which are known to contain butyrate-producing taxa. Moreover, treatment with ButM reduces disease severity in a T-cell-transfer model of colitis. These findings (Fig. 1) pave the way for future clinical translation of butyrate micelles in treating food allergies and inflammatory bowel disease.

## Results

### Copolymers formulate butyrate into water-suspensible micelles

The block copolymer amphiphile pHPMA-b-pBMA was synthesized through two steps of reversible addition-fragmentation chain-transfer (RAFT) polymerization (Fig. 2a). The hydrophilic block was formed from *N*-(2-hydroxypropyl) methacrylamide (HPMA), while the hydrophobic block was from *N*-(2-butanoyloxyethyl) methacrylamide (BMA), thus connecting a backbone sidechain to butyrate with an ester bond. This ester bond can be hydrolysed in the presence of digestive esterases and releases butyrate in the GI tract, resulting in a water-soluble polymer as a final product. In addition to pHPMA-b-pBMA, we also synthesized pMAA-b-pBMA, which has an anionic hydrophilic block formed from methacrylic acid (MAA) (Fig. 2a). At the block size ratios used herein, both pHPMA-b-pBMA and pMAA-b-pBMA contain 28% butyrate by weight (Supplementary Figs. 1–9). These block copolymers can be then formulated into nanoscale micelles to achieve high suspensibility in aqueous solutions as well as controlled release of butyrate from the core. The pHPMA-b-pBMA was self-assembled into neutral micelles (NtL-ButM) through a cosolvent evaporation method (Fig. 2a). The hydrophobic pBMA block forms the core, while the hydrophilic pHPMA forms the corona. In contrast, pMAA-b-pBMA cannot be formulated into micelles by this method because of the formation of intramolecular hydrogen bonds between pMAA chains[31]. Such bonding can, however, be disrupted when a strong base, here NaOH, is titrated into the mixture of pMAA-b-pBMA polymer to change methacrylic acid into ionized methacrylate[32–34]. Upon base titration, pMAA-b-pBMA polymer can then self-assemble into negatively charged micelles (Neg-ButM) (Fig. 2a). Cryogenic electron microscopy (CryoEM) revealed the detailed structure of the micelles, especially the core structure made of pBMA, which was more condensed with higher contrast. CryoEM images indicated that the diameter of the core of NtL-ButM was 30 nm, while Neg-ButM had a smaller core diameter of 15 nm (Fig. 2b,c). Both NtL-ButM and Neg-ButM have similar sizes of 44.7 ± 0.8 nm and 39.9 ± 1.6 nm, respectively, measured by dynamic light scattering (DLS) (Fig. 2d and Supplementary Fig. 10). Their low polydispersity index below 0.1

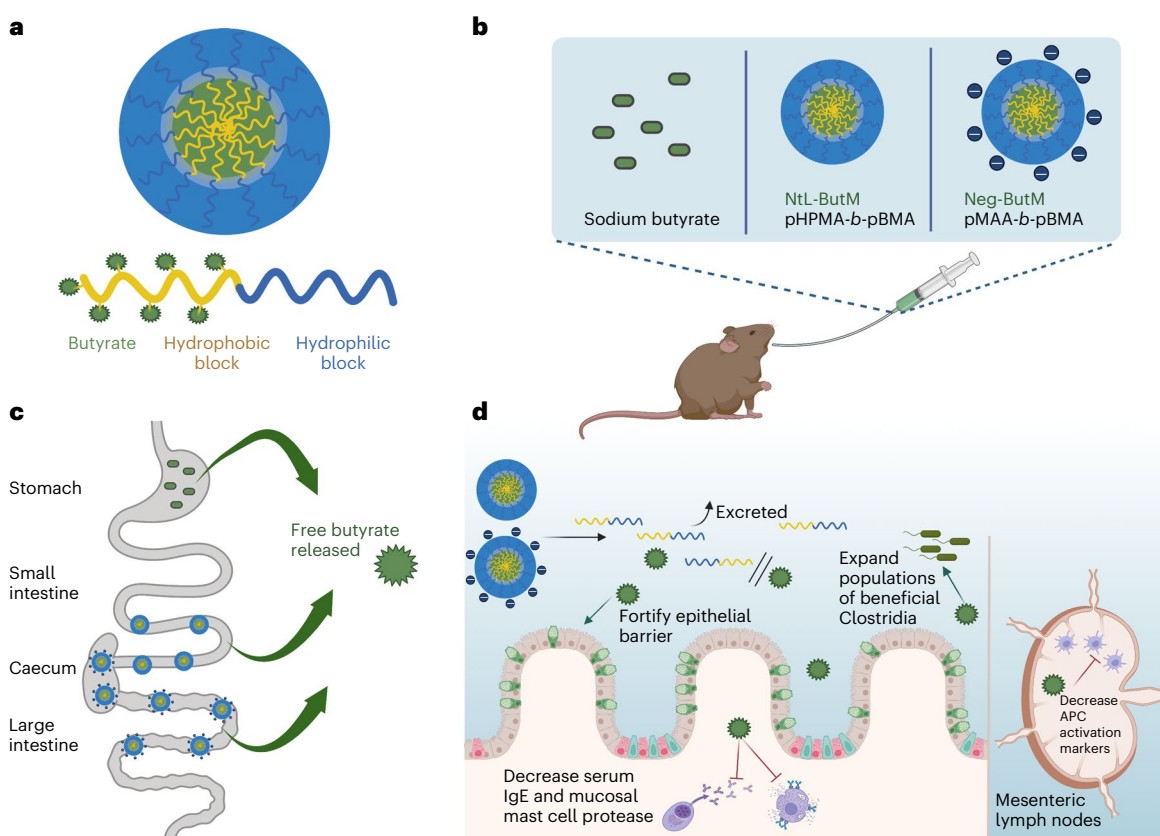

**Fig. 1 | Butyrate-conjugated polymeric micelles transit to the distal GI tract to deliver butyrate and mediate therapeutic effects. a**, Polymeric micelle systems were designed with butyrate conjugated to a polymer with a hydrophobic block, which makes up the inside of the micelle, and a hydrophilic outer block. **b**, Mice were intragastrically gavaged with neutral (NtL-ButM) or negatively charged (Neg-ButM) micelles containing butyrate. Sodium butyrate (NaBut) was used as a control. **c**, After intragastric administration, NaBut is absorbed in the stomach, but polymer micelles transit to and release butyrate in the distal small intestine (NtL-ButM) and caecum/colon (Neg-ButM). **d**, In the distal GI tract, butyrate is cleaved from the micelles' polymer backbone. The backbone is excreted while local butyrate has various therapeutic effects in the intestine and mesenteric lymph nodes in murine models of food allergy and colitis. Graphic created with BioRender.com.

indicated the monodispersity of these micelles. As expected, NtL-ButM has a near-zero ζ-potential of −0.34 ± 0.5 mV, while the ζ-potential of Neg-ButM is −31.5 ± 2.3 mV due to the ionization of methacrylic acid (Fig. 2d). To obtain the critical micelle concentration (CMC) of NtL-ButM and Neg-ButM, which indicates the likelihood of formation and dissociation of micelles in aqueous solutions, pyrene was added during the formulation and the fluorescence intensity ratio between the first and third vibronic bands of pyrene was plotted to calculate the CMC (Supplementary Fig. 11)[35]. Results showed that Neg-ButM had a higher CMC than NtL-ButM (14.0 ± 3.5 μM versus 0.8 ± 0.4 μM) (Fig. 2d). The higher CMC indicated that Neg-ButM micelles would be easier to dissociate in solution, possibly because the surface charge made the micellar structure less stable compared with the neutral micelle NtL-ButM. In addition, we conducted small-angle X-ray scattering (SAXS) analysis on both micelles to obtain the aggregation number (Supplementary Fig. 12). As indicated from Guinier plots, radii of gyration for NtL-ButM and Neg-ButM were 14.2 nm and 13.5 nm (Fig. 2d), respectively, and the structures of micelles were confirmed to be spheres from Kratky plots of SAXS data (Supplementary Fig. 12). We then fitted the SAXS data with a polydispersity core–shell sphere model, with the assumptions that the micelle has a spherical core with a higher scattering length density (SLD) and a shell with a lower SLD[36]. The model gave us the volume fraction of the micelles, the radius of the core and the thickness of the shell, allowing us to calculate the aggregation number and mean distance between micelles. According to the fitting results, aggregation numbers for NtL-ButM and Neg-ButM were 119 and 92, respectively (Fig. 2d).

## Butyrate micelles release butyrate in the lower GI tract

Given that butyrate is linked to the micelle-forming chain via ester bonds, we validated the release of butyrate in ex vivo conditions, including in simulated gastric and intestinal fluids that mimic those biological environments. In the simulated gastric fluid, both Neg-ButM and NtL-ButM showed negligible release of butyrate within hours and sustained slow release over 3 weeks, while Neg-ButM had even slower release rate than NtL-ButM (Fig. 3a). The anionic surface of Neg-ButM in the acidic environment is probably responsible for the resistance to hydrolysis of the BMA core. In addition, we observed that NtL-ButM was stable in simulated gastric fluid in vitro (Supplementary Fig. 13a) but Neg-ButM was not and could aggregate into larger polymer particles (Supplementary Fig. 13b). By contrast, in simulated intestinal fluid, both micelles released most of their butyrate within minutes in the presence of a high concentration of the esterase pancreatin (Fig. 3b).

We then measured butyrate levels in the mouse GI tract after administering a single dose of NtL-ButM or Neg-ButM by intragastric gavage (i.g.). Both liquid chromatography–ultraviolet (LC–UV) and liquid chromatography with tandem mass spectrometry (LC–MS/MS) methods have been used to measure butyrate concentrations in the luminal contents of the ileum, caecum and colon – the sites where butyrate-producing bacteria normally reside[37,38]. However, because the baseline concentration in the ileum was too low for the UV detector, we used LC–MS/MS to measure the butyrate concentration in that GI tract segment (Supplementary Fig. 14). In specific pathogen-free (SPF) (non-antibiotic-treated) mice, intragastric administration of NtL-ButM increased the butyrate concentration

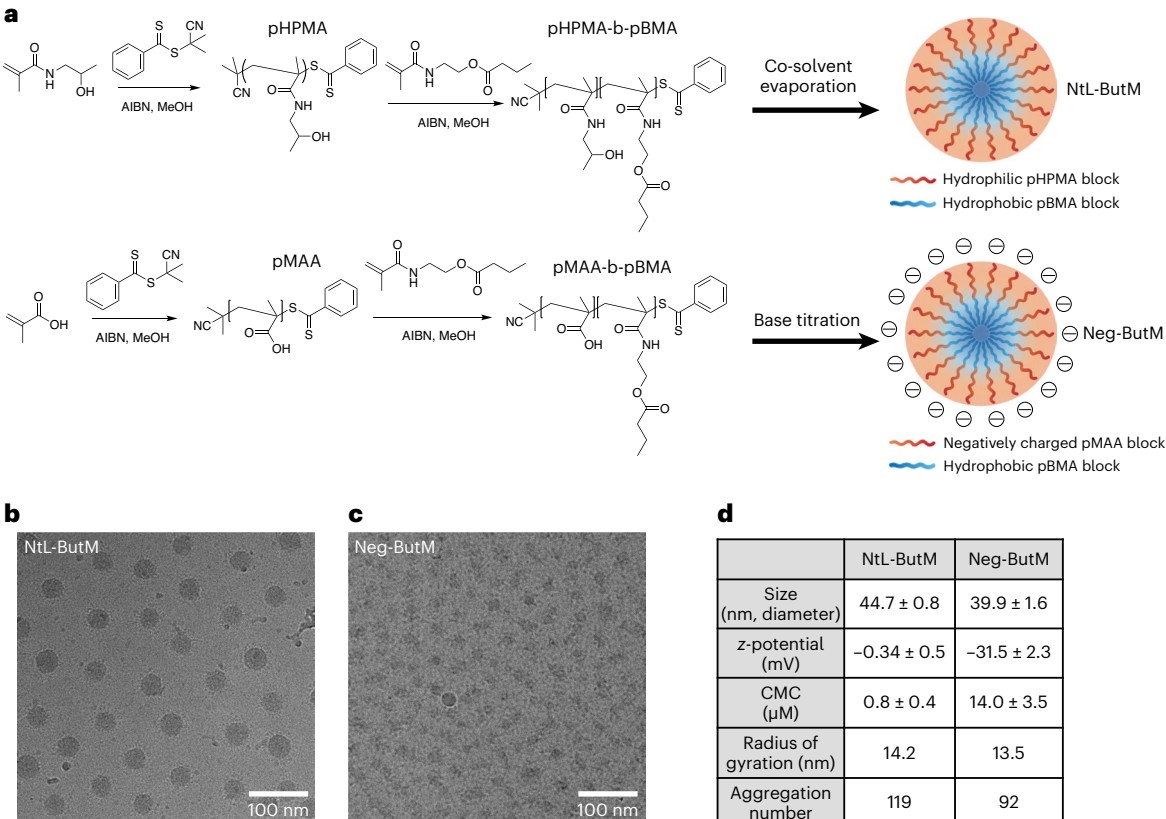

**Fig. 2 | Chemical composition and structural characterization of butyrate-prodrug micelles NtL-ButM and Neg-ButM.** NtL-ButM consists of the neutral block copolymer pHPMA-b-pBMA and Neg-ButM consists of the anionic block copolymer pMAA-b-pBMA. **a**, Synthetic route of pHPMA-b-pBMA and pMAA-b-pBMA, and fabrication of polymeric micelles. Top: NtL-ButM contains a hydrophilic (HPMA) block as the micelle corona, while a hydrophobic (BMA) block forms the micelle core. Bottom: Neg-ButM contains a hydrophilic (MAA) block that forms a negatively charged micelle corona, and the same hydrophobic (BMA) block as NtL-ButM. **b**,**c**, CryoEM images show the spherical structures of micelles NtL-ButM (**b**) or Neg-ButM (**c**). **d**, Table summarizing the characterization of micelles NtL-ButM and Neg-ButM, including hydrodynamic diameter and z-potential from DLS, critical micelle concentration, radius of gyration and aggregation number from SAXS.

in the ileum for up to 2 h after gavage (Fig. 3c), but this was short-lived, and the butyrate concentration did not increase in either the caecum or colon (Fig. 3d,e). Interestingly, Neg-ButM raised butyrate concentrations by 3-fold in the caecum starting from 4 h after gavage and lasting for at least another 8 h but not in the ileum or colon (Fig. 3c–e). It is possible that the butyrate released in the caecum will continuously flow into the colon; our inability to detect increased concentrations of butyrate in the colon is probably due to its rapid absorption and metabolism by the colonic epithelium. In addition, as expected, the polymer backbone of the micelles remained intact when passing through the GI tract. We observed less than 28% molecular weight loss—the percentage of butyrate content—of the polymer in faecal samples collected from 4–8 h after oral administration (Supplementary Fig. 15a,b). Moreover, when incubated in a hydrolytic environment in vitro, the polymer backbone remained intact after releasing most of the butyrate over 7 d in 125 mM sodium hydroxide solution (Supplementary Fig. 15c,d).

We further investigated how these two butyrate micelles transit through the GI tract by administering fluorescently labelled NtL-ButM or Neg-ButM to mice i.g. and visualizing their biodistribution via an in vivo imaging system (IVIS) (Supplementary Fig. 16). In this case, the fluorescent marker was conjugated to the polymer chain, allowing visualization of the transit of the polymer backbone itself. The IVIS results validated that the polymeric micelles were retained in the mouse GI tract for more than 6 h after gavage. The neutral micelle NtL-ButM passed through the stomach and small intestine within 2 h and accumulated in the caecum. However, negatively charged Neg-ButM

accumulated in the stomach first and then gradually travelled through the small intestine to the caecum. Overall, Neg-ButM had a longer retention time in the stomach and small intestine, which is possibly due to a stronger adhesive effect to the gut mucosa imparted by the negative charge[39,40]. Both micelles were cleared from the GI tract within 24 h after administration. In addition, we measured the fluorescence signal from other major organs and plasma by IVIS (Supplementary Fig. 16b), as well as the butyrate concentration in the plasma by LC–MS/MS. The signals were all below the detection limit from both methods, suggesting that there was negligible absorption of these butyrate micelles into the blood circulation from the intestine, consistent with our desire to deliver butyrate to the lower GI tract and to avoid any complexities of systemic absorption of the polymer or micelles.

Due to the existing high level of butyrate produced within the gut in SPF conditions (that is, with an intact microbiome), we conducted similar biodistribution experiments on vancomycin-treated mice (Extended Data Fig. 1a), where most of the Gram-positive bacteria, including butyrate-producing bacteria, were depleted to induce dysbiosis. We observed that both NtL-ButM and Neg-ButM transited to the lower GI tract within an hour (Extended Data Fig. 1b). Free butyrate from NaBut was measurable in the stomach from 1–4 h post gavage but was barely detectable in the lower GI sections such as the caecum and colon (Extended Data Fig. 1c). In contrast, NtL-ButM and Neg-ButM only released 2.3% and 8.8%, respectively, of the amount of butyrate in the stomach as compared to NaBut treatment. In antibiotic-treated mice, both NtL-ButM and Neg-ButM released butyrate in the caecum and

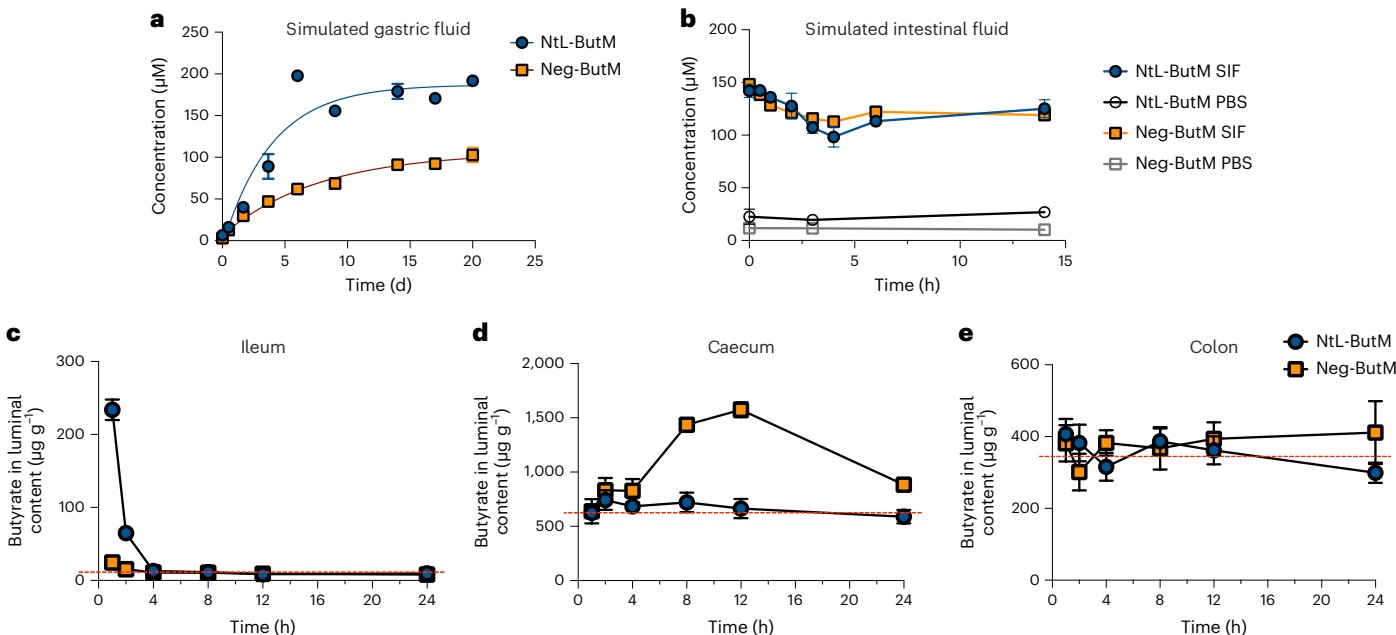

**Fig. 3 | In vitro and in vivo butyrate release from NtL-ButM and Neg-ButM in the GI tract. a,** Both NtL-ButM and Neg-ButM released butyrate slowly in the simulated gastric fluid over 20 d. **b,** Both NtL-ButM and Neg-ButM released their complete butyrate load within minutes in simulated intestinal fluid (SIF) containing high levels of the esterase pancreatin. Neither polymer released butyrate in PBS on these timescales. $n = 3$. **c–e,** The amount of butyrate released in the ileum (**c**), caecum (**d**) or colon (**e**) contents after a single intragastric administration of NtL-ButM or Neg-ButM at 800 mg kg$^{-1}$ to SPF C3H/HeJ mice. Butyrate was derivatized with 3-nitrophenylhydrazine and quantified by LC–MS/MS (ileum samples) or LC–UV (caecum and colon samples). The dotted red lines represent butyrate content in untreated mice. $n = 9$–10 mice per group. Data represent mean ± s.e.m.

colon. The peak concentrations were observed between 2–4 h after oral gavage, which is similar to what we observed from fluorescent signals from the IVIS images (Extended Data Fig. 1b).

Delivery of butyrate to the lower GI tract could affect the host immune response by interacting with the intestinal epithelium. To investigate whether and how our butyrate micelles regulate gene expression in the distal small intestine, we performed RNA sequencing of the ileal epithelial cell compartment (Extended Data Fig. 2a). Germ-free (and thus butyrate-depleted) C3H/HeN mice were treated daily with NtL-ButM i.g. for 1 week and ileal epithelial cells were collected for RNA isolation and sequencing. Because only NtL-ButM (and not Neg-ButM) released butyrate in the ileum, only NtL-ButM was used for this experiment to examine local effects. NtL-ButM-treated mice had unique gene expression signatures compared with those treated with PBS or control polymer, which consists of the same polymeric structure but does not contain butyrate. Interestingly, most genes upregulated by NtL-ButM treatment were Paneth cell-derived antimicrobial peptides (AMPs), including angiogenin 4 (*Ang4*), lysozyme-1 (*Lyz1*), intelectin (*Itln1*) and several defensins (*Defa3, Defa22, Defa24* and so on) (Extended Data Figs. 2a and 3). We quantified the protein level of intelectin—one of the upregulated AMPs (Extended Data Fig. 2b,c). We chose intelectin because it is known to be expressed by Paneth cells which reside in small intestinal crypts and can recognize the carbohydrate chains of the bacterial cell wall[41]. Paneth cell AMPs have largely been characterized in C57BL/6 mice and specific reagents are available for their detection in that strain[42]. GF C57BL/6 mice were gavaged daily with NtL-ButM or PBS for 1 week. Immunofluorescence microscopy of ileal sections revealed that the NtL-ButM-treated group expressed a large amount of intelectin in the crypts of the ileal tissue. However, images from the PBS group showed limited intelectin signal (Extended Data Fig. 2b). Quantification of relative fluorescence intensity per ileal crypt using ImageJ also showed that the NtL-ButM group had significantly higher expression of intelectin compared with the PBS control (Extended Data Fig. 2c). The intelectin staining thus further

supported the pharmacological effects of NtL-ButM; upregulation of intelectin induced by NtL-ButM was not only demonstrated on the transcriptional level by RNAseq but was also validated at the protein level. We next performed quantitative PCR with reverse transcription (RT–qPCR) on ileal epithelial cells from adult SPF C57BL/6 mice (that is, mice with a replete butyrate-producing microbiota) treated with PBS or NtL-ButM i.g. for 1 week (Supplementary Fig. 17). In contrast to what we observed in the GF mice, there were no significant differences in the expression of AMPs in SPF mice. It has been shown that these AMPs are constitutively expressed, and very few stimuli abrogate or increase their expression in SPF mice[42,43]. As there is no butyrate deficit in these SPF mice, administration of NtL-ButM did not show the effect of upregulation of AMP gene expression that we observed in GF mice. These results suggest that it is unlikely that ButM acts by modulation of AMPs in mice with an intact microbiota.

**Butyrate micelles repair intestinal barrier function**

As discussed above, butyrate-producing bacteria play an important role in the maintenance of the intestinal barrier. To assess the effects of locally delivered butyrate on intestinal barrier integrity, we treated mice with the chemical perturbant DSS for 7 d to induce epithelial barrier dysfunction[44]. Due to the different biodistribution and butyrate release behaviours in vivo from the two butyrate micelles, we reasoned that the combined dosing of NtL-ButM and Neg-ButM would cover the longest section of the lower GI tract and last for a longer time; thus, a 1:1 combination of NtL-ButM and Neg-ButM (abbreviated as ButM) was selected for study. Throughout DSS treatment, and for 3 d after DSS administration was terminated, mice were orally gavaged twice daily with either PBS or ButM at three different concentrations, or once daily with cyclosporin A (CsA) as the positive therapeutic control (as outlined in Fig. 4a). Intragastric gavage of 4 kDa FITC-dextran was used to evaluate intestinal barrier permeability. We detected a significantly higher concentration of FITC-dextran in the serum 4 h after gavage in DSS-treated mice that received only PBS, demonstrating

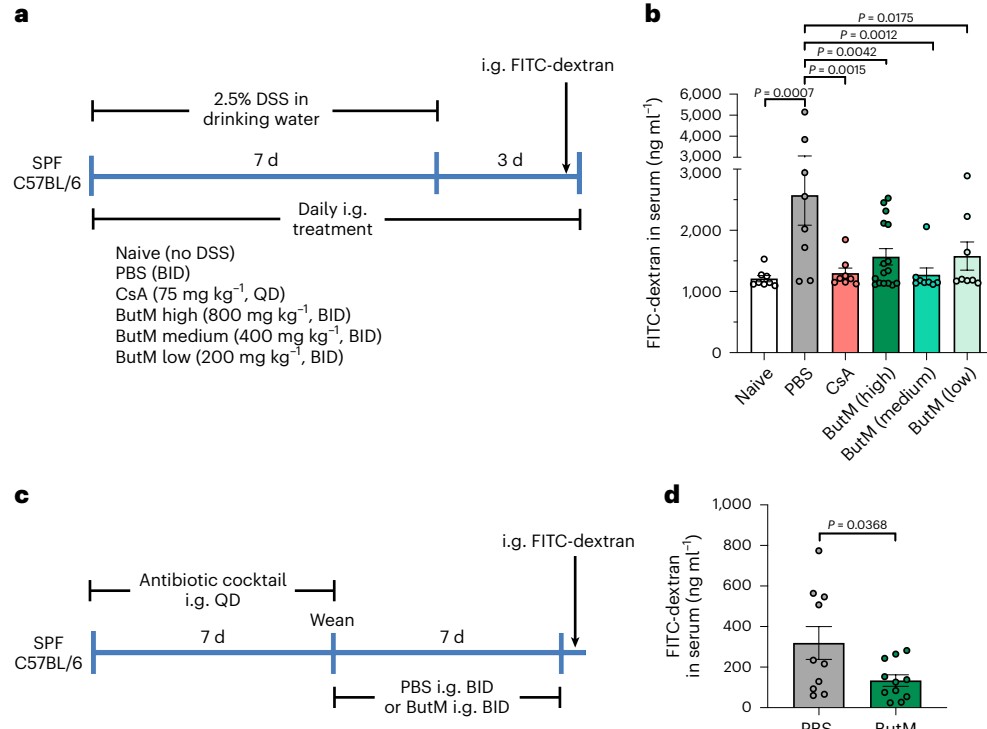

**Fig. 4 | Butyrate micelle treatment repaired intestinal barrier integrity in DSS-treated or antibiotic-treated mice. a**, Mice were given 2.5% DSS in the drinking water for 7 d to induce epithelial barrier dysfunction. DSS was removed from the drinking water on days 7–10. For treatment, mice were i.g. dosed daily with either PBS, CsA or ButM at different concentrations. QD, once a day; BID, twice daily at 10–12 h intervals. On day 10, all mice received an i.g. administration of 4 kDa FITC-dextran. Fluorescence was measured in the serum 4 h later. **b**, Concentration of FITC-dextran in the serum. $n = 8$ mice per group, except for high-dose ButM which had 16 mice per group. **c**, Mice were treated with a mixture of antibiotics, beginning at 2 weeks of age, for 7 d. After weaning, mice were i.g. administered with either PBS ($n = 10$) or ButM ($n = 11$) at 800 mg kg$^{-1}$ twice daily for 7 d. All mice then received an i.g. administration of 4 kDa FITC-dextran. Fluorescence was measured in the serum 1.5 h later. **d**, Concentration of FITC-dextran in the serum. Data in **d** are pooled from two independent experiments. Data represent mean ± s.e.m. Comparisons were made using one-way ANOVA with Dunnett's post-test (**b**) or two-sided Student's $t$-test (**d**).

an impaired intestinal barrier. Naïve mice (without DSS exposure) or the DSS-treated mice that also received either CsA or ButM at all three concentrations had similar serum levels of FITC-dextran, suggesting that treatment with ButM repaired the DSS-induced injury to the barrier (Fig. 4b). Additionally, neonatal antibiotic treatment impairs homoeostatic epithelial barrier function and increases permeability to food antigens[25]. Thus, we further evaluated whether ButM treatment can reduce intestinal barrier permeability in antibiotic-treated mice (Fig. 4c). In the antibiotic treatment model, serum was collected 1.5 h after FITC-dextran gavage (a timepoint at which the luminal FITC-dextran has transited through the ileum but has not yet reached the colon) to specifically analyse barrier permeability in the small intestine. Similar to what we observed in the DSS-induced model, mice treated with ButM had significantly lower FITC-dextran levels in the serum compared with mice that received PBS (Fig. 4d), demonstrating that ButM effectively rescued both DSS-induced and antibiotic-induced intestinal barrier dysfunction.

### Butyrate micelles ameliorate anaphylactic responses in peanut-allergic mice

To evaluate the efficacy of the butyrate-containing micelles in treating food allergy, we tested ButM in a well-established murine model of peanut-induced anaphylaxis[25,45]. All of the mice were treated with vancomycin to induce dysbiosis. Beginning at weaning, vancomycin-treated SPF C3H/HeN mice were intragastrically sensitized weekly for 4 weeks with peanut extract (PN) plus the mucosal adjuvant cholera toxin (CT) (Fig. 5a,b), as previously described[25,45]. Following sensitization, some of the mice were challenged with intraperitoneal (i.p.) PN and their

change in core body temperature was monitored to ensure that the mice were uniformly sensitized; a decrease in core body temperature is indicative of anaphylaxis (Fig. 5c). The rest of the sensitized mice were then treated i.g. twice daily for 2 weeks with either PBS or the combined micelle formulation ButM. After 2 weeks of therapy, the mice were challenged by i.p. injection of PN and their core body temperature was assessed to evaluate the response to allergen challenge. Compared with PBS-treated mice, allergic mice that were treated with ButM experienced a significantly reduced anaphylactic drop in core body temperature (Fig. 5d). In addition, ButM-treated mice also had significantly reduced concentrations of mouse mast cell protease-1 (mMCPT-1), histamine, peanut-specific IgE and peanut-specific IgG1 detected in the serum (Fig. 5e–h). Histamine and mMCPT-1 are released from degranulating mast cells upon allergen cross-linking of IgE and are reliable markers of anaphylaxis. mMCPT-1 is a chmyase expressed by intestinal mucosal mast cells; elevated concentrations of mMCPT-1 increase intestinal barrier permeability during allergic hypersensitivity responses[46,47]. While ButM very effectively reduced the allergic response to peanut exposure, free NaBut had no effect in protecting allergic mice from an anaphylactic response (Fig. 5i–k) and did not reduce serum peanut-specific IgE and IgG1 levels (Fig. 5l,m). This failure of NaBut to effectively reduce the allergic response to peanut is further evidence for the necessity to deliver butyrate to the lower GI tract. The effects of ButM on the peanut-allergic mice were dose-dependent, since we observed that reducing the dose of ButM by half was not as effective as the full dose in protecting mice from an anaphylactic response (Extended Data Fig. 4). We also evaluated and compared each of the two butyrate micelles (NtL-ButM and Neg-ButM) as monotherapies in

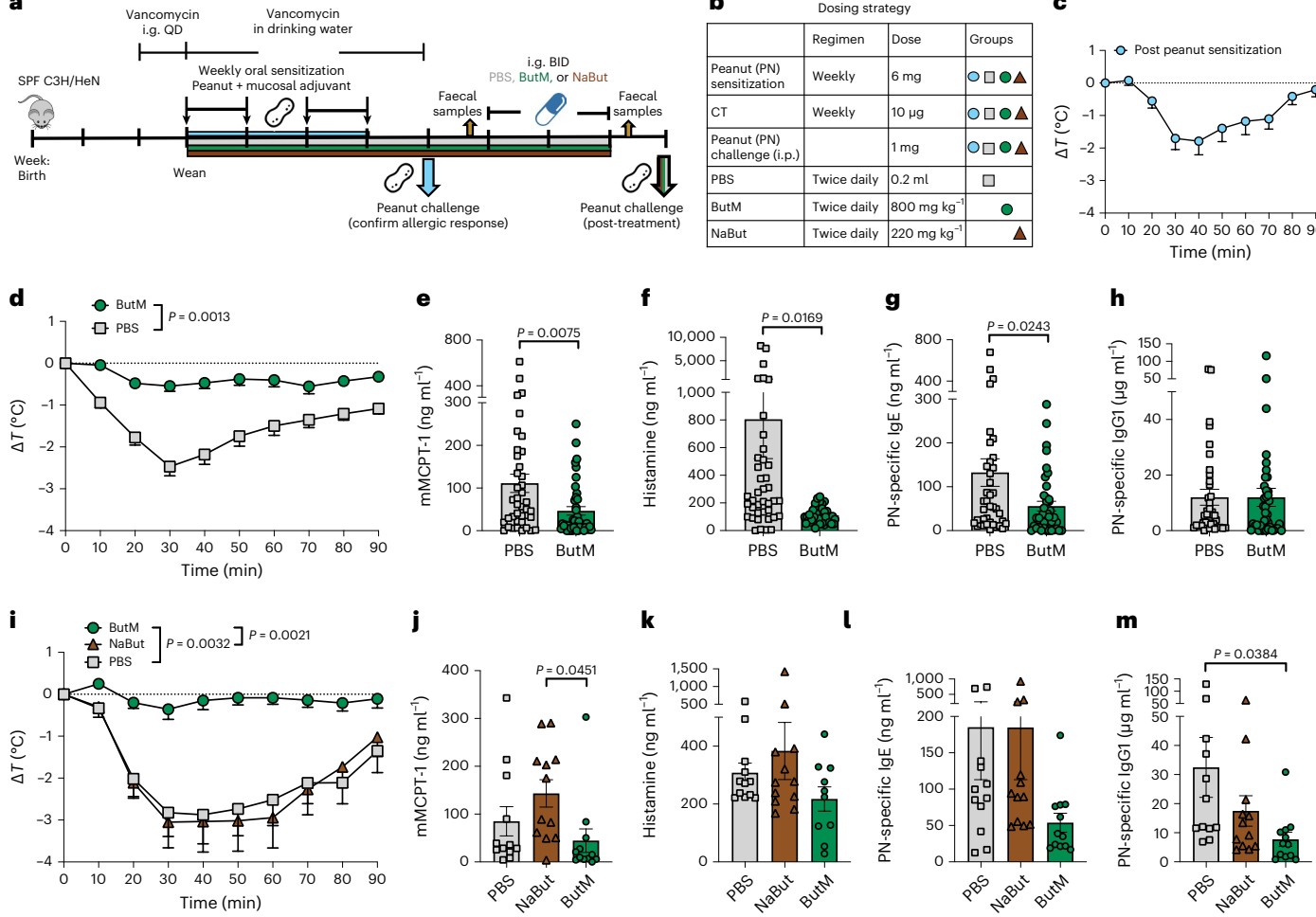

**Fig. 5 | Butyrate micelle treatment reduced the anaphylactic response to peanut challenge. a,b**, Experimental schema (**a**) and dosing strategy (**b**). All mice were sensitized weekly by intragastric gavage of 6 mg of peanut extract (PN) plus 10 μg of the mucosal adjuvant cholera toxin. After 4 weeks of sensitization, one group of mice (*n* = 20) was challenged by i.p. administration of 1 mg of PN to confirm that the sensitization protocol induced a uniform allergic response. **c**, Change in core body temperature following PN challenge where core body temperature drop indicates anaphylaxis. In the first study (**d**–**h**), the remaining mice were randomized into two treatment groups. One group was treated with PBS (*n* = 40) and the other group was treated with a 1:1 mix of NtL-ButM and Neg-ButM polymers at 400 mg kg⁻¹ each (ButM, *n* = 40). The data in **d**–**h** were pooled from four independent experiments. **d**, Change in core body temperature following challenge with PN in PBS- or ButM-treated mice. The AUC values were compared between the two groups. **e**–**h**, Serum mMCPT-1 (**e**), histamine (**f**), peanut-specific IgE (**g**) and peanut-specific IgG1 (**h**) from mice in **d**. In the second study (**i**–**m**), mice were treated with PBS (*n* = 12), ButM (*n* = 12) or sodium butyrate (NaBut, *n* = 12). **i**, Change in core body temperature following challenge with PN in PBS-, ButM- or NaBut-treated mice. **j**–**m**, Serum mMCPT-1 (**j**), histamine (**k**), peanut-specific IgE (**l**) and peanut-specific IgG1 (**m**) from mice in **i**. Data represent mean ± s.e.m. Data for **d**–**h** and **i**–**m** were analysed using two-sided Student's *t*-test and one-way ANOVA with Tukey's post-test, respectively.

treating peanut allergy in mice rather than the 1:1 combination (that is, ButM). Both NtL-ButM and Neg-ButM administered individually significantly reduced the anaphylactic response to peanut challenge, although not as effectively as the ButM polymer combination (Extended Data Fig. 4). Together, these results demonstrate that ButM is highly effective in preventing allergic responses to food in sensitized mice.

## Butyrate micelles alter the faecal microbiota and promote recovery of Clostridia after antibiotic exposure
We next examined whether treatment with ButM altered the faecal microbiome. In the mouse model of peanut allergy described above, we induced dysbiosis by treating mice with vancomycin 1 week before the start of allergen sensitization and throughout the sensitization regimen. Vancomycin depletes Gram-positive bacteria, including Clostridial species[48]. After sensitization, we removed vancomycin from the drinking water and compared the faecal microbial composition of the allergic mice before and after treatment with PBS or ButM

(see timepoints collected in Fig. 5a). 16S ribosomal RNA targeted sequencing confirmed depletion of Clostridia in vancomycin-treated mice; the faecal microbiota was instead dominated by *Lactobacillus* and Proteobacteria (Fig. 6a, left). After halting vancomycin administration, regrowth of Clostridia (including Lachnospiraceae and others) and Bacteroidetes was observed in both the PBS- and ButM-treated groups. This regrowth of Clostridia, and corresponding decrease in relative abundance of *Lactobacillus* and Proteobacteria, can be observed in the overall taxonomic composition (Fig. 6a, right) or by LEfSe analysis comparing pre- and post-treatment timepoints within each treatment group (Supplementary Fig. 18). When comparing differentially abundant taxa between treatment groups by LEfSe analysis, *Murimonas* and *Streptococcus* were significantly higher in relative abundance in the PBS post-treatment group compared with the ButM group (Fig. 6b). ButM treatment significantly increased the relative abundance of *Enterococcus*, *Coprobacter* and *Clostridium* cluster XIVa (Fig. 6b). *Clostridium* cluster XIVa is a numerically predominant

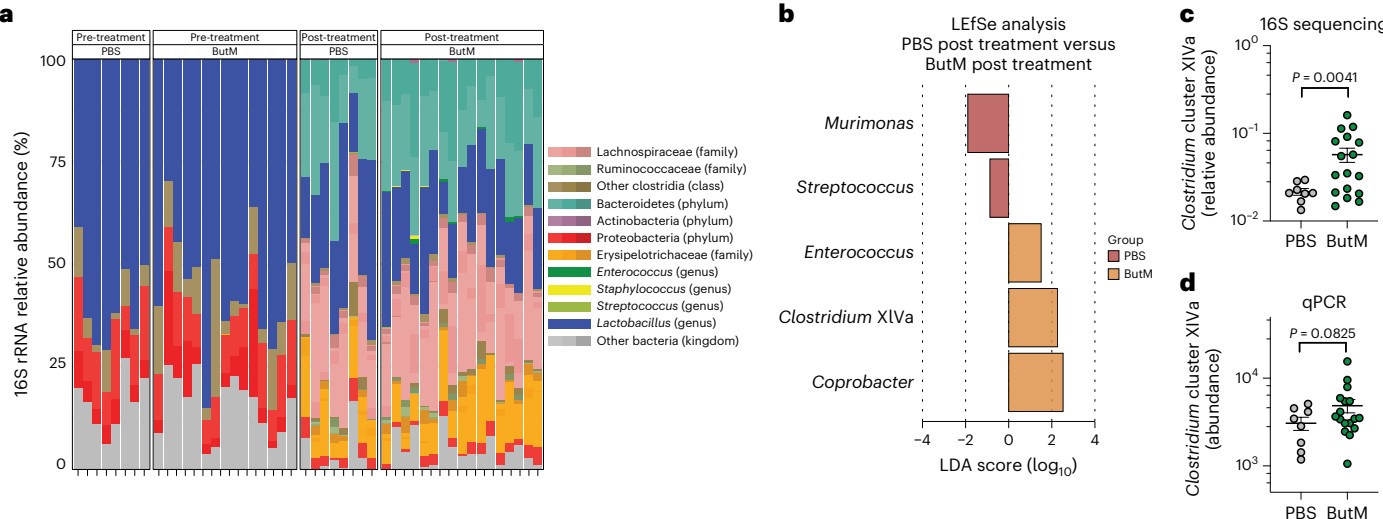

**Fig. 6 | Butyrate micelle treatment increased the abundance of *Clostridium* cluster XIVa. a**, 16S rRNA sequencing analysis of relative abundance of bacterial taxa in faecal samples of allergic mice collected before (left) or after (right) treatment with PBS (*n* = 8) or ButM (*n* = 17) from experiments outlined in Fig. 5. **b**, Differentially abundant taxa between mice treated with PBS or ButM

after treatment as analysed by LEfSe. **c**,**d**, Relative abundance of *Clostridium* cluster XIVa in faecal samples after treatment with PBS or ButM (from **a**) analysed from 16S sequencing data (**c**) or analysed by qPCR (**d**). Data for **c** and **d** represent mean ± s.e.m. Statistics analysed using two-sided Student's *t*-test with Welch's correction.

group of bacteria (in both mice and humans) that is known to produce butyrate, modulate host immunity and induce $T_{reg}$ cells[49]. We confirmed that the relative abundance of *Clostridium* cluster XIVa in mice treated with ButM was significantly increased in the 16S data set (Fig. 6c) and quantified the enriched abundance of this taxa by qPCR (Fig. 6d). Our finding of increased abundance of *Clostridium* cluster XIVa after treatment with ButM is in keeping with earlier work showing that butyrate sensing by peroxisome proliferator-activated receptor (PPAR-γ) shunts colonocyte metabolism toward β-oxidation, creating a local hypoxic niche for these oxygen-sensitive anaerobes[50]. We performed a second experiment to examine the effects of ButM treatment on the composition of the microbiota in the absence of allergic sensitization (to rule out a confounding influence of the sensitization protocol itself while holding all other variables constant). We found a similar increase in the relative abundance of Clostridia in the faeces and ileal contents of mice treated with ButM (Extended Data Fig. 5). In this experiment, the relative abundance of *Clostridium* cluster IV was increased in the faeces, and *Clostridium* cluster XVIII was increased in both faeces and ileal contents.

To gain additional mechanistic insights, we also evaluated the effect of ButM on immune cells in the peanut allergy model. We treated peanut-sensitized mice with either PBS, NaBut or ButM for 2 weeks and examined the effects of ButM treatment on the abundance of $T_{reg}$ cells (Extended Data Fig. 6) and on myeloid cell activation (Extended Data Fig. 7). We found that ButM did not affect the proportion or number of FoxP3⁺CD25⁺ $T_{reg}$ cells in the spleen, ileum or colon-draining lymph nodes (LNs) (Extended Data Fig. 6 and Supplementary Fig. 19). However, ButM, but not NaBut, significantly downregulated the expression of MHC Class II and the costimulatory marker CD86 on cells in the CD11c^hi, CD11b⁺F4/80⁺ and CD11b⁺CD11c⁻ compartments from both ileum and colon-draining LNs (Extended Data Fig. 7 and Supplementary Fig. 20). These data suggest a potential mechanism of action of ButM through suppression of myeloid cell activation in the mesenteric LNs.

### Butyrate micelles decrease severity of colitis in the CD45RB^hi T-cell-transfer model

Finally, we examined the efficacy of ButM in the CD45RB^hi T-cell-transfer model. In this model (Fig. 7a), adult SPF C57BL/6 *Rag2*⁻/⁻ mice (which lack all lymphocytes) receive i.p. injections of purified CD4⁺ splenic

T cells from SPF C57BL/6 mice. The transfer of CD45RB^hi naive effector T cells into a lymphopenic host induces acute inflammation in the colonic tissue and pancolitis[51]. Co-administration of CD45RB^lo CD25⁺ $T_{reg}$ cells prevents the development of colitis. The control group received a standard transfer of only CD4 T effectors (CD4⁺CD45RB^hi) to demonstrate maximal disease severity. Other experimental groups received co-transfer of CD4⁺CD45RB^hi cells plus $T_{reg}$ cells (CD4⁺CD25⁺CD45RB^lo; see table of treatment groups in Fig. 7b). Because butyrate can induce $T_{reg}$ cells[17–19], a small number was transferred such that $T_{reg}$ cells alone were not sufficient to fully prevent disease progression. Some groups that received the co-transfer of $T_{reg}$ cells were then treated, beginning at 14 d after adoptive transfer, with one of three doses of ButM (low, medium, high) twice daily for the duration of the experiment. All mice began to develop colitic symptoms around day 21, demonstrated by loss of body weight as percent of initial weight (Fig. 7c). Co-transfer of $T_{reg}$ cells did not improve body weight compared with standard transfer, as expected for the low number of $T_{reg}$ cells transferred. However, treatment with any dose of ButM noticeably improved body weight retention from day 30 to 39 (Fig. 7c, efficacy phase). The percent change in body weight on day 35 was significantly improved versus the co-transfer control in all three ButM treatment groups (Fig. 7d). On the basis of the results obtained in Fig. 6 and Extended Data Fig. 5, we predicted that after 2 weeks of twice daily treatment, the efficacy of the ButM micelles would be mediated, at least in part, by an expansion of butyrate-producing Clostridia. We therefore administered vancomycin to each of the ButM treatment groups daily from day 30–34 to deplete butyrate-producing Clostridia. As predicted, the efficacy of ButM was reduced after vancomycin treatment (Fig. 7c, reversal phase) and the percent body weight change in the ButM-treated mice (at all doses) was similar to that of co-transfer controls (Fig. 7c,e). However, even after the reversal of efficacy induced by vancomycin, mice treated with the medium dose of ButM had a significantly decreased colon weight-to-length ratio compared with co-transfer controls, indicative of less severe disease (Fig. 7f). Representative images of the distal colon of each treatment group are shown (Fig. 8a), highlighting areas of most severely affected mucosa (M) and areas of oedema (E). Histopathological analysis demonstrated that the medium dose of ButM also significantly reduced the amount of oedema (Fig. 8b) and reduced the occurrences of polymorphonuclear leucocyte cell

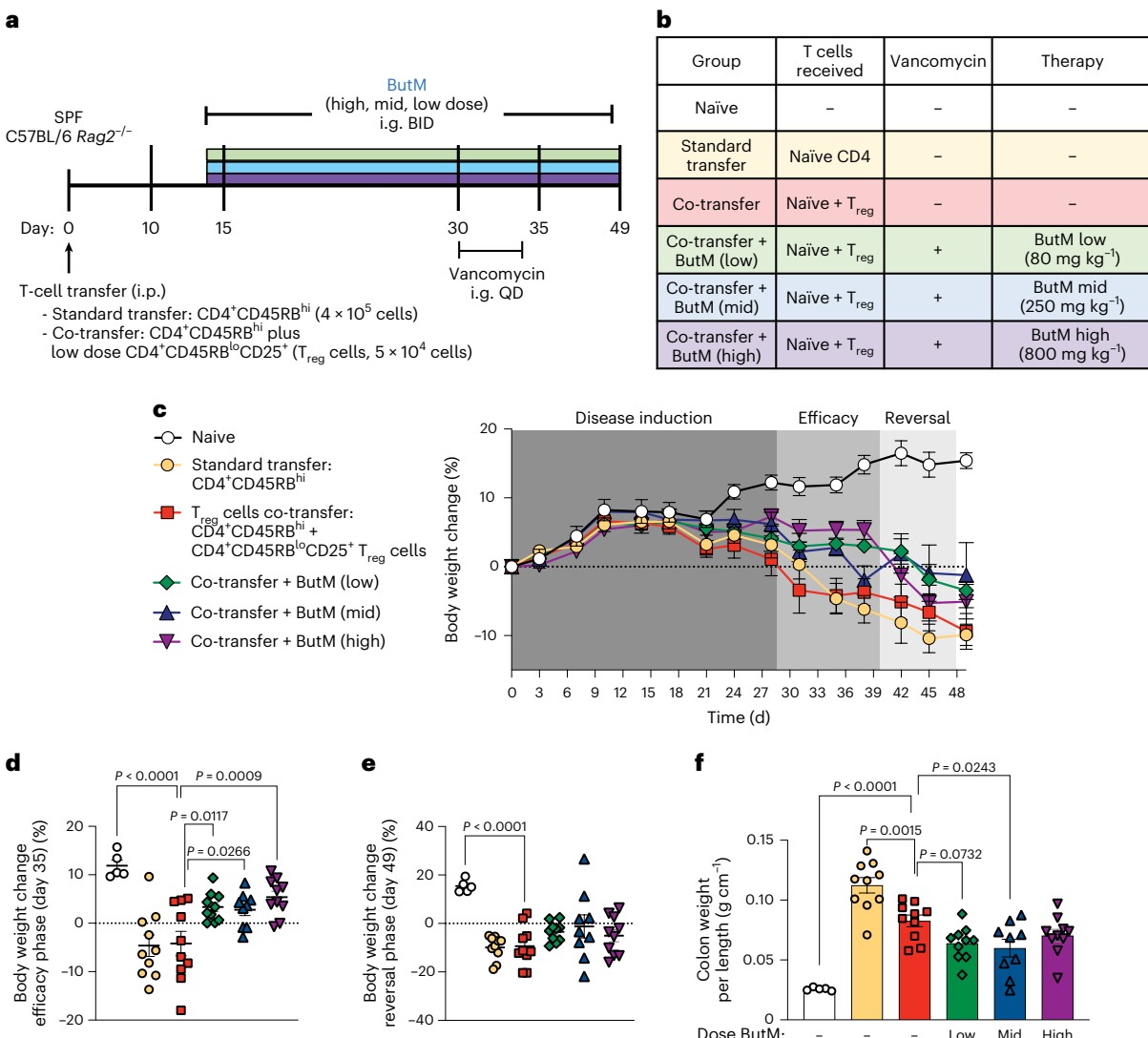

**Fig. 7 | ButM decreases severity of T-cell transfer colitis. a**, Experimental design. On day 0, *Rag2*[−/−] mice received an i.p. injection of purified naïve CD4 T cells (CD4[+]CD45RB[hi]) alone (standard transfer) or together with CD4[+]CD45RB[lo]CD25[+] T$_{reg}$ cells (co-transfer). Naïve animals received a sham injection. Some groups were gavaged twice daily with ButM at a low (80 mg kg[−1]), medium (250 mg kg[−1]) or high (800 mg kg[−1]) dose from day 14 until the end of the experiment. ButM-treated mice were also gavaged with vancomycin (0.1 mg) from day 30–34. **b**, Table detailing treatment groups. **c**, Body weight change as a percentage of starting weight over the duration of the study window. **d,e**, Body weight change (% of initial weight) from representative days during the efficacy (**d**; day 35) and reversal (**e**; day 49) phases of the study. **f**, Weight to length ratio of colons collected on day 49. In **c**, circles represent mean ± s.e.m. for all mice. In **d–f**, dots represent individual mice, and bars represent mean ± s.e.m. Statistics analysed by one-way ANOVA with Dunnett's multiple comparison's test comparing all groups against the T$_{reg}$ co-transfer group. *n* = 5 naïve mice, *n* = 10 for experimental groups.

infiltrates (Fig. 8c) across the colon. Additionally, the medium dose of ButM modestly decreased the neutrophil score (Fig. 8d) and the low dose modestly decreased the hyperplasia score (Fig. 8e) in the distal colon. These results show that ButM reduces disease severity in the CD45RB[hi] T-cell-transfer model of colitis, and that this effect is reversed by treatment with vancomycin.

## Discussion

The prevalence of NCCDs, including food allergy and inflammatory bowel diseases, has increased dramatically over the past 20–40 years, particularly in developed countries[11,52,53]. Lifestyle changes such as reduced consumption of dietary fibre, increased antibiotic use (including in the food chain) and sanitation have altered populations of commensal microbes. These alterations lead to several negative health effects, including the impairment of intestinal barrier function. Modulating the gut microbiome to redirect immunity has become a

substantial effort in both academia and industry. However, this has proven difficult: getting selected bacteria, especially obligate anaerobes, to colonize the gut is far from straightforward. Here we have instead focused on delivering the metabolites that are produced by these bacteria in a more direct manner, since their therapeutic efficacy relies largely on the action of their metabolites.

Specifically, we developed a polymeric nanoscale system to deliver butyrate to localized regions along the GI tract. The system was based on polymeric micelles formed by block copolymers, in which butyrate is conjugated to the hydrophobic block by an ester bond and can be hydrolysed by esterases in the GI tract for local release. The linked butyrate moieties drive hydrophobicity in that block and, as release occurs, the remainder of the construct (an inert, water-soluble polymer) continues to transit through the lower GI tract until it is excreted. The butyrate-containing block, when forming the core of micelles, was resistant to the acidic environment found in the stomach, which

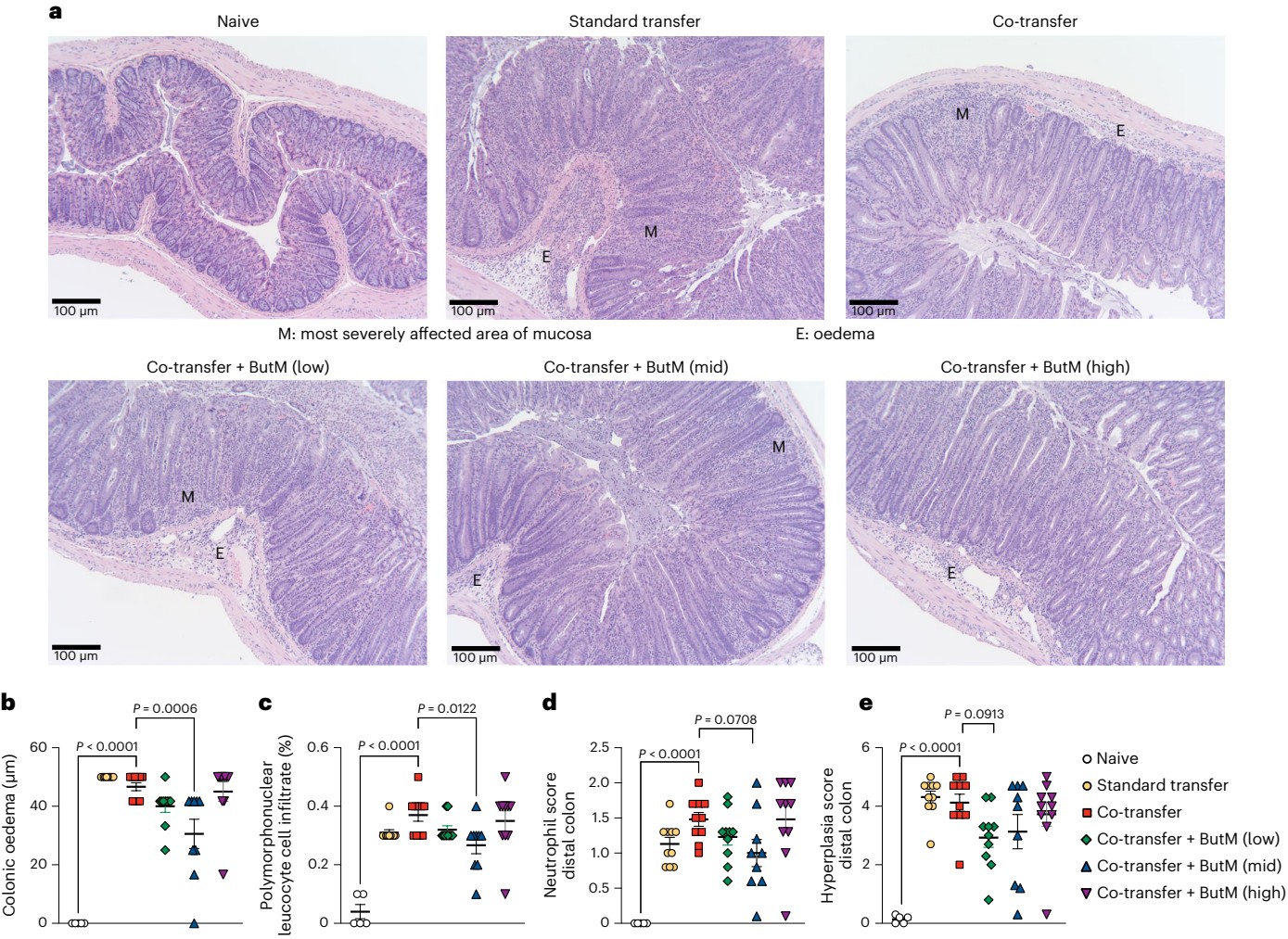

M: most severely affected area of mucosa

E: oedema

**Fig. 8 | Histopathological analysis of the T-cell-transfer colitis model in Fig. 7. a**, Representative images of transverse sections of distal colon tissue from mice in each treatment group. Tissues were stained with H&E and images were taken at ×100. Scale bar, 100 μm. **b**, Length of oedemas from colon histology. **c**, Abundance of polymorphonuclear cell infiltrates (% of cells) from colon histology. **d**, Neutrophil score in the distal colon from histological analysis. **e**, Hyperplasia score in the distal colon from histological analysis. Experiment performed by Inotiv. In **b**–**e**, dots represent average of all histopathological images per mouse, and bars represent mean ± s.e.m. across mice. Statistics analysed by one-way ANOVA with Dunnett's multiple comparison's test comparing all groups against the $T_{reg}$ co-transfer group. $n = 5$ naïve mice, $n = 10$ for experimental groups.

might prevent a burst release there before the micelle's transit into the intestine. The two butyrate-prodrug micelles, NtL-ButM and Neg-ButM, share similar structures but have corona charges of neutral and negative, respectively. This results in their distinct biodistribution in the lower GI tract, where they can release butyrate in the presence of enzymes.

We combined both neutral and negatively charged micelles to deliver butyrate along the distal gut. We showed that this combined formulation preserves barrier function, reduces severity of colitis and protects from severe anaphylactic responses to food with a short-term treatment. In our mouse model of peanut allergy, where the mice were previously exposed to vancomycin to induce dysbiosis, we showed that ButM treatment could favourably increase the relative abundance of protective bacteria, such as *Clostridium* cluster XIVa. A clinical trial showed that bacteria in the *Clostridium* cluster XIVa may be critical to the success of FMT for treatment of colitis[27]. Increasing the abundance of these bacteria may be one mechanism by which ButM treatment could improve epithelial barrier function after exposure to DSS or antibiotics or in the pre-clinical models of food allergy or colitis.

We conducted an experiment to further analyse the effects of ButM on immune cells in the peanut-allergic mice. When we examined the abundance of $T_{reg}$ cells in the mesenteric LNs and spleen, we did not observe any differences among treatment groups, suggesting that ButM did not affect these $T_{reg}$ populations in this model. We further evaluated whether ButM had any effect on the myeloid cells in peanut-allergic mice. ButM treatment significantly downregulated the expression of MHC Class II and the costimulatory marker CD86 on the dendritic cells and macrophages in the colon-draining and ileal-draining LNs, suggesting downmodulation of activation and antigen presentation. It has also been previously shown that butyrate inhibits mast cell activation through FcεRI-mediated signalling[54]. This is also consistent with the absence of hypothermia and elevated serum mMCPT-1 and histamine (all indicators of anaphylaxis) after peanut exposure in ButM-treated mice.

Investigations on the therapeutic potential of butyrate in animal models have supplemented butyrate in the drinking water or diet at a high dose for three or more weeks[15–21]. Such dosing to achieve therapeutic effects from sodium butyrate is challenging for clinical translation owing to the uncontrolled dosing regimen, difficulties with replication in humans, and the unpleasant odour and taste of butyrate as a sodium salt. Our formulation has incorporated butyrate in the polymeric micelles at a high load (28 wt%) and is able to deliver and release most of the butyrate in the lower GI tract in a manner that

masks butyrate's taste and smell. We used a daily dose of 800 mg kg$^{-1}$ of total ButM to treat peanut-allergic mice for 2 weeks. This can be translated to ~65 mg kg$^{-1}$ of total ButM (or equivalent butyrate dose of 18.2 mg kg$^{-1}$) human dose given the differences in body surface area between rodents and humans[55]. This butyrate dose in ButM micelles is comparable to other butyrate dosage forms that have been tested clinically[56,57]; however, through local targeting and sustained release in the lower GI tract, we expect our ButM formulation to achieve higher therapeutic potential in food allergies and beyond.

Our approach is not antigen-specific, because antigen delivery was not part of the treatment regimen, so our proof-of-concept in peanut allergy could be readily extended to other food allergens, such as other nuts, milk, egg, soy and shellfish. Additionally, the approach may be applicable to inflammatory bowel disease and to other diseases caused by hyperinflammation along the GI tract. Moreover, the technology could also be easily adapted to deliver other SCFAs or other microbiome-derived metabolites in a single form or in combination, providing a more controlled and accessible way to achieve potential therapeutic efficacy.

## Methods

### Materials for polymer synthesis

*N*-(2-hydroxyethyl) methacrylamide (HPMA) monomer was obtained from Sigma Aldrich or Polysciences. Solvents including dichloromethane, methanol, hexanes and ethanol were ACS reagent grade and were obtained from Thermo Fisher. All other chemicals were obtained from Sigma Aldrich.

### Synthesis of *N*-(2-hydroxyethyl) methacrylamide (2)

To synthesize *N*-(2-hydroxyethyl) methacrylamide (HEMA, **2**), ethanolamine (3.70 ml, 61.4 mmol, 2.0 eq), triethylamine (4.72 ml, 33.8 mmol, 1.1 eq) and 50 ml dichloromethane (DCM) were added into a 250 ml flask. After the system was cooled in an ice bath, methacryloyl chloride (1, 3.00 ml, 30.7 mmol, 1.0 eq) was added dropwise under the protection of nitrogen. The reaction was allowed to warm to room temperature and reacted overnight. Then the reaction mixture was concentrated by rotary evaporation and purified on a silica column using DCM/MeOH (MeOH fraction v/v, 0–5%). The product was obtained as a colourless oil (3.42 g, 86.3%). The mass-to-charge ratio ($m/z$) of $C_6H_{11}NO_2$ $(M + H)^+$ was calculated as 129.08 and measured as 129.0 by electrospray ionization mass spectrometry (ESI–MS). $^1$H-NMR (500 MHz, CDCl$_3$) δ 1.93 (s, 3H), 3.43 (m, 2H), 3.71 (m, 2H), 5.32 (s, 1H), 5.70 (s, 1H), 6.44 (br s, 1H) (Supplementary Fig. 1 and Scheme 1).

### Synthesis of *N*-(2-butanoyloxyethyl) methacrylamide (3)

To synthesize *N*-(2-butanoyloxyethyl) methacrylamide (BMA, **3**), *N*-(2-hydroxyethyl) methacrylamide (3.30 ml, 25.6 mmol, 1.0 eq), triethylamine (7.15 ml, 51.2 mmol, 2.0 eq) and 50 ml DCM were added into a 250 ml flask. After the reaction system was cooled in an ice bath, butyric anhydride (5.00 ml, 30.7 mmol, 1.2 eq) was added dropwise under the protection of nitrogen. The system was allowed to react overnight. The reaction mixture was filtered and washed with NH$_4$Cl solution, NaHCO$_3$ solution and water. After drying using anhydrous MgSO$_4$, the organic layer was concentrated by rotary evaporation and purified on a silica column using DCM/MeOH (MeOH fraction v/v, 0–5%). The product was obtained as a pale-yellow oil (4.56 g, 89.6%). The $m/z$ of $C_{10}H_{17}NO_3 (M + H)^+$ was calculated as 199.12 and measured as 199.1 by ESI–MS. $^1$H-NMR (500 MHz, CDCl$_3$) δ 0.95 (t, 3H), 1.66 (m, 2H), 1.97 (s, 3H), 2.32 (t, 2H), 3.59 (dt, 2H), 4.23 (t, 2H), 5.35 (s, 1H), 5.71 (s, 1H), 6.19 (br s, 1H) (Supplementary Fig. 2 and Scheme 2).

### Synthesis of poly(2-hydroxypropyl methacrylamide) (pHPMA, 5)

pHPMA was prepared using 2-cyano-2-propyl benzodithioate as the RAFT chain transfer agent and 2,2′-azobis(2-methylpropionitrile) (AIBN) as the initiator. Briefly, HPMA (**4**, 3.0 g, 20.9 mmol, 1.0 eq),

2-cyano-2-propyl benzodithioate (28.3 mg, 0.128 mmol, 1/164 eq) and AIBN (5.25 mg, 0.032 mmol, 1/656 eq) were dissolved in 10 ml MeOH in a 25 ml Schlenk tube. The reaction mixture was subjected to four freeze-pump-thaw cycles. The polymerization was conducted at 70 °C for 30 h. The polymer was precipitated in a large volume of petroleum ether and dried in a vacuum chamber overnight. The product obtained was a light pink solid (1.8 g, 60%). $^1$H-NMR (500 MHz, DMSO-d6) δ 0.8–1.2 (m, 6H, CH(OH)-CH$_3$ and backbone CH$_3$), 1.5–1.8 (m, 2H, backbone CH$_2$), 2.91 (m, 2H, NH-CH$_2$), 3.68 (m, 1H, C(OH)-H), 4.70 (m, 1H, CH-OH), 7.18 (m, 1H, NH) (Supplementary Fig. 3 and Scheme 3).

### Synthesis of pHPMA-b-pBMA (6)

The block copolymer pHPMA-b-pBMA was prepared using pHPMA (**5**) as the macro-RAFT chain transfer agent and BMA (**3**) as the monomer of the second RAFT polymerization. Briefly, pHPMA (1.50 g, 0.105 mmol, 1.0 eq), *N*-(2-butanoyloxyethyl) methacrylamide (4.18 g, 21.0 mmol, 200 eq) and AIBN (8.3 mg, 0.050 mmol, 0.50 eq) were dissolved in 10 ml MeOH in a 50 ml Schlenk tube. The reaction mixture was subjected to four freeze-pump-thaw cycles. The polymerization was conducted at 70 °C for 20 h. The polymer was precipitated in petroleum ether and dried in the vacuum chamber overnight. The product obtained was a light pink solid (4.22 g, 74%). $^1$H-NMR (500 MHz, DMSO-d6) δ 0.80–1.1 (m, 9H, CH(OH)-CH$_3$ (HPMA), CH$_2$-CH$_3$ (BMA), and backbone CH$_3$), 1.55 (m, 4H, CH$_2$-CH$_2$ (BMA) and backbone CH$_2$), 2.28 (m, 2H, CO-CH$_2$ (BMA)), 2.91 (m, 2H, NH-CH$_2$ (HPMA)), 3.16 (m, 2H, NH-CH$_2$ (BMA)), 3.67 (m, 1H, CH(OH)-H), 3.98 (m, 2H, O-CH$_2$ (BMA)), 4.71 (m, 1H, CH-OH (HPMA)), 7.19 (m, 1H, NH), 7.44 (m, 1H, NH) (Supplementary Fig. 4 and Scheme 4).

### Synthesis of pMAA (7) and pMAA-b-pBMA (8)

pMAA (**7**) was prepared using 2-cyano-2-propyl benzodithioate as the RAFT chain transfer agent and AIBN as the initiator. Briefly, methacrylic acid (MAA) (4.0 ml, 47.2 mmol, 1.0 eq), 2-cyano-2-propyl benzodithioate (104.4 mg, 0.472 mmol, 1/100 eq) and AIBN (19.4 mg, 0.118 mmol, 1/400 eq) were dissolved in 20 ml MeOH in a 50 ml Schlenk tube. The reaction mixture was subjected to four freeze-pump-thaw cycles. The polymerization was conducted at 70 °C for 24 h. The polymer was precipitated in hexanes and dried in the vacuum oven overnight. The product obtained was a light pink solid (4.0 g, 100%). $^1$H-NMR (500 MHz, DMSO-d6) δ 0.8–1.2 (m, 3H, backbone CH$_3$), 1.5–1.8 (m, 2H, backbone CH$_2$), 7.4–7.8 (three peaks, 5H, aromatic H), 12.3 (m, 1H, CO-OH) (Supplementary Fig. 5 and Scheme 5).

The block copolymer pMAA-b-pBMA (**8**) was prepared using pMAA (**7**) as the macro-RAFT chain transfer agent and BMA (**3**) as the monomer of the second RAFT polymerization. Briefly, pMAA (0.50 g, 0.058 mmol, 1.0 eq), *N*-(2-butanoyloxyethyl) methacrylamide (1.47 g, 7.38 mmol, 127 eq) and AIBN (2.4 mg, 0.015 mmol, 0.25 eq) were dissolved in 10 ml MeOH in a 25 ml Schlenk tube. The reaction mixture was subjected to four freeze-pump-thaw cycles. The polymerization was conducted at 70 °C for 24 h. The polymer was precipitated in hexanes and dried in the vacuum oven overnight. The product obtained was a light pink solid (1.5 g, 70%). $^1$H-NMR (500 MHz, DMSO-d6) δ 0.8–1.1 (m, 6H, CH$_2$-CH$_3$ (BMA), and backbone CH$_3$), 1.5–1.7 (m, 4H, CH$_2$-CH$_2$ (BMA) and backbone CH$_2$), 2.3 (m, 2H, CO-CH$_2$ (BMA)), 3.2 (m, 2H, NH-CH$_2$ (BMA)), 4.0 (m, 2H, O-CH$_2$ (BMA)), 7.4 (m, 1H, NH), 12.3 (m, 1H, CO-OH) (Supplementary Fig. 6 and Scheme 5).

### Synthesis of N$_3$-PEG$_4$-MA (9) and azide-PEG polymer

To include an azide group into pHPMA-b-pBMA or pMAA-b-pBMA polymers, monomer *N*-(2-(2-(2-(2-azidoethoxy)ethoxy)ethoxy)ethyl) methacrylamide (**9**) was synthesized and used in the copolymerization with HPMA or MAA to obtain the hydrophilic block with azide function. Briefly, N$_3$-PEG$_4$-NH$_2$ (0.5 g, 2.14 mmol, 1.0 eq) and triethylamine (0.60 ml, 4.3 mmol, 2.0 eq) were dissolved in anhydrous DCM. After the reaction system was cooled in an ice bath, methacrylic chloride (0.42 ml, 2.6 mmol, 1.2 eq) was added dropwise under the protection

of nitrogen. The system was allowed to react overnight. The reaction mixture was filtered and washed with $NH_4Cl$ solution, $NaHCO_3$ solution and water. After being dried using anhydrous $MgSO_4$, the organic layer was concentrated by rotary evaporation and purified on a silica column using DCM/MeOH (MeOH fraction v/v, 0–5%). The product obtained was a pale-yellow oil (0.47 g, 73%). The $m/z$ of $C_{12}H_{22}N_4O_4$ $(M + H)^+$ was calculated as 287.16 and measured as 287.2 by ESI–MS. $^1$H-NMR (500 MHz, $CDCl_3$) δ 6.35 (br, 1H), 5.70 (s, 1H), 5.32 (s, 1H), 3.55–3.67 (m, 12H), 3.52 (m, 2H), 3.38 (t, 2H), 1.97 (s, 3H) (Supplementary Fig. 7 and Scheme 6). Monomer $N_3$-$PEG_4$-MA was mixed with HPMA or MAA in a 2:98 w:w ratio during the RAFT polymerization to obtain $N_3$-pHPMA or $N_3$-pMAA. Then, the second block of BMA was added to the macro initiator to obtain $N_3$-pHPMA-b-pBMA or $N_3$-pMAA-b-pBMA, respectively. The synthesis procedures were the same as the previous description.

## Synthesis of *N*-hexyl methacrylamide (10) and control polymer

To synthesize a control polymer that did not contain butyrate ester, monomer *N*-hexyl methacrylamide (**10**) was synthesized and used in the polymerization of the hydrophobic block. Briefly, hexanamine (5.8 ml, 46.0 mmol, 1.5 eq), triethylamine (4.7 ml, 33.8 mmol, 1.1 eq) and 50 ml DCM were added into a 250 ml flask. After the system was cooled in an ice bath, methacryloyl chloride (3.0 ml, 30.7 mmol, 1.0 eq) was added dropwise under the protection of nitrogen. The reaction was allowed to warm to room temperature and reacted overnight. Then the reaction mixture was concentrated by rotary evaporation and purified on a silica column using DCM/MeOH (MeOH fraction v/v, 0–5%). The product obtained was a colourless oil (4.6 g, 88%). The $m/z$ of $C_{11}H_{21}NO$ $(M + H)^+$ was calculated as 184.16 and measured as 184.2 by ESI–MS. $^1$H-NMR (500 MHz, $CDCl_3$) δ 5.75 (br, 1H), 5.66 (s, 1H), 5.30 (s, 1H), 3.31 (t, 2H), 1.96 (s, 3H), 1.54 (m, 2H), 1.28–1.32 (m, 8H), 0.88 (t, 3H) (Supplementary Fig. 8 and Scheme 7). After the synthesis of pHPMA or pMAA, monomer *N*-hexyl methacrylamide (**10**) was used in the polymerization of the second block instead of *N*-(2-butanoyloxyethyl) methacrylamide to obtain control polymers as pHPMA-b-pHMA or pMAA-b-pHMA, respectively (Supplementary Fig. 9). The synthesis procedures were the same as described above.

## Formulation of polymeric micelles

NtL-ButM micelle was formulated using the cosolvent evaporation method. pHPMA-b-pBMA polymer (80 mg) was dissolved in 10 ml ethanol under stirring. After the polymer was completely dissolved, the same volume of 1× PBS was added slowly to the solution. The solution was allowed to evaporate at room temperature for at least 6 h until ethanol was removed. After the evaporation, the NtL-ButM solution was filtered through a 0.22 μm filter and stored at 4 °C. The size of the micelles was measured by DLS.

Neg-ButM micelle was prepared by base titration[32,33]. pMAA-b-pBMA polymer (60 mg) was added to 8 ml of 1× PBS under vigorous stirring. Sodium hydroxide solution in molar equivalent to methacrylic acid was added to the polymer solution in three portions over the course of 2 h. After adding base solution, the polymer solution was stirred at room temperature overnight. PBS (1×) was then added to reach the target volume and the solution was filtered through a 0.22 μm filter. The pH of the solution was checked to confirm it was neutral, and the size of the micelles was measured by DLS.

## DLS characterization of micelles

DLS data were obtained from a Zetasizer Nano ZS90 (Malvern Instruments). Samples were diluted 400 times in 1× PBS and 700 μl was transferred to a DLS cuvette for data acquisition. The intensity distributions of DLS were used to determine the hydrodynamic diameter of micelles. For *z*-potential data, micelles were diluted 100 times in 0.1× PBS (1:10 of 1× PBS to MilliQ water) and transferred to disposable folded capillary zeta cells for data acquisition.

## CryoEM imaging of micelles

CryoEM images were acquired on an FEI Talos 200 kV FEG electron microscope. Polymeric nanoparticle samples were prepared in 1× PBS and diluted to 2 mg ml$^{-1}$ with MilliQ water. Sample solution (2 μl) was applied to the electron microscopy grid (Agar Scientific) with holey carbon film. Sample grids were blotted and flash vitrified in liquid ethane using an automatic plunge freezing apparatus (Vitrbot) to control humidity (100%) and temperature (20 °C). Analysis was performed at −170 °C using the Gatan 626 cry-specimen holder (×120,000 magnification; −5 μm defocus). Digital images were recorded on an in-line Eagle CCD camera and processed using ImageJ.

## Measurement of critical micelle concentration

The critical micelle concentrations of NtL-ButM and Neg-ButM were determined by a fluorescence spectroscopic method using pyrene as a hydrophobic fluorescent probe[35,58]. A series of polymer solutions with concentration ranging from $1.0 × 10^{-4}$ to 2.0 mg ml$^{-1}$ were mixed with pyrene solution with a concentration of $1.2 × 10^{-3}$ mg ml$^{-1}$. The emission spectra of samples were recorded on a fluorescence spectrophotometer (HORIBA Fluorolog-3) at 20 °C using 335 nm as excitation wavelength. The ratio between the first (372 nm) and the third (383 nm) vibronic bands of pyrene was plotted against the concentration of the polymer. The data were processed on Prism software and fitted using Sigmoidal model (Supplementary Fig. 11).

## SAXS analysis of micelles

SAXS samples were made in 1× PBS and filtered through 0.2 μm filters. Data for all samples were acquired at the Stanford Synchrotron Radiation Lightsource, SLAC National Accelerator Laboratory. SAXS data were analysed using the Igor Pro 8 software (Supplementary Fig. 12). To acquire the radius of gyration ($R_g$), data were plotted as ln (intensity) versus $q^2$ at low $q$ range. $I(0)$ is extrapolated intensity at origin. $q$ represents momentum transfer, and $I(q)$ represents scattered intensity. Then $R_g$ was calculated from the slope of the linear fitting as shown in equation (1).

$$\ln(I(q)) = \ln(I(0)) - \frac{q^2 R_g^2}{3} \quad (1)$$

Kratky plots of the data were plotted from $I q^2$ versus $q$ to show the structure of the particles. Moreover, the data were fitted using the polydispersed core–shell sphere model (Supplementary Fig. 12f,g)[36]. From the fitting, the radius of the core, thickness of the shell and volume fraction of the micelle were derived and used to calculate the molecular weight of micelles and the mean distance between micelles using equations (2–4):

$$N = \frac{\phi_{\text{micelle}}}{v_{\text{micelle}}} \quad (2)$$

$$M_w = \frac{cN_A}{N} \quad (3)$$

$$D = N^{-1/3} × 10^7 \quad (4)$$

where $N$ is the number of micelles per unit volume. $\Phi_{\text{micelle}}$ is the volume fraction of micelles derived from fitting. $V_{\text{micelle}}$ is the volume of a single micelle, which is calculated from $4/3\pi R^3$, where $R$ is the sum of the radius of the core and the thickness of the shell. $M_w$ is the molecular weight of micelles. $c$ is the polymer concentration. $N_A$ is the Avogadro constant. $D$ is the mean distance between the micelles in nanometres. The aggregation number of micelles was calculated by dividing the molecular weight of the micelle by the molecular weight of the polymer.

## Mice

C3H/HeN and C3H/HeJ mice were maintained in a *Helicobacter*, *Pasteurella* and murine norovirus-free SPF facility at the University of Chicago.

Breeding pairs of C3H/HeJ mice were originally purchased from the Jackson Laboratory. Breeding pairs of C3H/HeN mice were transferred from the germ-free (GF) facility. All experimental mice were bred in house and weaned at 3 weeks of age on a plant-based mouse chow (Purina Lab Diet 5K67) and autoclaved sterile water. Mice were maintained on a 12 h light/dark cycle at a room temperature of 20–24 °C. GF C3H/HeN or C57BL/6 mice were bred and housed in the Gnotobiotic Research Animal Facility (GRAF) at the University of Chicago. GF mice were maintained in Trexler-style flexible film isolator housing units (Class Biologically Clean) with Ancare polycarbonate mouse cages (N10HT) and Teklad Pine Shavings (7088; sterilized by autoclave) on a 12 h light/dark cycle at a room temperature of 20–24 °C. All experiments were littermate controlled. All protocols used in this study were approved by the Institutional Animal Care and Use Committee of the University of Chicago. The FITC-dextran intestinal permeability assay in DSS-treated mice was performed by Inotiv; SPF C57BL/6 mice were obtained from Taconic and housed in the Inotiv animal facility. The T-cell-transfer colitis model was performed by Inotiv using SPF C57BL/6 donor mice and SPF Ragn12 mice. The studies were conducted in accordance with 'The Guide for the Care and Use of Laboratory Animals (8th edn)' and therefore in accordance with all Inotiv IACUC approved policies and procedures.

## Biodistribution study using IVIS

SPF C3H/HeJ mice were used for biodistribution studies. Azide-labelled pHPMA-b-pBMA or pMAA-b-pBMA polymer was reacted with IR 750-DBCO (Thermo Fisher) and purified by hexane precipitation. After formulation into micelles, the fluorescently labelled NtL-ButM or Neg-ButM was administered to mice by i.g. gavage. After 1 h, 3 h, 6 h or 24 h, mice were euthanized, the major organs were collected and whole-organ fluorescence was measured via an IVIS Spectrum in vivo imaging system (Perkin Elmer). Images were processed and analysed using Living Imaging 4.5.5 (Perkin Elmer). In another experiment, SPF C3H/HeJ mice were given 200 mg l⁻¹ vancomycin in the drinking water for 3 weeks and administered with the same fluorescently labelled NtL-ButM or Neg-ButM by i.g. gavage. Mice were euthanized at 1 h, 2 h, 4 h, 8 h, 12 h or 24 h post administration, and the whole GI tract was collected for IVIS imaging.

## Butyrate derivatization and quantification using LC–UV or LC–MS/MS

Simulated gastric fluid and simulated intestinal fluid (Thermo Fisher) were used for in vitro release analysis as described previously[59,60]. The simulated gastric fluid (Ricca Chemical) contains 0.2% (w/v) sodium chloride in 0.7% (v/v) hydrochloric acid and was added with 3.2 mg ml⁻¹ pepsin from porcine gastric mucosa (Sigma). The simulated intestinal fluid (Ricca Chemical) contains 0.68% (w/w) potassium dihydrogen phosphate, 0.06% (w/w) sodium hydroxide and pancreatin at 1% (w/w). NtL-ButM or Neg-ButM were added to simulated gastric fluid or simulated intestinal fluid at a final concentration of 2 mg ml⁻¹ at 37 °C. At pre-determined timepoints, 20 µl of the solution was transferred into 500 µl of water:acetonitrile (1:1 v/v). The sample was centrifuged using Amicon Ultra filters (Merck, 3 kDa molecular mass cut-off) at 13,000 × g for 15 min to remove polymers. The filtrate was stored at −80 °C before derivatization. For the in vivo release study in the mouse GI tract, NtL-ButM or Neg-ButM micelle solutions were i.g. administered to SPF C3H/HeJ mice at 800 mg kg⁻¹ body weight. Mice were euthanized at 1 h, 2 h, 4 h, 8 h, 12 h and 24 h after gavage. Luminal contents from the ileum, caecum or colon were collected in an EP tube. After adding 500 µl of 1× PBS, the mixture was vortexed and sonicated for 10 min, and then centrifuged at 13,000 × g for 10 min. The supernatant was transferred and filtered through 0.45 µm filter. The filtered solution was stored at −80 °C before derivatization.

A similar in vivo release experiment was performed in vancomycin-treated SPF C3H/HeJ mice. Mice (4-week-old) were treated with 200 mg l⁻¹ vancomycin in the drinking water for 3 weeks, followed by oral gavage of sodium butyrate, NtL-ButM or Neg-ButM, at the same butyrate dose of 224 mg kg⁻¹ body weight. At different timepoints, the stomach, ileum, caecum and colon were collected. To maximize butyrate detection, the whole tissue including the content was homogenized in a solution of 1:1 (v/v) water:acetonitrile, and butyrate was extracted from the supernatant after centrifugation and filtered for derivatization and LC–MS/MS measurement.

Samples were prepared and derivatized for HPLC and LC–MS/MS (Supplementary Fig. 14) as described in the literature[37]. Stock solution of 3-nitrophenylhydrazine (NPH) was prepared at 0.02 M in water:acetonitrile (1:1 v/v). 1-ethyl-3-(3-dimethylaminopropyl)carbodiimide stock solution was prepared at 0.25 M in water:acetonitrile (1:1 v/v), and 4-methylvaleric acid was added as internal standard. Samples were mixed with NPH stock and 1-ethyl-3-(3-dimethylaminopropyl)carbodiimide stock at 1:1:1 ratio by volume. The mixture was heated by heating the block at 60 °C for 30 min. Samples were filtered through 0.22 µm filters, transferred into HPLC vials and stored at 4 °C before analysis.

LC conditions: The instrument used for quantification of butyrate was Agilent 1290 UHPLC. Column: Thermo Fisher C18 4.6 × 50 mm, 1.8 µm particle size, at room temperature. Mobile phase A: water with 0.1% v/v formic acid. Mobile phase B: acetonitrile with 0.1% v/v formic acid. Injection volume: 5.0 µl. Flow rate: 0.5 ml min⁻¹. Gradient of solvent: 15% mobile phase B at 0.0 min; 100% mobile phase B at 3.5 min; 100% mobile phase B at 6.0 min; 15% mobile phase B at 6.5 min.

ESI-MS/MS method: The instrument used to detect butyrate was an Agilent 6460 Triple Quad MS-MS. Both derivatized butyrate-NPH and 4-methylvaleric-NPH were detected in negative mode. The MS conditions were optimized on pure butyrate-NPH or 4-methylvaleric-NPH at 1 mM. The fragment voltage was 135 V and collision energy was set to 18 V. Multiple reaction monitoring (MRM) of 222 → 137 was assigned to butyrate (Supplementary Fig. 14b), and MRM of 250 → 137 was assigned to 4-methylvaleric acid as internal standard. The ratio between MRM of butyrate and 4-methylvaleric acid was used to quantify the concentration of butyrate.

## RNA sequencing and data analysis

Starting at the time of weaning, GF C3H/HeN mice were i.g. administered with PBS, NtL-ButM or control polymer at 800 mg kg⁻¹ of body weight once daily for 1 week. After that time, mice were euthanized, and the ileum tissue was collected and washed thoroughly. The ileal epithelial cells (IECs) were separated from intestinal tissue by inverting ileal tissue in 0.30 mM ethylenediaminetetraacetic acid (EDTA), incubating on ice for 30 min with agitation every 5 min. RNA was extracted from the IECs using an RNA isolation kit (Thermo Fisher) according to manufacturer's instruction. RNA samples were submitted to the University of Chicago Functional Genomics Core for library preparation and sequencing on a HiSeq2500 instrument (Illumina). Single-end reads (50 bp) were generated. The quality of raw sequencing reads was assessed by FastQC (v0.11.5). Transcript abundance was quantified by Kallisto (v0.45.0) with Gencode gene annotation (release M18, GRCm38.p6), summarized to gene level by tximport (v1.12.3), trimmed mean of $M$-values (TMM) normalized and log₂ transformed. Lowly expressed genes were removed (defined as, counts per million reads mapped (CPM) < 3). Differentially expressed genes (DEGs) between groups of interest were detected using limma voom with precision weights (v3.40.6)[61]. Experimental batch and gender were included as covariates for the model fitting. Significance level and fold changes were computed using empirical Bayes moderated $t$-statistics test implemented in limma. Significant DEGs were filtered by FDR-adjusted $P < 0.05$ and fold change ≥1.5 or ≤−1.5. A more stringent $P$ value cut-off (for example, FDR-adjusted $P < 0.005$) was used for visualization of a select number of genes on expression heat maps.

## Intelectin stain and microscope imaging

GF C57BL/6 mice were i.g. administered with NtL-ButM at 0.8 mg g$^{-1}$ body weight or PBS once daily for 1 week beginning at weaning. After that time, the mice were euthanized and perfused, small intestine tissue was obtained, rolled into Swiss rolls and prepared into tissue section slides. The tissue section slides were fixed and stained with a rat anti-intelectin monoclonal antibody (R&D Systems, Clone 746420) with fluorescent detection and DAPI (ProLong antifade reagent with DAPI). The slides were imaged using a Leica fluorescence microscope. Images were processed using ImageJ and data were plotted and analysed using using GraphPad Prism.

## RT–qPCR of epithelial cells

Intestinal epithelial cells (IECs) were isolated from underlying lamina propria by incubating in 0.03 M EDTA plus 0.0015 M dithiothreitol on ice for 20 min, then in 0.03 M EDTA for 10 min at 37 °C. IECs were resuspended in Trizol and stored in −80 °C until RNA extraction. For Paneth cell product expression, RNA was isolated from homogenized tissue using a previously described guanidine thiocyanate–caesium chloride gradient method[42,62–64]. Complementary DNA synthesis was performed using the Superscript III reverse transcriptase kit (Invitrogen) as outlined by the manufacturer. The cDNA was purified using a Qiagen PCR purification kit (Qiagen), and then diluted at a concentration representing 10 ng μl$^{-1}$ of total RNA input per reaction. The RT–qPCR reactions were performed in a Roche Diagnostics Lightcycler 2.0 (Roche) using a previously described protocol[42,63]. Primers used to amplify beta actin (*Actb*), α-defensins (*Defa3*, *Defa5*, *Defa20*, *Defa21*, *Defa22*, *Defa23*, *Defa24* and *Defa26*), lysozyme (*Lyz1*), intelectin-1 (*Itln1*) and solute carrier family 10 member 2 (*Slc10a2*) were reported previously, and are summarized in Supplementary Table 1. The quantitative assays utilized to enumerate absolute transcript copy number were as reported[42,63–65].

## In vivo FITC-dextran permeability assay

SPF C57BL/6 female mice (8–10-week-old) were treated with 2.5% DSS in their drinking water for 7 d. The mice received intragastric administration of either PBS or ButM (800, 400 or 200 mg kg$^{-1}$) twice daily at approximately 10–12 h intervals, or once daily with CsA at 75 mg kg$^{-1}$ as the positive treatment control. On day 7, DSS was removed from the drinking water for the remainder of the study. On day 10, mice were fasted for 3 h and dosed with 0.1 ml FITC-dextran (4 kDa at 100 mg ml$^{-1}$). At 4 h post dose, mice were anaesthetized with isoflurane and bled to exsanguination, followed by cervical dislocation. The concentration of FITC in the serum was determined by spectrofluorometry using serially diluted FITC-dextran as standard. Serum from mice not administered FITC-dextran was used to determine the background. A similar permeability assay was also performed in the antibiotic-depletion model as previously described[25]. Littermate-controlled SPF C57BL/6 mice at 2 weeks of age were gavaged daily with a mixture of antibiotics (0.4 mg kanamycin sulfate, 0.035 mg gentamycin sulfate, 850 U colistin sulfate, 0.215 mg metronidazole and 0.045 mg vancomycin hydrochloride in 100 μl PBS) for 7 d until weaning. At weaning, mice were then treated with either PBS or ButM (800 mg kg$^{-1}$) twice daily for 7 d. After the final treatment, the mice were fasted for 3 h and dosed with 50 mg kg$^{-1}$ body weight of FITC-dextran (4 kDa at 50 mg ml$^{-1}$). Blood was collected at 1.5 h post administration via cheek bleed and the concentration of FITC in the serum was measured as described above.

## CD45RB$^{hi}$ T-cell-transfer model of colitis

CD4$^+$CD45RB$^{hi}$ and CD4$^+$CD45RB$^{lo}$CD25$^+$ cells were isolated from splenocytes of SPF C57BL/6 mice using the CD4 Cell Enrichment kit (STEMCELL), followed by cell sorting after staining with antigen presenting cell (APC) anti-CD4 (clone GK1.5, BioLegend), FITC anti-CD45RB (clone MB4B4, BioLegend) and PE anti-CD25 (clone PC61, BioLegend) antibodies[51]. Female SPF Rag12n mice (6–7-week-old) were injected i.p. with either 4 × 10$^5$ CD45RB$^{hi}$ cells (standard transfer) or with a co-transfer of CD45RB$^{hi}$ cells (minimum 4 × 10$^5$) and CD4$^+$CD45RB$^{lo}$CD25$^+$ T$_{reg}$ cells (5 × 10$^4$) in a volume of 200 ml. On the day of transfer, mice were randomly sorted into treatment groups by body weight. Body weight was measured approximately every 3 d for the duration of the experiment to examine incidence of colitis. Beginning at 2 weeks after T-cell transfer, some groups received twice-daily intragastric gavages of ButM at one of three doses (low, 80 mg kg$^{-1}$; medium, 250 mg kg$^{-1}$; high, 800 mg kg$^{-1}$) for the duration of the experiment. Beginning on day 30 after T-cell transfer, mice receiving ButM treatment additionally received a daily gavage of vancomycin (1 mg per 100 ml) for 5 d. Controls received water gavage. Mice were euthanized on day 50 after T-cell transfer, and the weight and length of colon tissue were measured at necropsy. Tissue sections from the proximal and distal colon were stored in 10% neutral buffered formalin for histological analysis.

## Colon histology

Proximal and distal colon tissues of mice from the CD45RB$^{hi}$ T-cell-transfer colitis model were paraffin embedded, cut in transverse sections, and stained with hematoxylin and eosin (H&E). Oedema was quantified as the distance from the outer muscle layer to the muscularis mucosa. Tissue hyperplasia was quantified in a scoring system determined from the size of the hyperplasia region (0 = normal, <200 mm; 0.5 = very minimal, 201–250 mm; 1 = minimal, 251–350 mm; 2 = mild, 351–450 mm; 3 = moderate, 451–550 mm; 4 = marked, 551–650 mm; 5 = severe, >650 mm). The total inflammation was evaluated in a scoring system determined by the extent of immune cell infiltration. The approximate percent of polymorphonuclear leucocytes (PMN) thought to be neutrophils are presented as PMN %. The PMN score was then multiplied by the overall inflammation score to generate a neutrophil score. Histological quantifications are either presented for individual colonic region (distal) or averaged across a distal and proximal region per mouse (whole colon).

## Peanut sensitization, ButM treatment and challenge

SPF C3H/HeN mice were treated with 0.45 mg of vancomycin in 0.1 ml volume by intragastric gavage for 7 d pre-weaning and then with 200 mg l$^{-1}$ vancomycin in the drinking water throughout the remainder of the sensitization protocol. Age- and sex-matched 3-week-old littermates were sensitized weekly by intragastric gavage with defatted, in-house-made peanut extract prepared from unsalted roasted peanuts (Hampton Farms, Severn, NC) and cholera toxin (CT) (List Biologicals) as previously described[25,45]. Sensitization began at weaning and continued for 4 weeks. Before each sensitization, the mice were fasted for 4–5 h and then given 200 ml 0.2 M sodium bicarbonate to neutralize stomach acids. After 30 min, the mice received 6 mg of peanut extract and 10 μg of cholera toxin (CT) in 150 ml PBS by intragastric gavage. After 4 weeks of sensitization, mice were permitted to rest for 1 week before a subset of mice was challenged by i.p. administration of 1 mg peanut extract in 200 ml PBS to confirm that the sensitization protocol induced a uniform allergic response. Rectal temperature was measured immediately following challenge and every 10 min for up to 90 min using an intrarectal probe, and the change in core body temperature of each mouse was recorded. The remaining mice were not challenged and were randomly assigned into experimental groups. In Fig. 5, one group of mice was treated with ButM twice daily by intragastric gavage at 800 mg of total polymer per gram of mouse body weight (800 mg kg$^{-1}$) for 2 weeks, one group received PBS and another group received sodium butyrate twice daily at 224 mg kg$^{-1}$ (equivalent amount of butyrate to 800 mg kg$^{-1}$ ButM). In the dose-dependent study (Extended Data Fig. 4), mice were treated with either PBS, ButM at 800 mg kg$^{-1}$ (full dose) or ButM at 400 mg kg$^{-1}$ (half dose) twice daily. In addition, groups of mice received NtL-ButM or Neg-ButM only at the full dose twice daily. After the treatment window, mice were challenged with i.p. administration of 1 mg peanut extract and

core body temperature was measured for 90 min. Serum was collected from mice 90 min after challenge for measurement of mMCPT-1 and histamine, and additionally at 24 h after challenge for measurement of peanut-specific IgE and IgG1. Collected blood was incubated at room temperature for 1 h and centrifuged for 7 min at 12,000 $g$ at room temperature, and sera were collected and stored at −80 °C before analysis. Serum antibodies and mMCPT-1 were measured by ELISA.

### Measurement of mMCPT-1, histamine and serum peanut-specific IgE and IgG1 antibodies using ELISA

mMCPT-1 was detected using the MCPT-1 mouse uncoated ELISA kit (Thermo Fisher) following the manufacturer's protocol. Histamine was assayed using Histamine EIA kit (Oxford Biomedical Research) according to the manufacturer's protocol. For the peanut-specific IgE ELISA, sera from individual mice were added to peanut-coated Maxisorp immunoplates (Nalge Nunc). Peanut-specific IgE antibodies were detected with goat anti-mouse IgE-unlabelled (Southern Biotechnology) and rabbit anti-goat IgG-alkaline phosphatase (Invitrogen) and developed with $p$-nitrophenyl phosphate (SeraCare Life Sciences). For peanut-specific IgG1 ELISA, sera from individual mice were added to peanut-coated Maxisorp immunoplates (Nalge Nunc). Peanut-specific IgG1 was detected using goat anti-mouse IgG1-HRP conjugated (Southern Biotechnology Associates) and TMB liquid substrate system for ELISA (Sigma Aldrich). The plates were read in an ELISA plate reader at 405 nM (IgE) or 450 nm (IgG1). Optical density values were converted to nanograms per millilitre of IgE or IgG1 by comparison with standard curves of purified IgE or IgG1 using linear regression analysis and are expressed as the mean concentration for each group of mice ± s.e.m.

### Flow cytometric analysis on immune cells isolated from spleen and mesenteric LNs

Peanut-sensitized mice were intragastrically administered twice daily with PBS, sodium butyrate or ButM at an equivalent dose of 0.224 mg g$^{-1}$ (butyrate per mouse) for 2 weeks. After the treatment, spleen, ileum and colon-draining LNs from mice were collected, and digested in DMEM supplemented with 5% FBS, 2.0 mg ml$^{-1}$ collagenase D (Sigma Aldrich) and 1.2 ml CaCl$_2$. Single-cell suspensions were prepared by mechanically disrupting the tissues through a cell strainer (70 μm, Thermo Fisher). Splenocytes ($4 \times 10^6$) or cells from LNs ($1 \times 10^6$) were plated in a 96-well plate. Cells were stained with LIVE/DEAD Fixable Aqua Dead Cell Stain kit (Thermo Fisher), followed by surface staining with antibodies in PBS with 2% FBS, and intracellular staining according to the manufacturer's protocols for eBioscience Foxp3/Transcription Factor Staining Buffer set (Invitrogen). The following anti-mouse antibodies were used: CD3 APC/Cy7 (clone 145-2C11, BD Biosciences), CD4 BV605 (clone RM4-5, Biolegend), CD25 PE/Cy7 (clone PC61, Biolegend), Foxp3 AF488 (clone MF23, BD Biosciences), CD11b BV711 (clone M1/70, BD Biosciences), CD11c PE/Cy7 (clone HL3, BD Biosciences), F4/80 APC (clone RM8, Biolegend), I-A/I-E (MHCII) APC/Cy7 (clone M5/114.15.2, Biolegend) and CD86 BV421 (clone GL-1, Biolegend). Stained cells were analysed using an LSR Fortessa flow cytometer (BD Biosciences).

### 16S rRNA targeted sequencing

Bacterial DNA was extracted using the QIAamp PowerFecal Pro DNA kit (Qiagen). The V4-V5 hypervariable region of the 16S rRNA gene from the purified DNA was amplified using universal bacterial primers: 563F (5′-nnnnnnnn-NNNNNNNNNNNN-AYTGGGYDTAAA-GNG-3′) and 926R (5′-nnnnnnnn-NNNNNNNNNNNN-CCGTCAATTYHT-TTRAGT-3′), where 'N' represents the barcodes and 'n' represents additional nucleotides added to offset primer sequencing. Illumina sequencing-compatible unique dual index adapters were ligated onto pools using the QIAseq 1-step amplicon library kit (Qiagen). Library quality control was performed using Qubit and Tapestation before sequencing on an Illumina MiSeq platform at the Duchossois Family Institute Microbiome Metagenomics Facility at the University of

Chicago. This platform generates forward and reverse reads of 250 bp which were analysed for amplicon sequence variants (ASVs) using the divisive amplicon denoising algorithm (DADA2 v1.14)[66]. Taxonomy was assigned to the resulting ASVs using the Ribosomal Database Project (RDP) database with a minimum bootstrap score of 50[67]. The ASV tables, taxonomic classification and sample metadata were compiled using the phyloseq data structure[68]. Subsequent 16S rRNA relative abundance analyses and visualizations were performed using R version 4.1.1 (R Development Core Team).

### Microbiome analysis

To identify changes in the microbiome across conditions, a linear discriminant analysis effect size (LEfSe) analysis was performed in R using the microbiomeMarker package and the run_lefse function[69,70]. Features, specifically taxa, can be associated with or without a given condition (for example, ButM post treatment versus PBS post treatment) and an effect size can be ascribed to that difference in taxa at a selected taxonomic level (LDA score). For the LEfSe analysis, genera were compared as the main group, a significance level of 0.05 was chosen for both the Kruskall-Wallis and Wilcoxon tests and a linear discriminant analysis cut-off of 1.0 was implemented. The abundance of *Clostridium* cluster XIVa in post-treatment samples was also determined by quantitative PCR (qPCR) using the same DNA analysed by 16S rRNA targeted sequencing. Commonly used primers 8F[71] and 338R[72] were used to quantify total copies of the 16S rRNA gene for normalization purposes. Primers specific for *Clostridium* cluster XIVa[73] were validated by PCR and qPCR. Primer sequences are listed in Supplementary Table 1. qPCR was performed using PowerUp SYBR Green master mix (Applied Biosystems) according to the manufacturer's instructions. The abundance of *Clostridium* cluster XIVa was calculated by $2^{-CT}$ (CT, cycle threshold), multiplied by a constant to bring all values above 1 ($1 \times 10^{16}$) and expressed as a ratio to total copies 16S rRNA per gram of raw faecal content.

### Statistical analysis

Statistical analysis and plotting of data were performed using Graphpad Prism 9.0, as indicated in the figure legends. One-way analysis of variance (ANOVA) with Dunnett's or Tukey's post-test for multiple comparisons were used in Figs. 4b, 5j–m, 7d–f and 8b–e. Two-sided Student's $t$-test was used in Figs. 4d, 5e–h and 6c,d. In Fig. 5d,i, the area under curve (AUC) values of temperature changes were compared using two-sided Student's $t$-test (i) or one-way ANOVA with Tukey's post-test (d). Data represent mean ± s.e.m.; $n$ is stated in the figure legend.

### Reporting summary

Further information on research design is available in the Nature Portfolio Reporting Summary linked to this article.

## Data availability

The main data supporting the results in this study are available within the paper and its Supplementary Information. Additional processed data are available from the corresponding authors on request. The 16S rRNA and RNAseq raw FastQ data files are available from the National Center for Biotechnology (NCBI) Sequence Read Archive, via the accession numbers PRJNA863725 and PRJNA885957, respectively. Source data for the figures are provided with this paper.

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

## Acknowledgements

We thank A. A. Alshaikh, Z. Khosravi, H.-N. Shim, N. Talasani, H. Wu, M. Bauer and S. Gomes for technical assistance; B. Theriault from GRAF and P. Faber from the Functional Genomics Facility. Parts of this work were carried out at the Cytometry and Antibody Technology Core Facility (Cancer Center Support Grant P30CA014599), the Duchossois Family Institute Microbiome Metagenomics Facility, the Soft Matter Characterization Facility, the Mass Spectrometry Facility (NSF instrumentation grant CHE-1048528), the Nuclear Magnetic Resonance Facility, the Advanced Electron Microscopy Facility (RRID:SCR_019198), the Functional Genomics Core at the University of Chicago, and the SLAC National Accelerator Laboratory. Some of the bioinformatics analysis was performed on the high-performance computing (HPC) clusters at the University of Chicago Center for Research Informatics, and we thank M. Jarsulic for technical assistance on the HPC clusters. This work was supported by a Sponsored Research Agreement from ClostraBio, Inc. to C.R.N., J.A.H and the University of Chicago. The Chicago Immunoengineering Innovation Center of the University of Chicago provided additional support. D.S.W. was supported by a fellowship from the Whitaker Foundation.

## Author contributions

C.R.N. and J.A.H. oversaw all research. R.W., S.C., M.E.H.B., L.A.H., C.R.N. and J.A.H. designed the research strategy. R.W., D.S.W. and J.A.H. conceptualized materials. R.W. and S.C. synthesized materials and fabricated micelles. R.W., S.C., M.E.H.B., L.A.H., Y.S., S.M.C.H., A.T., E.C., M.S. and E.B.N. performed experiments. R.W., S.C., M.E.H.B, C.L.B. and L.A.H. analysed experiments. R.B. performed data analysis on the RNA sequencing experiment. N.P.D. and E.C. performed 16S rRNA targeted microbiome analysis. R.W., S.C., M.E.H.B., L.A.H., C.R.N. and J.A.H. wrote the manuscript. All authors approved the manuscript.

## Competing interests

C.R.N. and J.A.H. are founders and shareholders of ClostraBio, Inc., which is developing the technology described in this study. R.W., S.C., M.E.H.B., D.S.W., C.R.N. and J.A.H. are inventors on patents filed by the University of Chicago describing the micelles reported in this study. The other authors declare no competing interests.

## Additional information

**Extended data** is available for this paper at https://doi.org/10.1038/s41551-022-00972-5.

**Correspondence and requests for materials** should be addressed to Jeffrey A. Hubbell or Cathryn R. Nagler.

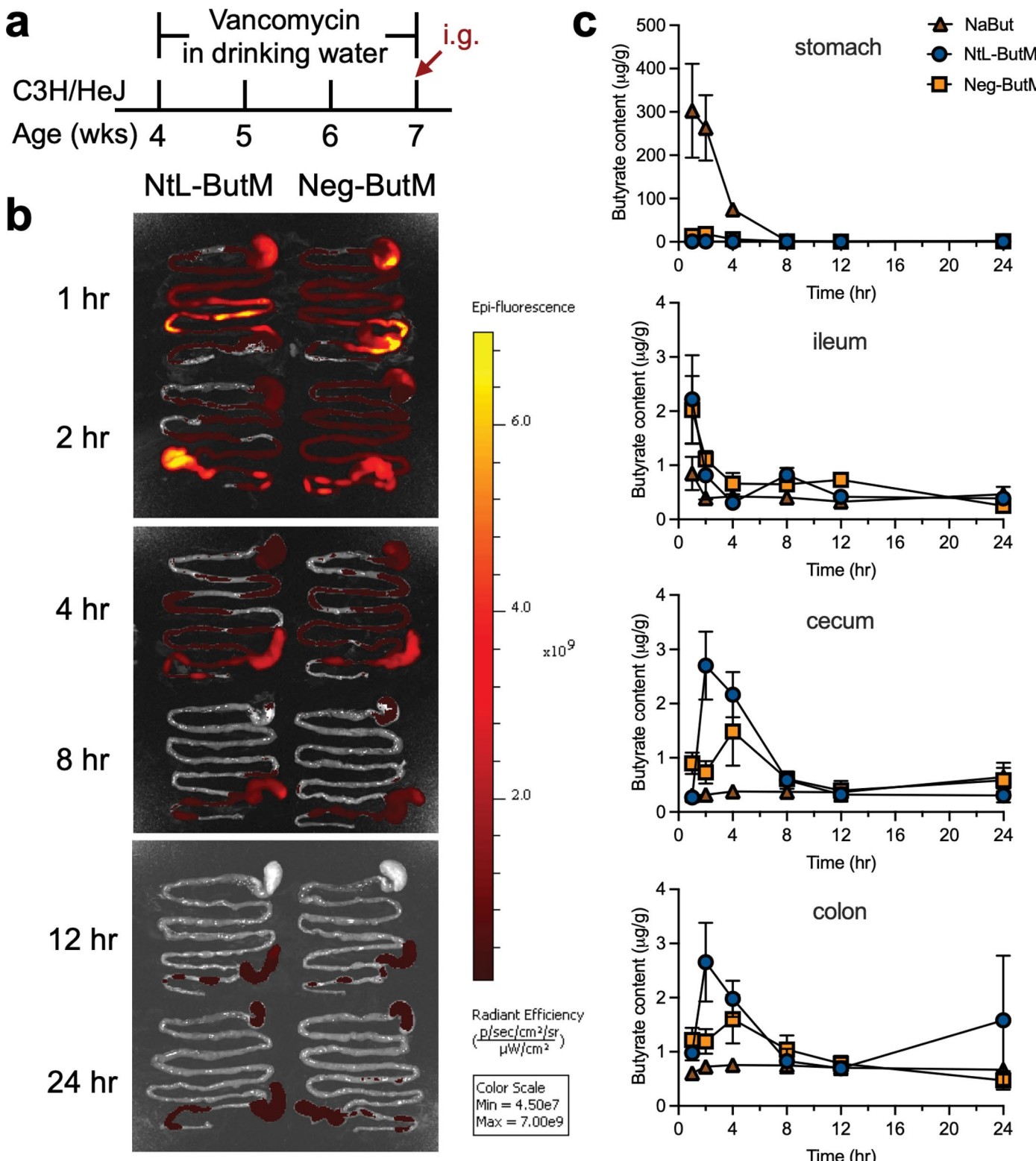

**Extended Data Fig. 1 | The biodistribution and *in vivo* butyrate release of NtL-ButM or Neg-ButM in the GI tract of vancomycin-treated mice. a,** Butyrate producing bacteria were depleted by treating 4 week old SPF C3H/HeJ mice with 200 mg l⁻¹ vancomycin in the drinking water for three weeks prior to intragastric (i.g.) administration of each of the butyrate formulations. **b,** Mice were administered IR750 labelled NtL-ButM or Neg-ButM intragastrically and euthanized at the indicated time points to measure the fluorescent signals in the GI tract by IVIS. **c,** The amount of butyrate released in the stomach, ileum, cecum, or colon after a signal intragastric administration of sodium butyrate, NtL-ButM, or Neg-ButM (at an equivalent butyrate dose of 224 mg kg⁻¹) to vancomycin-treated mice. At the indicated time points, the stomach, ileum, cecum, and colon were harvested, and the whole tissues were homogenized, followed by butyrate extraction and quantification by LC-MS/MS. $n = 5$ mice per group per time point. Data represent mean ± s.e.m.

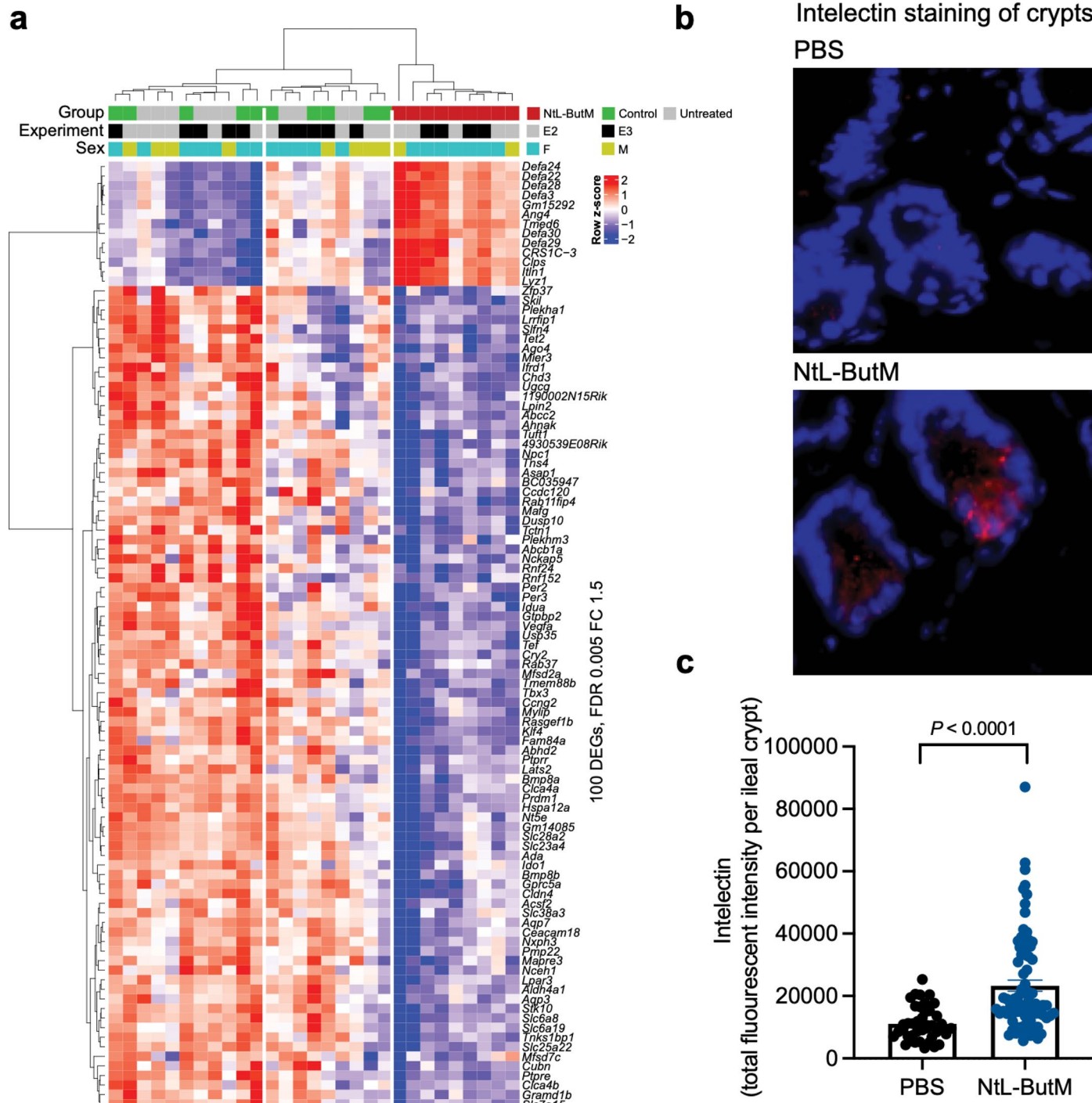

**Extended Data Fig. 2 | NtL-ButM induced an ileal gene expression signature that is almost entirely anti-microbial peptides (AMPs). a**, One week of daily dosing of 800 mg kg$^{-1}$ NtL-ButM to germ-free (GF) C3H/HeN mice induced a unique gene expression signature in the ileum compared to untreated and inactive polymer controls as measured by RNA sequencing of isolated intestinal epithelial cells. Top 100 significant differentially expressed genes (DEGs) at False Discovery Rate (FDR)-adjusted $P < 0.005$ and fold change (FC) ≥ 1.5 or ≤ −1.5 are shown. Annotation bars of the three groups, experiment batches (E2 and E3), and gender (female, male) are shown above the heatmap. **b**, Fluorescent imaging of intelectin protein in small intestine sections from control or treated mice. Blue (DAPI), red (intelectin). **c**, Intelectin protein is quantified by total fluorescence signal per crypt of small intestine. $n = 3$ PBS-treated and 4 NtL-ButM treated mice, with >15 crypts quantified per mouse. Data represent mean ± s.e.m. limma voom with precision weights was used in **a**. Two-sided Student's $t$-test was used in **c**.

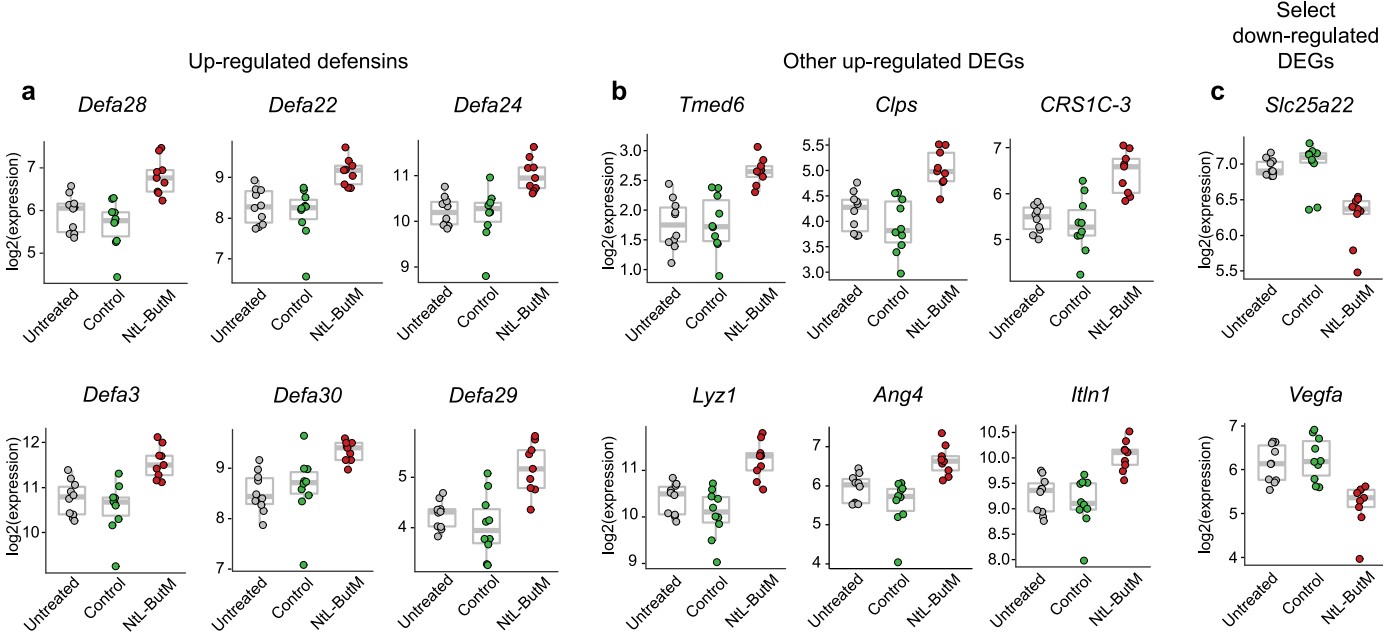

FDR-adjusted *P*<0.005

**Extended Data Fig. 3 | Differentially expressed genes (DEGs) in the ileum of GF mice that were treated daily with 800 mg kg⁻¹ NtL-ButM for one week, compared to untreated and inactive polymer controls as measured by RNA sequencing of isolated ileal epithelial cells (see Extended Data Fig. 2).** The unit of the value is TMM-normalized and log2-transformed read counts.

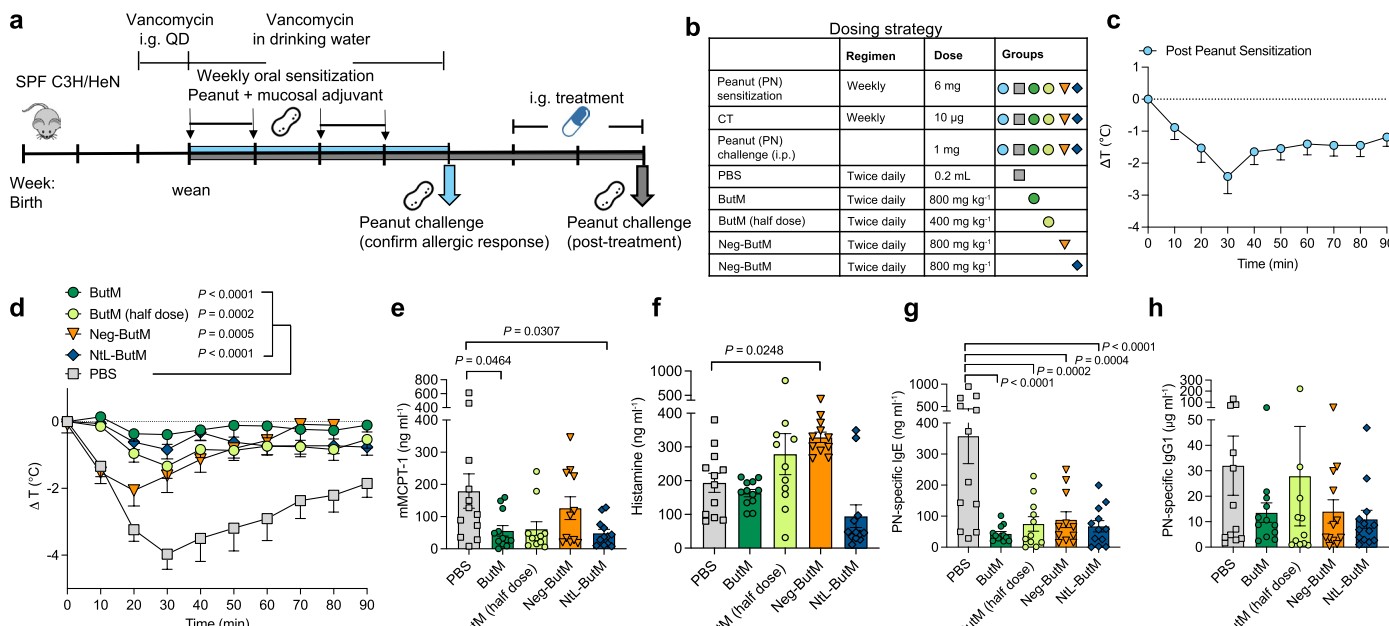

**Extended Data Fig. 4 | Butyrate micelle treatment reduced the anaphylactic response to peanut challenge in a dose-dependent manner. a, b**, Experimental schema, and the dosing strategy. All the mice were sensitized weekly by i.g. gavage of 6 mg of PN plus 10 µg of the mucosal adjuvant cholera toxin (CT). **c**, A uniform allergic response was confirmed by challenging one group of mice ($n=7$) with 1 mg of PN i.p. and measuring reduction in core body temperature as an indication of anaphylaxis. QD: once a day. **d-h**, The rest of mice were randomized into five treatment groups and received either PBS ($n=12$), ButM ($n=12$), ButM (half dose) ($n=11$), Neg-ButM only ($n=11$), or NtL-ButM only ($n=12$). **d**, Change in core body temperature following challenge with peanut extract. The area under curve (AUC) was compared among groups. **e-h**, Serum mMCPT-1 (**e**), histamine (**f**), peanut-specific IgE (**g**), and peanut-specific IgG1 (**h**) from mice in **d**. Data represent mean ± s.e.m. Data analyzed using one-way ANOVA with Dunnett's post-test.

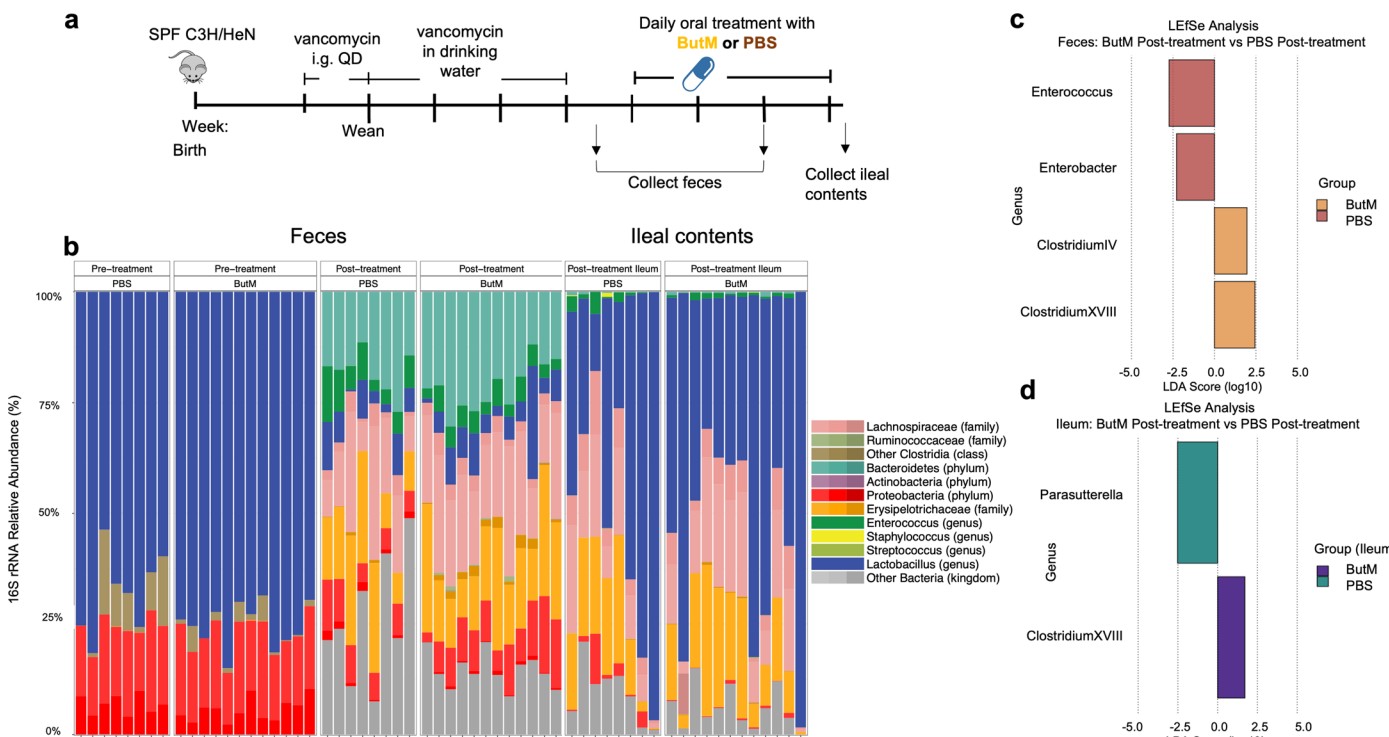

**Extended Data Fig. 5 | Treatment with ButM in unsensitized mice increased relative abundance of Clostridium clusters in the ileum and feces. a**, Experimental schema. Mice were treated for five weeks with vancomycin, either by i.g. gavage pre-weaning or in the drinking water post-weaning. Upon cessation of treatment, mice were treated with PBS (n = 8) or 800 mg kg$^{-1}$ ButM (n = 12) twice daily. Fecal samples were collected just after ending antibiotic administration (pre-treatment) or after two weeks of treatment with PBS or ButM (post-treatment) (as in Fig. 4), and ileal contents were collected at euthanasia. **b**, Relative abundance of bacterial taxa in feces and ileal contents. **c**, Differentially abundant taxa in feces between mice treated with PBS or ButM post-treatment analyzed by LEfSe. **d**, Differentially abundant taxa in ileal contents between mice treated with PBS or ButM post-treatment analyzed by LEfSe. QD: once a day.

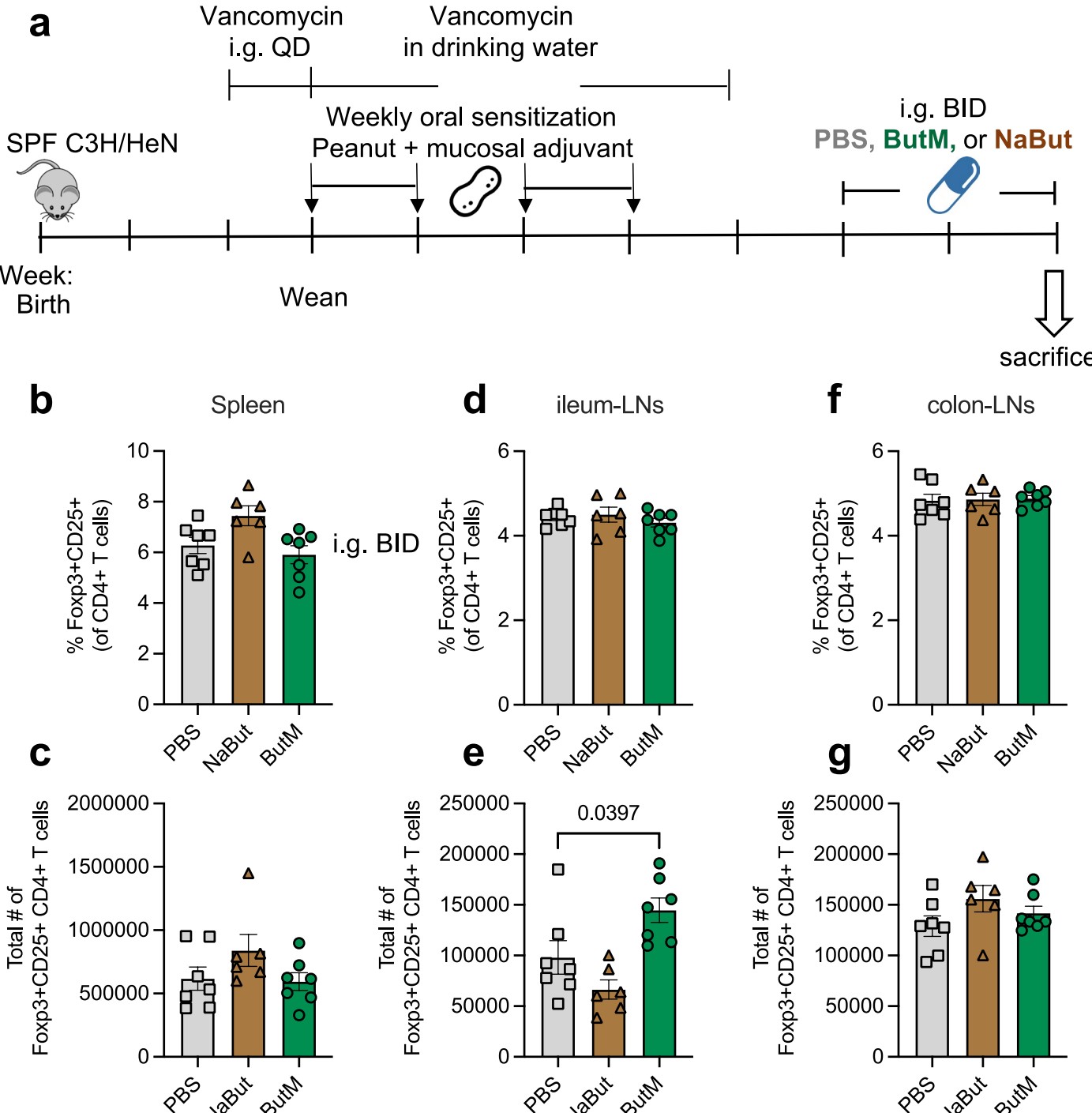

**Extended Data Fig. 6 | Butyrate micelle treatment did not significantly affect regulatory T cell (Treg) populations in spleen or mesenteric LNs.**
**a**, Experimental schema. All of the vancomycin-treated mice were sensitized weekly by i.g. gavage of 6 mg of PN plus 10 μg of the mucosal adjuvant cholera toxin. The mice were then intragastrically treated with either PBS (*n* = 7), sodium butyrate (NaBut, *n* = 6), or ButM (*n* = 7) twice daily for two weeks. The mice were euthanized after treatment, and their spleen (**b, c**), ileum draining LNs (**d, e**), and colon draining LNs (**f, g**) were harvested and processed for flow cytometric analysis. Gating strategy is shown in Supplementary Fig. 19. Data represent mean ± s.e.m. Data analyzed using one-way ANOVA with Dunnett's post-test.

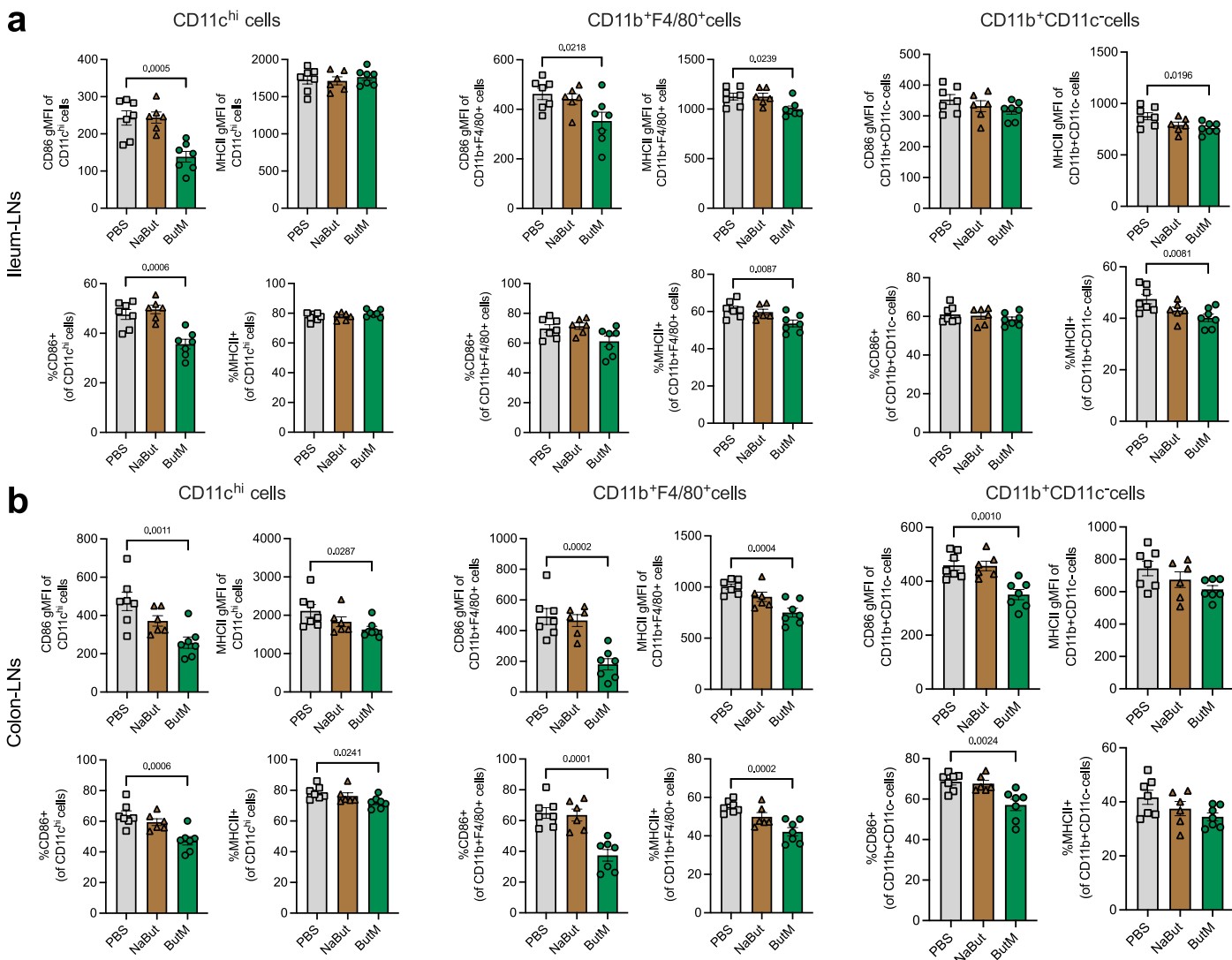

**Extended Data Fig. 7 | Butyrate micelle treatment downregulated MHCII and the costimulatory molecule CD86 on the CD11c^hi, CD11b^+F4/80^+ and CD11b^+CD11c^− cells in the ileum (a) and colon (b) draining LNs.** The experimental schema is shown in Extended Data Fig. 6a. Gating strategy is shown in Supplementary Fig. 20. Data represent mean ± s.e.m. Data analyzed using one-way ANOVA with Dunnett's post-test.

| | |
|---|---|

# Reporting Summary

## Statistics

For all statistical analyses, confirm that the following items are present in the figure legend, table legend, main text, or Methods section.

| n/a | Confirmed | |
|---|---|---|
| ☐ | ☒ | The exact sample size (*n*) for each experimental group/condition, given as a discrete number and unit of measurement |
| ☐ | ☒ | A statement on whether measurements were taken from distinct samples or whether the same sample was measured repeatedly |
| ☐ | ☒ | The statistical test(s) used AND whether they are one- or two-sided *Only common tests should be described solely by name; describe more complex techniques in the Methods section.* |
| ☐ | ☒ | A description of all covariates tested |
| ☒ | ☐ | A description of any assumptions or corrections, such as tests of normality and adjustment for multiple comparisons |
| ☐ | ☒ | A full description of the statistical parameters including central tendency (e.g. means) or other basic estimates (e.g. regression coefficient) AND variation (e.g. standard deviation) or associated estimates of uncertainty (e.g. confidence intervals) |
| ☐ | ☒ | For null hypothesis testing, the test statistic (e.g. *F*, *t*, *r*) with confidence intervals, effect sizes, degrees of freedom and *P* value noted *Give P values as exact values whenever suitable.* |
| ☒ | ☐ | For Bayesian analysis, information on the choice of priors and Markov chain Monte Carlo settings |
| ☒ | ☐ | For hierarchical and complex designs, identification of the appropriate level for tests and full reporting of outcomes |
| ☒ | ☐ | Estimates of effect sizes (e.g. Cohen's *d*, Pearson's *r*), indicating how they were calculated |

*Our web collection on statistics for biologists contains articles on many of the points above.*

## Software and code

Policy information about availability of computer code

| Data collection | Bruker TopSpin (3.x) was used to collect NMR data. Agilent ChemStation was used to collect LCMS and Triple Quad MS-MS data. Wyatt Dynamics was used to collect DLS and zeta-potential data. Tosoh EcoSEC software or Empower Pro analysis software and Wyatt analysis software (ASTRA) was used to collect GPC data. Chromeleon CDS 7 was used to collect LC-UV data. FastQC (v0.11.5) was used to assess raw RNA sequencing data. LSR Fortessa flow cytometer (BD Biosciences) was used to collect flow cytometry data. Data collection for 16S rRNA targeted sequencing is detailed in the Methods session. |
|---|---|
| Data analysis | MNova (14.2.0) was used to analyze NMR data. Igor Pro 8 was used to analyze SAXS data. ImageJ (1.8.0) was used to analyze CryoEM data. Living Imaging 4.5.5 (Perkin Elmer) was used to analyze IVIS data. Fiji was used to analyze fluorescent images of intelectin staining. FlowJo 10.8.0 was used to analyze flow cytometry data. Data analysis for the RNA sequencing, 16S rRNA targeted sequencing, and microbiome analysis are detailed in the Methods session. Prism software (GraphPad v9) was used to conduct all statistical tests. Microsoft Excel was used to carry out simple operations. |

For manuscripts utilizing custom algorithms or software that are central to the research but not yet described in published literature, software must be made available to editors and reviewers. We strongly encourage code deposition in a community repository (e.g. GitHub). See the Nature Portfolio guidelines for submitting code & software for further information.

## Data

Policy information about availability of data

All manuscripts must include a data availability statement. This statement should provide the following information, where applicable:
- Accession codes, unique identifiers, or web links for publicly available datasets
- A description of any restrictions on data availability
- For clinical datasets or third party data, please ensure that the statement adheres to our policy

The main data supporting the results in this study are available within the paper and its Supplementary Information. Additional processed data are available from the corresponding authors upon request. The 16S rRNA and RNAseq raw FastQ data files have been deposited into the National Center for Biotechnology (NCBI) Sequence Read Archive are available under accession numbers PRJNA863725 and PRJNA885957, respectively.

# Field-specific reporting

Please select the one below that is the best fit for your research. If you are not sure, read the appropriate sections before making your selection.

☒ Life sciences          ☐ Behavioural & social sciences          ☐ Ecological, evolutionary & environmental sciences

For a reference copy of the document with all sections, see nature.com/documents/nr-reporting-summary-flat.pdf

# Life sciences study design

All studies must disclose on these points even when the disclosure is negative.

| Sample size | Sample size was determined using the results obtained from previous and preliminary studies. Each experiment contained 9-16 mice per group. See figure legends for details on n for each display figure. |
|---|---|
| Data exclusions | No data was excluded from the analysis. |
| Replication | At least 5, and in most cases 10, independent biological replicates were examined for each datapoint analyzed. |
| Randomization | Littermate controlled mice were bred in house and randomly allocated into each treatment. For experiments performed by Inotiv, mice were purchased from a vendor and randomly allocated into treatment groups. |
| Blinding | The person assessing the temperature for all challenges was a second person different from the person taking care of the mice and treatments and blinded to the treatment group. All mice were identified by a unique 5 digit ear-tag which allowed for blinding during ELISA assays and other analysis as results were first analyzed and tabulated based on the 5 digit identifiers before being matched to experimental groups. The person performing fluorescent imaging and image analysis was blinded to treatment group. |

# Reporting for specific materials, systems and methods

We require information from authors about some types of materials, experimental systems and methods used in many studies. Here, indicate whether each material, system or method listed is relevant to your study. If you are not sure if a list item applies to your research, read the appropriate section before selecting a response.

### Materials & experimental systems

| n/a | Involved in the study |
|---|---|
| ☐ | ☒ Antibodies |
| ☒ | ☐ Eukaryotic cell lines |
| ☒ | ☐ Palaeontology and archaeology |
| ☐ | ☒ Animals and other organisms |
| ☒ | ☐ Human research participants |
| ☒ | ☐ Clinical data |
| ☒ | ☐ Dual use research of concern |

### Methods

| n/a | Involved in the study |
|---|---|
| ☒ | ☐ ChIP-seq |
| ☐ | ☒ Flow cytometry |
| ☒ | ☐ MRI-based neuroimaging |

## Antibodies

| Antibodies used | Goat anti-mouse IgE UNLB Southern Biotech Cat# 1110-01 (1:2,000)<br>Rabbit anti-goat IgG (H+L) secondary antibody AP conjugate, Invitrogen, Cat# 31300, (1:5,000)<br>Goat anti-mouse IgG1-HRP conjugated, Southern Biotechnology Associates, Cat#1071-05<br>Mouse MCPT-1 Uncoated ELISA Kit: Thermofisher Scientific, Cat# 88-7503-88<br>Histamine EIA Kit, Oxford Biomedical Research, Cat#EA31 |
|---|---|

Rat anti-mouse intelectin-1, R&D Systems, Cat# MAB8074
APC anti-mouse CD4, clone GK1.5, BioLegend, Cat#100412
FITC anti-CD45RB, clone GK1.5, BioLegend, Cat#100412
PE anti-CD25, clone GK1.5, BioLegend, Cat#100412
APC-Cy™7 Hamster Anti-Mouse CD3e, clone 145-2C11, BD Biosciences, Cat#557596
Brilliant Violet 605™ anti-mouse CD4, clone RM4-5, Biolegend, Cat#100547
PE/Cyanine7 anti-mouse CD25, clone PC61, Biolegend,Cat#102016
Alexa Fluor® 488 Rat anti-Mouse Foxp3, clone MF23, BD Biosciences, Cat#560407
BV711 Rat Anti-CD11b, clone M1/70, BD Biosciences, Cat#563168
PE-Cy™7 Hamster Anti-Mouse CD11c, clone HL3, BD Biosciences, Cat#561022
APC anti-mouse F4/80, clone RM8, Biolegend, Cat#123116
APC/Cyanine7 anti-mouse I-A/I-E, clone M5/114.15.2, Biolegend, Cat#107628
Brilliant Violet 421™ anti-mouse CD86, clone GL-1, Biolegend, Cat#105032

Validation

The antibodies and kits for ELISA and Immunohistochemistry were validated according to the manufacturers' websites using suggested working dilutions as described in their Technical Bulletins.

## Animals and other organisms

Policy information about studies involving animals; ARRIVE guidelines recommended for reporting animal research

Laboratory animals

Male and female C3H/HeN, C3H/HeJ, C57BL/6, and Ragn12 mice were used for the studies in this paper. See methods section for details.

Wild animals

The study did not involve wild animals.

Field-collected samples

The study did not involve samples collected from the field.

Ethics oversight

All protocols used in this study were approved by the Institutional Animal Care and Use Committee of the University of Chicago and Inotiv.

Note that full information on the approval of the study protocol must also be provided in the manuscript.

## Flow Cytometry

### Plots

Confirm that:

☒ The axis labels state the marker and fluorochrome used (e.g. CD4-FITC).

☒ The axis scales are clearly visible. Include numbers along axes only for bottom left plot of group (a 'group' is an analysis of identical markers).

☒ All plots are contour plots with outliers or pseudocolor plots.

☒ A numerical value for number of cells or percentage (with statistics) is provided.

### Methodology

Sample preparation

Spleen, ileum and colon draining LNs from mice were collected, and digested in DMEM supplemented with 5% FBS, 2.0 mg/mL collagenase D (Sigma Aldrich) and 1.2 mL CaCl2. Single-cell suspensions were prepared by mechanically disrupting the tissues through a cell strainer (70 μm, Thermo Fisher). Splenocytes (4 x 10^6) or cells from LNs (1 x 10^6) were plated in a 96 well-plate. Cells were stained with LIVE/DEAD™ Fixable Aqua Dead Cell Stain Kit (Thermo Fisher), followed by surface staining with antibodies in PBS with 2% FBS, and intracellular staining according to the manufacturer's protocols from eBioscience™ Foxp3/Transcription Factor Staining Buffer Set (Invitrogen).

Instrument

LSR Fortessa flow cytometer (BD Biosciences)

Software

FlowJo 10.8.0

Cell population abundance

Representative cell population abundance are shown in supplementary Fig. S19 and Fig. S20.

Gating strategy

Representative gating strategies are shown in supplementary Fig. S19 and Fig. S20.

☒ Tick this box to confirm that a figure exemplifying the gating strategy is provided in the Supplementary Information.

