## [Peer Review File · Nature Biomedical Engineering]

Treatment of peanut allergy and colitis in mice via the intestinal release of butyrate from polymeric micelles

Corresponding author: Cathryn Nagler

Editorial note

This document includes relevant written communications between the manuscript's corresponding author and the editor and reviewers of the manuscript during peer review. It includes decision letters relaying any editorial points and peer-review reports, and the authors' replies to these (under 'Rebuttal' headings). The editorial decisions are signed by the manuscript's handling editor, yet the editorial team and ultimately the journal's Chief Editor share responsibility for all decisions.

Any relevant documents attached to the decision letters are referred to as **Appendix #**, and can be found appended to this document. Any information deemed confidential has been redacted or removed. Earlier versions of the manuscript are not published, yet the originally submitted version may be available as a preprint. Because of editorial edits and changes during peer review, the published title of the paper and the title mentioned in below correspondence may differ.

Correspondence

Sat 12 Mar 2022

Decision on Article nBME-22-0241

Dear Dr Nagler,

Thank you again for submitting to *Nature Biomedical Engineering* your manuscript, "Microbial metabolite butyrate-prodrug polymeric micelles promote gut health and treat food allergies". The manuscript has been seen by three experts, whose reports you will find at the end of this message.

You will see that the reviewers appreciate the work. However, they express concerns about the degree of support for the claims, and provide useful suggestions for improvement. We hope that with significant further work you can address the criticisms and convince the reviewers of the merits of the study. In particular, as per the reviewer points, we would expect that a revised version of the manuscript provides extended characterization data, additional mechanistic insight, and a performance comparison with butyrylated starches.

When you are ready to resubmit your manuscript, please upload the revised files, a point-by-point rebuttal to the comments from all reviewers, the reporting summary, and a cover letter that explains the main improvements included in the revision and responds to any points highlighted in this decision.

Please follow the following recommendations:

* Clearly highlight any amendments to the text and figures to help the reviewers and editors find and understand the changes (yet keep in mind that excessive marking can hinder readability).

* If you and your co-authors disagree with a criticism, provide the arguments to the reviewer (optionally, indicate the relevant points in the cover letter).* If a criticism or suggestion is not addressed, please indicate so in the rebuttal to the reviewer comments and explain the reason(s).

* Consider including responses to any criticisms raised by more than one reviewer at the beginning of the rebuttal, in a section addressed to all reviewers.

* The rebuttal should include the reviewer comments in point-by-point format (please note that we provide all reviewers will the reports as they appear at the end of this message).

* Provide the rebuttal to the reviewer comments and the cover letter as separate files.

We hope that you will be able to resubmit the manuscript within 20 weeks from the receipt of this message. If this is the case, you will be protected against potential scooping. Otherwise, we will be happy to consider a revised manuscript as long as the significance of the work is not compromised by work published elsewhere or accepted for publication at *Nature Biomedical Engineering*.

We hope that you will find the referee reports helpful when revising the work. Please do not hesitate to contact me should you have any questions.

Best wishes,

Pep

Pep Pàmies
Chief Editor, Nature Biomedical Engineering

Reviewer #1 (Report for the authors (Required)):

In this manuscript, the authors developed two polymeric micellar systems, which could release butyrate from their polymeric core in the ileum or the cecum, respectively. Oral delivery of butyrate to the ileum by these nanomicelles up-regulated genes expressing antimicrobial peptides in the ileal epithelium in germ-free mice. The use of the combination of these two types of nanomicelles restored a barrier-protective response in mice treated with either dextran sodium sulfate or antibiotics. Treatment with the micelles also protected peanut-allergic mice from an anaphylactic reaction to peanut challenge and rescued their dysbiosis by increasing the abundance of Clostridium Cluster XIVa. This work is interesting and offers insights for developing new therapies for food allergies.

Comments for improvement:

1. The advantages of these butyrate-loaded nanomicelles over other formulations such as commonly used butyrylated starches mentioned in the introduction should be demonstrated as this is critical for claiming the advance and significance of this work.
2. As introduced, butyrate is the preferred energy substrate for colonic epithelial cells and the oral delivery of butyrate to the colon has been a challenge, while as the authors evidenced the current nanomicelles did not increase the level of butyrate in the colon following intragastric administration. Are these micelles still lack of capability to tackle this difficulty?
3. It was described that the core structure made of pBMA was more condensed with higher contrast, while this could not be observed from cryo-TEM images. Furthermore, the sizes of these two types of micelles calculated from cryo-TEM images are quite different. However, as stated, the DLS results showed similar sizes of 45 and 40 nm respectively, which was confusing. Please clarify.
4. The detailed formulation of simulated gastric fluid and simulated intestinal fluid should be added as this is key for evaluating the stability of the micelles as well as their release behaviors. For example, whether these fluids contained pepsin or trypsin.

5. The release studies showed that most of their butyrate was released within minutes in simulated intestinal fluid. It would be expected that the release may even faster after gastric empty in vivo. The burst release would inevitably promote absorption and metabolism in the small intestine. This may be the main reason why increased level of butyrate in the colon was not achieved.
6. IVIS imaging indicated that accumulation in the stomach was observed for Neg-ButM, which has a relative high CMC value. It was speculated that the micelles would suffer instability as the presence of continuous flush by gastric fluid. Is there any data showing the in vivo butyrate release in stomach and the percent of butyrate released there?
7. The reason for the selection of a 1:1 combination of NtL-ButM and Neg-ButM for efficacy study should be explained. Moreover, to clarify the individual effects of each type of micelle, both NtL-ButM and Neg-ButM should be used as controls in the assessment study in peanut allergic mice.
8. As the authors observed that reducing the dose of ButM by half was not as effective as the full dose in protecting mice from an anaphylactic response. Whether increasing the dose could further improve the treatment efficacy. This is important to disclose the potency of these nanomicelles.
9. At least, a control group of free butyrate in the same dose should be set for all in vivo experiments, to compare the treatment efficacy.
10. It was explained that butyrate sensing by peroxisome proliferator-activated receptor shunts colonocyte metabolism toward β -oxidation, creating a local hypoxic niche for these oxygen sensitive anaerobes. While, most of the gut microbiota including both beneficial and pathogenic bacteria are anaerobic. The mechanism of ButM-mediated increased abundance of Clostridium Cluster XIVA should be further clarified.
11. Assessment of ameliorated anaphylaxis was incomprehensive. The authors detected the levels of mouse mast cell protease-1 (mMCPT-1) and specific IgE in anaphylactic responses. However, the pathogenesis of the allergic response is that binding of allergens to specific IgE trigger cross-linking of Fc ϵ RI on mast cells, which induce cellular degranulation and release of histamine. Crucial factors including degree of cellular degranulation and level of histamine should be detected. Moreover, the authors should show the gating strategy to obtain the final fraction of mast cells.
12. Measurement of the intestinal permeability with lactulose/mannitol test is recommended as the uncontrolled flux of food antigen across the small intestine is associated with the development of food allergy. However, FITC-dextran in 4 kDa mainly refers to the permeability of entire gut, particularly the large intestine.

Reviewer #2 (Report for the authors (Required)):

The authors developed two polymeric micelle systems that release butyrate from their polymeric core in the ileum or the cecum. They found that these butyrate-containing micelles could protect the intestinal barrier. There could also protect peanut-allergic mice from an anaphylactic reaction to peanut challenge. This butyrate-containing micelles can overcome the problem of butyrate acid application at present. However, there are some questions :

1. The FITC-dextran experiment could only evaluate the overall characteristics of the intestinal barrier. As the butyrate micelles could mainly release butyrate in the ileum and cecum, it is suggested that the key mucosal barrier proteins and mucus layer should be detected.
2. NtL-ButM dramatically increased the butyrate concentration in the ileum for up to 2 hr after gavage (Fig. 2c). Partial of Neg-ButM could traveled to the ileum 1 h after gavage (Fig. S15a). It could not enhance the butyrate in the ileum. Why?
3. There is no control group in Fig. 2c, 2d and 2e. It is difficult to evaluate the effect of NtL-ButM or Neg-ButM on the butyrate concentration in intestine.

4. In Fig. 5, the number of mice in PBS group and ButM group were 31 and 42, respectively. Will the difference in the number of mice between the two groups affect the results?
5. In Fig. 6, Lactobacillus genus dominated the microbiota of mice in both group before treatment, while Bacteroides increased significantly after treatment. Besides, In Fig. S19, Clostridium Cluster XIVa increased in both group after treatment by PBS or ButM. Is it possible that the difference caused by age? And there are other genus significantly, whether micelles protects peanut-allergic mice by increasing the abundance of Clostridium Cluster XIVa? It is not sure.
6. The level of phylum, family and genus were shown in Fig. 6A at the same time. It is difficult to understand.
7. In Fig. S18d, after low dose PN treatment, the change of body temperature in the PN group (blue square) was greater than that in the PBS group for a long time (Fig. 5d). Can you explain?
8. The butyrate-containing micelles could up-regulated genes expressing antimicrobial peptides in the ileal epithelium in germ-free mice. Whether it could affect the expression of genes encoding antimicrobial peptides in SPF mice?

Overall, though the butyrate-containing micelles could release the butyrate in small intestine and protect peanut-allergic mice from an anaphylactic reaction to peanut challenge, its role in restoring microbial and mucosal homeostasis still needs to be shown.

Reviewer #3 (Report for the authors (Required)):

Here, the authors developed two Butyrate-releasing polymeric micelles and examined their effects on intestinal barrier functions and allergic responses. The authors showed that NtL-ButM increased AMPs and ButM modulated gut microbiome with increased Clostridium Cluster XIVa in a dysbiosis model. While this presents an interesting concept, there are many mechanistic aspects that need to be addressed:

1. Neg-ButM was prepared via base titration. The stability of the anionic micelles in low pH buffer can be shown (Fig. 1e). Do the anionic micelles resist the low pH in the stomach without further segregation? Does it affect the enzyme-mediated butyrate release?
2. The release of butyrate was proposed to be dependent on the esterase in the GI tract (L117-118). Does the amide group present in the pBMA chain resist enzymatic and microbial degradation and consequently release of 2-hydroxypropylamine?
3. Butyrate was released within minutes upon incubation with SIF (Fig. 2b) but NtL-ButM did not reach cecum and colon in vivo (Fig. 2c,e). What are the proposed mechanisms of Neg-ButM specifically targeting cecum and NtL-ButM targeting ileum (Fig. 2c,d)? The IVIS data in Figure S15 showed that the fluorescence intensity of Neg-ButM is already very low even at 6 h post oral administration. However, the butyrate release in cecum would last up to 12 h (Figure 2D), please explain this.
4. Does the micelle disproportionately promote barrier integrity along the long intestinal tract? Can the authors examine integrity of the intestinal barrier in the ileum, cecum, vs. colon? Is there possible tissue injury associated with burst release at local sites?
5. Authors showed reduced systemic leakage of FITC-dextran after combined ButM treatment (Fig. 4b, d). While colitis and food allergy may share similar pathological mechanisms such as tight junction dysfunction, they are etiologically different. The use of DSS-induced model might not be suitable for food allergy, while the dysbiosis model is more relevant to food allergy. Mechanistically, they have not shown that ButM-mediated promotion of intestinal barrier function is crucial for protecting mice against peanut allergen challenge.
6. The authors showed that NtL-ButM upregulates gene expression signatures of AMPs associated with Paneth cells (Fig. 3a). But mechanistically, how does that affect protection against food allergen challenge? They should also measure AMPs in their peanut allergen challenge model.

7. The ButM modulates barrier functions but the authors used i.p. PN challenge (Fig5 c, d). What was the rationale for this instead of i.g. challenge? Can the authors show elevated levels of butyrate after ButM treatment in this model? Furthermore, there are peanut allergy models where antibiotics are not given. Does ButM protect mice against peanut allergy challenge even when mice are not pre-treated by antibiotics?
8. Can authors show the relative abundance of Clostridium XIVa before versus after treatment with PBS or ButM? Figure 6C only shows that after the treatment.
9. Mechanistically, how does butyrate delivery increase Clostridium XIVa? Also, they haven't shown whether the overall protection is mediated via increased Tregs, reduced mMCP-1, or reduced PN-IgE.
10. Authors should also dose animals with the equivalent dose of butyrate as in ButM (full dose) and directly compare their efficacy in the peanut allergy challenge model.
11. In Figure 5, can the authors also provide the PN-specific IgG1?
12. Can the authors show the efficacy of ButM in a food allergy model other than the peanut challenge model?

Thu 25 Aug 2022

Decision on Article nBME-22-0241A

Dear Dr Nagler,

Thank you for your revised manuscript, "Microbial metabolite butyrate-prodrug polymeric micelles demonstrate therapeutic efficacy in pre-clinical models of food allergy and colitis", which has been seen by the original reviewers. In their reports, which you will find at the end of this message, you will see that the reviewers acknowledge the improvements to the work and that Reviewer #3 raises a few additional technical criticisms that I am hoping you will be able to address.

As before, when you are ready to resubmit your manuscript, please upload the revised files, a point-by-point rebuttal to the comments from Reviewer #3, the reporting summary, and a cover letter that explains the main improvements included in the revision.

We look forward to receive a further revised version of the work. Please do not hesitate to contact me should you have any questions.

Best wishes,

Pep

Pep Pàmies

Chief Editor, Nature Biomedical Engineering

Reviewer #1 (Report for the authors (Required)):

Basing on my previous comments, the authors have examined the therapeutic efficacy of NtL-ButM and Neg-ButM individually in peanut allergic mice and also added a control of free butyrate. Furthermore, the release of butyrate from the micelles in the colon has been verified by newly added data. So, my major concerns have been addressed appropriately. It is recommended for publication now.

Jinyao Liu

Reviewer #2 (Report for the authors (Required)):

The authors developed two polymeric micelle systems that release butyrate from their polymeric core in the ileum or the cecum. They found that these butyrate-containing micelles could protect the intestinal barrier. There could also protect peanut-allergic mice from an anaphylactic reaction to peanut challenge. This butyrate-containing micelles can overcome the problem of butyrate acid application at present. The author's reply and modification make the result of this article more convincing. I have no other questions.

Reviewer #3 (Report for the authors (Required)):

The authors addressed the previous reviews and they also presented a new dataset showing the efficacy of ButM in a new colitis model. The new dataset raises some questions.

1. The authors stated that "Finally, we are not aware of another reliable peanut allergy model that does not use antibiotic treatment." However, there are many papers showing peanut allergy models that do not use antibiotic treatment. The authors used antibiotics to deplete butyrate-producing microbes so that they can test their butyrate-polymers. It remains to be seen whether this approach works in other peanut allergy

models without pretreatment of antibiotics.

2. Figure 4: there is a typo. I think they used 10 microgram of cholera toxin (not 10 mg).

3. Figure 5: It took me a while to understand their study design and dataset. Panels d-i) seem to show the dataset after vancomycin treatment. But they should show the dataset (panels d-i) before or after vancomycin treatment in separate experiments so that readers can understand the impact of ButM treatment (before vancomycin-mediated reversal). Also, does ButM work in this model in the absence of Treg transfer? In the peanut allergy model, they showed ButM doesn't affect Tregs. So in this colitis model, it is not clear whether ButM works by directly affecting Tregs or the epithelial barrier integrity.

4. Figure 5: panel c and d show "33%" and "22%" watermark in their graphs and it's not clear what they are referring to.

Fri 30 Sep 2022

Decision on Article nBME-22-0241A

Dear Dr Nagler,

Thank you for your revised manuscript, "Microbial metabolite butyrate-prodrug polymeric micelles demonstrate therapeutic efficacy in pre-clinical models of food allergy and colitis". Having consulted with Reviewer #3 (who has no further concerns), I am pleased to write that we shall be happy to publish the manuscript in *Nature Biomedical Engineering*.

We will be performing detailed checks on your manuscript, and in due course will send you a checklist detailing our editorial and formatting requirements. You will need to follow these instructions before you upload the final manuscript files.

Best wishes,

Pep

Pep Pàmies
Chief Editor, Nature Biomedical Engineering

Reviewer #3 (Report for the authors (Required)):

They have addressed my previous reviews.

Rebuttal 1

We thank Editors and Reviewers for their attention to our manuscript. Our response to their suggestions and comments have improved the manuscript and helped us to better convey some of our main findings and their significance. We have included a detailed response to the reviewers' questions and comments below. We have addressed all of the concerns raised by each of the reviewers with substantial revision of the manuscript text and the addition of ELEVEN NEW figures of data (Figures **S13, S17, S20, S21, S23, S24, S25, S26, S27, Fig. 4i-m** and the T cell transfer colitis model (**Figure 5**). **We have used red type font to highlight the new text and figures added to the manuscript.** We hope that the revised manuscript is now acceptable for publication in Nature Biomedical Engineering.

Editor:

In particular, as per the reviewer points, we would expect that a revised version of the manuscript provides extended characterization data, additional mechanistic insight, and a performance comparison with butyrylated starches.

There are pragmatic difficulties with comparing to butyrylated starch, in that this material is not commercially available. Moreover, the material is very heterogeneous and poorly characterized, meaning that if we were to try to synthesize it ourselves, we would certainly not obtain the same material as has been reported in the literature (with essentially no characterization by which to compare). Thus, as explained below, we have compared with free butyrate, a gold standard that has been explored clinically. Moreover, we now also compare the blended micelles (1:1, neutral and negative) to each micelle formulation as monotherapy, which also showed a benefit of our favored formulation. Thus, we now have both external and internal comparisons, all with well-characterized and controlled standards.

There are several studies using butyrylated starches, as a substitute (ranging from 5%-25%) for feeding animals¹⁻⁴. The butyrylated starches have shown ability to increase the butyrate concentration in the lower GI tract^{2,3}, however, none of them have shown therapeutic effects in the disease settings. There are also important limitations towards clinical translation. These butyrylated starches have limited capacity for loading butyrate, and thus require a high percentage in the regular diet and continuous feeding for weeks to achieve biological efficacy. As sodium butyrate is the current clinical standard and has been more widely used in pre-clinical models⁵⁻¹⁰, we use this to demonstrate improved efficacy of our micelle constructs in NEW Fig. 4i-m, Fig. S17, S24, S26.

Reviewer #1:

In this manuscript, the authors developed two polymeric micellar systems, which could release butyrate from their polymeric core in the ileum or the cecum, respectively. Oral delivery of butyrate to the ileum by these nanomicelles up-regulated genes expressing antimicrobial peptides in the ileal epithelium in germ-free mice. The use of the combination of these two types of nanomicelles restored a barrier-protective response in mice treated with either dextran sodium sulfate or antibiotics. Treatment with the micelles also protected peanut-allergic mice from an anaphylactic reaction to peanut challenge and rescued their dysbiosis by increasing the abundance of Clostridium Cluster XIVa. This work is interesting and offers insights for developing new therapies for food allergies.

Comments for improvement:

1. The advantages of these butyrate-loaded nanomicelles over other formulations such as commonly used butyrylated starches mentioned in the introduction should be demonstrated as this is critical for claiming the advance and significance of this work.

We thank the reviewer for bringing up this point. The butyrylated starch that is used mostly in preclinical animal models often requires continuous feeding for weeks as the only accessible food source for animals. This would be challenging for clinical translation to human use, especially given that a high dose of butyrate is needed to achieve biological efficacy. In contrast, our butyrate-loaded micelles provide a controllable and supplemental approach to deliver butyrate to the distal gut. Our

micelles contain a high content of butyrate that would allow feasible dosing regimens of butyrate as therapeutics. We have included new data using a sodium butyrate control group in Fig. 4 and Fig. S17, S24, and S26. These data demonstrated that at the same butyrate dose, the ButM micelles protect from the allergic response and release content in the lower GI tract, but sodium butyrate is predominantly absorbed in the stomach and has no therapeutic effect. As sodium butyrate is the current clinical standard and has been more widely used in pre-clinical models⁵⁻¹⁰, we can use this to demonstrate improved efficacy of our micelle constructs. Moreover, as explained above to the Editor, the butyrylated starch is not available to us, and its characterization has not been published sufficiently well as to enable us to synthesize a comparable material ourselves. Thus, our reliance on well-characterized benchmark compounds.

2. As introduced, butyrate is the preferred energy substrate for colonic epithelial cells and the oral delivery of butyrate to the colon has been a challenge, while as the authors evidenced the current nanomicelles did not increase the level of butyrate in the colon following intragastric administration. Are these micelles still lack of capability to tackle this difficulty?

The previous experiments we did in SPF mice showed no increased butyrate level in the colon. We reasoned that could be explained by the existing high level of butyrate produced by the healthy microbiome. To address this comment, we recently performed another biodistribution study on vancomycin-treated mice. Vancomycin largely depletes Gram-positive bacteria, including most of the butyrate-producing bacteria. Our new data demonstrates that both NtL-ButM and Neg-ButM deliver and release butyrate in the colon (NEW Fig. S17). In this experiment, we have also included sodium butyrate as a control group and observed that sodium butyrate was mainly detected in the stomach after oral gavage. Sodium butyrate could not reach the lower GI tract including the cecum and colon in vancomycin-treated mice, demonstrating increased potential for efficacy of NtL-ButM and Neg-ButM. This new result has been added to the Result section line 208-216.

3. It was described that the core structure made of pBMA was more condensed with higher contrast, while this could not be observed from cryo-TEM images. Furthermore, the sizes of these two types of micelles calculated from cryo-TEM images are quite different. However, as stated, the DLS results showed similar sizes of 45 and 40 nm respectively, which was confusing. Please clarify.

We appreciate the reviewer's comments on the difference of sizes measured by DLS and the CryoEM. The CryoEM only showed pBMA core, and it is likely Neg-ButM has a smaller BMA block. The DLS measures the size of whole structure, including the hydrophilic corona, and indicates a hydrodynamic diameter. Thus, it is expected the DLS gave a larger value of the size compared to cryo-EM, especially for the charged nanoparticles such as Neg-ButM.

4. The detailed formulation of simulated gastric fluid and simulated intestinal fluid should be added as this is key for evaluating the stability of the micelles as well as their release behaviors. For example, whether these fluids contained pepsin or trypsin.

We thank the reviewer for this comment and added the information on simulated gastric fluid and simulated intestinal fluid to the Methods. The simulated gastric fluid was purchased from Ricca Chemical Company, which contains 0.2% (w/v) sodium chloride in 0.7% (v/v) hydrochloric acid and was added with 3.2 mg/mL pepsin from porcine gastric mucosa (Sigma). The simulated intestinal fluid was purchased from Ricca Chemical Company, which contains 0.68% (w/w) potassium dihydrogen phosphate, 0.06% (w/w) sodium hydroxide, and pancreatin at 1% (w/w). This has been added to the Method section line 621-625.

5. The release studies showed that most of their butyrate was released within minutes in simulated intestinal fluid. It would be expected that the release may even faster after gastric empty in vivo. The burst release would inevitably promote absorption and metabolism in the small intestine. This may be the main reason why increased level of butyrate in the colon was not achieved.

We appreciate this insightful comment. We have now measured the butyrate concentration in the stomach after oral gavage in vancomycin-treated mice. Compared to sodium butyrate, both NtL-ButM and Neg-ButM exhibit a very low level of butyrate release in the stomach (NEW Fig. S17c). Both micelles transited from the stomach to the lower GI tract quickly, within an hour, according to a new biodistribution study using IVIS (NEW Fig. S17b). With a limited amount of fluid in the GI tract, it is possible that the micelles release less butyrate in vivo as they transit through the GI tract, compared to what we observed in the simulated intestinal fluid.

6. IVIS imaging indicated that accumulation in the stomach was observed for Neg-ButM, which has a relative high CMC value. It was speculated that the micelles would suffer instability as the presence of continuous flush by gastric fluid. Is there any data showing the in vivo butyrate release in stomach and the percent of butyrate released there?

As discussed in the comment above, we measured the butyrate concentration in the stomach (NEW Fig. S17). Comparing the area under the curve from sodium butyrate, NtL-ButM and Neg-ButM treatment, the release of butyrate from NtL-ButM and Neg-ButM were 2.3%, or 8.8% as compared to amount of free sodium butyrate delivered in the stomach. The Neg-ButM did show a relatively higher release of butyrate compared to NtL-ButM, which is possibly due to its instability in the acidic condition. We thank the reviewer for bringing up this question. This new result has been added to the Result section line 208-216.

7. The reason for the selection of a 1:1 combination of NtL-ButM and Neg-ButM for efficacy study should be explained. Moreover, to clarify the individual effects of each type of micelle, both NtL-ButM and Neg-ButM should be used as controls in the assessment study in peanut allergic mice.

We are grateful that the reviewer has brought up this important point about the efficacy from combination or single butyrate micelles. Due to the different biodistribution and butyrate release behaviors in vivo from the two butyrate micelles, we reasoned that the combined dosing of NtL-ButM and Neg-ButM would cover the longest section of the lower GI tract for a longer period of time. Thus, we chose 1:1 molar combination of NtL-ButM and Neg-ButM to maximize the potential therapeutic effects. In new data provided with this revised manuscript, we have also examined the therapeutic efficacy of NtL-ButM and Neg-ButM individually in peanut allergic mice (NEW Fig. S21). We tested But M (mix of NtL-ButM and Neg-ButM), twice daily as previously. We also added the controls the reviewer requested by examining treatment with each of the polymers individually (NtL-ButM or Neg-ButM, twice daily). We included an additional group in which we tested a half dose of ButM, twice daily. We found that butyrate micelle treatment reduced the anaphylactic response in a dose dependent manner. Both NtL-ButM and Neg-ButM administered individually significantly reduced the anaphylactic response to peanut challenge, although not as effectively as the polymer combination. This new result has been added to the Result section line 293-296.

8. As the authors observed that reducing the dose of ButM by half was not as effective as the full dose in protecting mice from an anaphylactic response. Whether increasing the dose could further improve the treatment efficacy. This is important to disclose the potency of these nanomicelles.

We thank the reviewer for this comment. The current dose of 1:1 combination of NtL-ButM and Neg-ButM has achieved remarkable efficacy in protecting allergic mice from anaphylactic reactions. The room for further improvement is very limited in this model. In addition, we wanted to choose the lowest dose that can maintain effectiveness to facilitate clinical translation. In new data provided in the revised manuscript we have shown that the half dose is not as effective as the full dose (NEW Fig. S21). This suggests that we should consider the full dose to be the minimal effective dose in future studies.

9. At least, a control group of free butyrate in the same dose should be set for all in vivo experiments, to compare the treatment efficacy.

We are grateful that the reviewer has brought up this important point. We have conducted another experiment with the peanut allergy model testing free sodium butyrate in a head-to-head comparison with PBS and ButM treatment. The data clearly shows that free sodium butyrate, at the same butyrate dose as ButM, does not protect peanut allergic mice from an anaphylactic response (new Fig. 4 i-m, line 287-291). The mice treated with sodium butyrate experienced a similar drop in core body temperature compared to the PBS-treated group after peanut challenge. We predict that this inability of sodium butyrate to have a therapeutic effect in the peanut allergy model may be because sodium butyrate does not transit to the small intestine or colon (NEW Fig. S17).

10. It was explained that butyrate sensing by peroxisome proliferator-activated receptor shunts colonocyte metabolism toward β -oxidation, creating a local hypoxic niche for these oxygen sensitive anaerobes. While, most of the gut microbiota including both beneficial and pathogenic bacteria are anaerobic. The mechanism of ButM-mediated increased abundance of Clostridium Cluster XIVA should be further clarified.

We thank the reviewer for this thoughtful comment. This work cited is from Baumler and colleagues (ref#49). While most of the gut microbiota is anaerobic, some are facultative anaerobes and can also grow in the presence of oxygen. Numerically, Clostridia are the dominant butyrate producers in the gut, and are highly oxygen sensitive obligate anaerobes. Because they are so abundant, and under homeostatic conditions, they produce the butyrate that creates a hypoxic niche when colonocyte metabolism is shunted to β -oxidation, their expansion is favored by enhanced concentrations of butyrate. In our models we create dysbiosis by depleting butyrate producing Clostridia with antibiotics. Treatment with ButM restores homeostatic butyrate concentrations (symbiosis). Other taxa are also expanded in the presence of hypoxia as our data also shows. The pathway characterized by Baumler's study is depicted in the cartoon included below.

Reviewer Figure 1.

11. Assessment of ameliorated anaphylaxis was incomprehensive. The authors detected the levels of mouse mast cell protease-1 (mMCPT-1) and specific IgE in anaphylactic responses. However, the pathogenesis of the allergic response is that binding of allergens to specific IgE trigger cross-linking of Fc ϵ R1 on mast cells, which induce cellular degranulation and release of histamine. Crucial factors including degree of cellular degranulation and level of histamine should be detected. Moreover, the authors should show the gating strategy to obtain the final fraction of mast cells.

In our hands, mouse mucosal mast cell protease-1 is an excellent marker of mast cell degranulation. According to the manufacturer mMCPT-1 is the “only chymase expressed by intestinal mucosal mast cells, which are found in the intestinal epithelium. Elevated MCPT-1 levels are also observed during intestinal allergic hypersensitivity reactions.” We have also measured plasma histamine levels in prior studies (Bashir et al J. Immunol. 2004) and, as the reviewer has requested, we

have now added plasma histamine levels to this report (Fig. 4f and k). Upon allergen challenge ButM treatment significantly ameliorated the anaphylactic response by reducing the drop in core body temperature, serum levels of mMCP-1 and histamine, as well as PN-specific IgE and IgG1 (Fig. 4d-h).

12. Measurement of the intestinal permeability with lactulose/mannitol test is recommended as the uncontrolled flux of food antigen across the small intestine is associated with the development of food allergy. However, FITC-dextran in 4 kDa mainly refers to the permeability of entire gut, particularly the large intestine.

Reviewer Figure 2. Visual inspection of the extent of FITC-dextran motility 1.5 hours after i.g. gavage. The black line denotes the most distal location of FITC dextran (light yellow coloration).

Measurement of lactulose/mannitol in 24 hr urine samples has been standardized as clinical assay for epithelial barrier permeability in humans but is difficult to perform and standardize and is only available at one site (Mayo Clinic). This test has not been developed for use in mice. Cochran, et al have recently validated the utility of DSS to induce a clinically relevant loss of intestinal barrier function (ref#43). For the DSS-induced colitis model, we have collected blood 4 hr after FITC-dextran gavage, which measures permeability of the entire gut (because the green fluorescence of FITC is visible we can confirm that the FITC has traversed the entire GI tract by visual inspection when the mice are euthanized). For the Abx-treated model, we bleed at an earlier time point (1.5 hr) before the FITC-dextran has transited to the colon (again, we can monitor this visually – see Reviewer Fig. 2). We have thoroughly developed this model in our laboratory to validate that FITC-dextran consistently transits through the small intestine but does not reach the colon within 1.5 hr after gavage.

Reviewer #2:

The authors developed two polymeric micelle systems that release butyrate from their polymeric core in the ileum or the cecum. They found that these butyrate-containing micelles could protect the intestinal barrier. There could also protect peanut-allergic mice from an anaphylactic reaction to peanut challenge. This butyrate-containing micelles can overcome the problem of butyrate acid application at present. However, there are some questions :

1.The FITC-dextran experiment could only evaluate the overall characteristics of the intestinal barrier. As the butyrate micelles could mainly release butyrate in the ileum and cecum, it is suggested that the key mucosal barrier proteins and mucus layer should be detected.

Please see response to Reviewer #1. If we shorten the timeframe post-gavage we can euthanize mice before the FITC-dextran has transited to the colon and use it as a measure of barrier permeability in the small intestine.

We also looked at changes in relative expression of several mucosal barrier proteins by RT-qPCR. Although some significant differences were detected, this analysis needs to be developed further to examine the influence of ButM on tight junction protein complexes in future studies.

Reviewer Figure 3. ButM treatment increases the relative expression of some genes involved in epithelial barrier function. SPF C57BL/6 mice were gavaged daily with an antibiotic cocktail for 7 days until weaning. Once weaned, the mice were gavaged twice daily with PBS ($n = 5$) or ButM ($n = 6$) for 7 days. Relative expression of tight junction protein 1 (*TJP1*), claudlin-1 (*Cldn1*), claudlin-2 (*Cldn2*), claudlin-4 (*Cldn4*), *Reg3β*, and *Reg3γ* in ileum intestinal epithelial cells. Data were analyzed by Student's t test.

2.NtL-ButM dramatically increased the butyrate concentration in the ileum for up to 2 hr after gavage (Fig. 2c). Partial of Neg-ButM could traveled to the ileum 1 h after gavage (Fig. S15a). It could not enhance the butyrate in the ileum. Why?

We thank the reviewer for this question. In a new biodistribution study we conducted on vancomycin-treated mice, both NtL-ButM and Neg-ButM transit to the lower GI tract within an hour. We observed moderate butyrate release from both NtL-ButM and Neg-ButM in the ileum only in the first 2 hours, then more release in the cecum and colon between 1-8 hours (NEW Fig. S17c). As to the Neg-ButM's low release in the ileum, we observed that the Neg-ButM was not stable in acidic conditions, e.g., in the simulated gastric fluid in vitro, and can aggregate into larger polymer particles (NEW Fig. S13b). This might partially explain the reason why, in the non-antibiotic treated SPF mice, the Neg-ButM stayed longer in the stomach, and did not release butyrate in the ileum. Those aggregates might move slowly through the upper GI tract, and after they reached the cecum, release more butyrate in the presence of more degradative enzymes. This new result has been added to the Result section line 171-172, and 208-216.

3.There is no control group in Fig.2c, 2d and 2e. It is difficult to evaluate the effect of NtL-ButM or Neg-ButM on the butyrate concentration in intestine.

In Fig. 2c, 2d and 2e, the red dotted line represented the endogenous butyrate levels in the untreated SPF mice. This has been clarified in the figure legend. In a new experiment using vancomycin-treated mice, we added sodium butyrate as another control group, and compared the butyrate released from ButM with the free sodium butyrate. This new data is included in the revision as NEW Fig. S17c, and line 208-216.

4. In Fig. 5, the number of mice in PBS group and ButM group were 31 and 42, respectively. Will the difference in the number of mice between the two groups affect the results?

With a NEW experiment added, we now have 40 mice for each PBS group and ButM group (NEW Fig. 4d-h). The difference in numbers of mice between the two groups did not affect the results. We also included an additional experiment comparing PBS, ButM and sodium butyrate with 12 mice in each treatment group (Fig. 4i-m). The mice in 4d-h and 4i-m are independent experiments.

5. In Fig. 6, Lactobacillus genus dominated the microbiota of mice in both group before treatment, while Bacteroides increased significantly after treatment. Besides, In Fig. S19, Clostridium Cluster XIVa increased in both group after treatment by PBS or ButM. Is it possible that the difference caused by age? And there are other genus significantly, whether micelles protects peanut-allergic mice by increasing the abundance of Clostridium Cluster XIVa?

While it has been shown that the microbiota shifts with age, the specific changes observed in this experiment are more likely to result from the cessation of antibiotic treatment. Both Bacteroidetes and Lachnospiraceae (including Clostridium Cluster XIVa) are susceptible to vancomycin, and when the vancomycin is halted, they are able to regrow in both treatment groups. This is why the differentially abundant taxa appear similar in both Fig. S22 a, b. However, the LEfSe analysis in Fig. 4o demonstrates that while many taxa increase in response to cessation of antibiotics, Clostridium Cluster XIVa increases to a higher abundance in the ButM treated mice than the increase observed in PBS treated mice. We predict that ButM may increase the rate or total amount at which Lachnospiraceae rebound after antibiotics by creating a hypoxic niche as described above in the response to reviewer #1. This increased rebound of Clostridium clusters has also been validated in unsensitized mice (NEW Fig. S23). In this experiment, after cessation of vancomycin and treatment with ButM or PBS, ButM-treated mice had increased relative abundance of Clostridium Cluster IV in the feces and increased abundance of Clostridium Cluster XVIII in the feces and ileal contents.

6. The level of phylum, family and genus were shown in Fig. 6A at the same time. It is difficult to understand.

This method of representing bacterial taxa at different levels (phylum, class, etc.) has been developed to show the reader the highest resolution of information possible without overcrowding the figure. These levels are microbiologically relevant to the gut microbiome. For example, the genus Lactobacillus is highly abundant, so we can classify this group of bacteria down to a very specific level. In contrast, other taxa such as Bacteroidetes and Actinobacteria have very low abundance of bacteria that often cannot be classified to the genus level. For this reason, it makes sense to represent the group at large. This representation has been more fully described in the text to help the reader's understanding and we have used a similar color scheme previously¹¹⁻¹³.

7. In Fig. S18d, after low dose PN treatment, the change of body temperature in the PN group (blue square) was greater than that in the PBS group for a long time (Fig. 5d). Can you explain?

The low dose PN treatment in original Fig. S18 was not optimized and failed to induce any desensitization to peanut. Because ButM is highly effective as a monotherapy (without antigen specific desensitization) we discontinued these studies. We have removed this suboptimal low dose PN treatment from the revised manuscript.

8. The butyrate-containing micelles could up-regulated genes expressing antimicrobial peptides in the ileal epithelium in germ-free mice. Whether it could affect the expression of genes encoding antimicrobial peptides in SPF mice?

In collaboration with Charles Bevins, an expert on AMPs at UC Irvine, we have performed RT-qPCR on ileal epithelial cells from SPF mice treated with PBS or NtL-ButM (NEW Fig. S20). In contrast to what we observed in the GF mice, there were no significant differences in the expression of AMPs in SPF mice. We have moved all the data on AMPs to the supplement and de-emphasized the role of AMPs in our mechanism of action since we only saw changes in gene expression in germ free mice. This new result has been added to the Result section line 238-245.

Overall, though the butyrate-containing micelles could release the butyrate in small intestine and protect peanut-allergic mice from an anaphylactic reaction to peanut challenge, its role in restoring microbial and mucosal homeostasis still needs to be shown.

We appreciate the reviewer's constructive comments. We conducted an experiment to further analyze the effects of ButM on immune cells in the peanut allergic mice. ButM did not alter regulatory T cell populations in the mesenteric lymph nodes (LNs) or spleen (NEW Fig. S24). However, it significantly down-regulated the expression of MHC Class II and the co-stimulatory marker CD86 on the dendritic cells and macrophages in the colon-draining and ileal-draining LNs (NEW Fig. S26), suggesting downmodulation of antigen presentation. This has been added to the Result section line 328-335. It has also been previously shown that butyrate inhibits mast cell activation through FcεRI-mediated signaling (new reference#54). This is also consistent with absence of hypothermia and elevated serum mMCPT-1 and histamine (all indicators of anaphylaxis) in ButM treated mice. This has been added to the Discussion section line 398-406 as well.

Reviewer #3:

Here, the authors developed two Butyrate-releasing polymeric micelles and examined their effects on intestinal barrier functions and allergic responses. The authors showed that NtL-ButM increased AMPs and ButM modulated gut microbiome with increased Clostridium Cluster XIVa in a dysbiosis model. While this presents an interesting concept, there are many mechanistic aspects that need to be addressed:

1. Neg-ButM was prepared via base titration. The stability of the anionic micelles in low pH buffer can be shown (Fig. 1e). Do the anionic micelles resist the low pH in the stomach without further segregation? Does it affect the enzyme-mediated butyrate release?

We appreciate this reviewer's question. We conducted a stability test in the simulated gastric fluid (SGF) which contains 0.2% (w/v) sodium chloride in 0.7% (v/v) hydrochloric acid (pH=1.3) and was added to 3.2 mg/mL pepsin from porcine gastric mucosa. The NtL-ButM remained intact and the size remained the same. However, we did observe that when we added Neg-ButM to the SGF, the micelles aggregated, and the size increased to ~2 μ m due to the aggregation (NEW Fig. S13). This might partially explain why, in the SPF mice, the Neg-ButM stayed longer in the stomach and did not release butyrate in the ileum. Those aggregates might move slowly through the upper GI tract, and after they reach the cecum, release more butyrate in the presence of more bacteria and degradative enzymes. This new result has been added to the Result section line 171-172.

2. The release of butyrate was proposed to be dependent on the esterase in the GI tract (L117-118). Does the amide group present in the pBMA chain resist enzymatic and microbial degradation and consequently release of 2-hydroxypropylamine?

We have shown that the polymer only lost the mass of butyrate after excretion from mice (Fig. S15). The amide group required much harsher conditions to hydrolyze than its ester homologue. It is possible that the hydrolysis of amide takes longer time than the micelles transit through the GI tract, thus, only butyrate from the ester hydrolysis was released into the GI tract.

3. Butyrate was released within minutes upon incubation with SIF (Fig. 2b) but NtL-ButM did not reach cecum and colon in vivo (Fig. 2c,e). What are the proposed mechanisms of Neg-ButM specifically targeting cecum and NtL-ButM targeting ileum (Fig. 2c,d)? The IVIS data in Figure S15 showed that the fluorescence intensity of Neg-ButM is already very low even at 6 h post oral administration. However, the butyrate release in cecum would last up to 12 h (Figure 2D), please explain this.

We thank the reviewer for these questions. We observed that the Neg-ButM can aggregate into larger polymer particles in acidic conditions, e.g., in the simulated gastric fluid. (NEW Fig. S13b). This might partially explain why, in the non-antibiotic treated SPF mice, the Neg-ButM did not release butyrate in the ileum. Those aggregates might move slowly through the upper GI tract, and after they reached the cecum, release more butyrate in the presence of more degradative enzymes and bacteria. Additionally, higher pH in the distal gut may lead to more binding of the negatively charged micelles to the gut mucosa.

We anticipate that the butyrate released from Neg-ButM and detected in the cecum after 6 hr could be due to an accumulation at this site. In the new IVIS study in the vancomycin-treated mice, we added a 12 hr time point to match the biodistribution study (NEW Fig. S17b). We observed that both NtL-ButM and Neg-ButM transited faster in the vancomycin-treated mice compared to SPF mice, resulting in butyrate release in both cecum and colon from both micelles. The peak concentrations were observed between 2-4 hours after oral gavage (Fig. S17c), which is similar to what we observed from fluorescent signals from the IVIS data. This new result has been added to the Result section line 208-216.

4. Does the micelle disproportionately promote barrier integrity along the long intestinal tract? Can the authors examine integrity of the intestinal barrier in the ileum, cecum, vs. colon? Is there possible tissue injury associated with burst release at local sites?

As mentioned in the response to reviewer #1, for the DSS-induced colitis model, we have collected blood 4 hr after FITC-dextran gavage, which measures permeability of the entire gut (because the green fluorescence of FITC is visible we can confirm that the FITC has transversed the entire GI tract by visual inspection when the mice are euthanized). For the Abx-treated model, we bleed at an earlier time point (1.5 hr) before the FITC-dextran has transited to the colon (again, we can monitor this visually – see Reviewer Fig. 1). RT-qPCR of ileal epithelial cells suggests that ButM may influence expression of barrier-regulating genes in ileal epithelial cells (Reviewer Fig. 3, see response to Reviewer #1). However, we need to continue to develop methods for analyzing barrier integrity in individual sites.

The reviewer raises the possibility of burst release as the micelles transit the gut. Due to the distributed nature of the dosage, spread between vast numbers of micelles which are then spread over and transiting through the gut, such burst release is highly unlikely. The ester links between the butyrate payload and the polymer backbone are cleaved individually, i.e., where one cleavage leads to release of one molecule of butyrate, also making burst release unlikely.

5. Authors showed reduced systemic leakage of FITC-dextran after combined ButM treatment (Fig. 4b, d). While colitis and food allergy may share similar pathological mechanisms such as tight junction dysfunction, they are etiologically different. The use of DSS-induced model might not be suitable for food allergy, while the dysbiosis model is more relevant to food allergy. Mechanistically, they have not shown that ButM-mediated promotion of intestinal barrier function is crucial for protecting mice against peanut allergen challenge.

We have previously shown that neonatal administration of antibiotics reduces intestinal microbial diversity and impairs epithelial barrier function, resulting in increased access of food allergens to the systemic circulation (reference #24). In this study, we examined whether ButM can restore epithelial barrier function in a variety of conditions, including the dysbiosis model that is more relevant to food allergy. In addition, we have now included new data in the CD45RB^{hi} transfer model of colitis (NEW Fig. 5). Taken together these data demonstrate that ButM can restore barrier function and protect against both food allergy and colitis in pre-clinical murine models. In addition, we have provided new data that one mechanism by which ButM may protect against allergen challenge is by reducing antigen presentation and activation of myeloid cell subsets (NEW Fig. S26). These new results have been added to the Result section line 328-366.

6. The authors showed that NtL-ButM upregulates gene expression signatures of AMPs associated with Paneth cells (Fig. 3a). But mechanistically, how does that affect protection against food allergen challenge? They should also measure AMPs in their peanut allergen challenge model.

In collaboration with Charles Bevins, an expert on AMPs at UC Irvine we have performed RT-qPCR on ileal epithelial cells from SPF mice treated with PBS or NtL-ButM (NEW Fig. S20). In contrast to what we observed in the GF mice, there was no significant difference in gene expression for AMP production in the SPF mice. We have revised the manuscript text to downplay a mechanistic role for the induction of AMPs by ButM.

7. The ButM modulates barrier functions, but the authors used i.p. PN challenge (Fig5 c, d). What was the rationale for this instead of i.g. challenge? Can the authors show elevated levels of butyrate after ButM treatment in this model? Furthermore, there are peanut allergy models where antibiotics are not given. Does ButM protect mice against peanut allergy challenge even when mice are not pre-treated by antibiotics?

We thank the reviewers for these thoughtful comments. In all of our previously published work in preclinical food allergy models (Bashir et al, J Immunol., 2004; Stefka et al PNAS, 2014, Feehley et al Nat. Med., 2019) the mice were sensitized intragastrically and also challenged intragastrically. The response to intragastric challenge exhibited some variability, including a few non-responders. The pre-clinical model utilized in this study is different from our earlier work in several ways. First, this is a therapeutic model – no treatment was initiated until the mice had a documented allergic response. Because this therapeutic model is evaluating the efficacy of a drug treatment, we reasoned that all of

the mice must exhibit a uniform allergic response so that we would be able to distinguish a mouse that failed to respond to sensitization from a mouse whose allergic response was inhibited by treatment with our drugs. This was the rationale for changing to an intraperitoneal challenge in this model. The data in Fig. 4c clearly shows the uniformity of the response to i.p. challenge. The use of an i.p. challenge also provides an exciting new mechanistic insight. The best explanation for the effect of butyrate treatment (in the absence of allergen) on the response to allergen challenge is a *direct* effect of butyrate on mast cell degranulation, as has been reported by others (new reference #54).

In new Fig. S17 we have examined butyrate release after one dose of NtL-ButM and NegButM in SPF mice that were treated for three weeks with vancomycin treatment in the drinking water. We were able to detect butyrate release after just one treatment with ButM. We did not also examine butyrate concentrations at the time of euthanasia in the full allergy model.

Finally, we are not aware of another reliable peanut allergy model that does not use antibiotic treatment. Instead, we tested the efficacy of our drugs in a second model of colitis (the CD45RB^{hi} transfer model) (NEW Fig. 5, line 337-366).

8. Can authors show the relative abundance of Clostridium XIVa before versus after treatment with PBS or ButM? Figure 6C only shows that after the treatment.

In both treatment groups, the abundance of Clostridium XIVa was zero before treatment from 16S sequencing data and near zero (or limit of detection) from qPCR. This can be visualized in Fig. 4n, showing no Lachnospiraceae in either group prior to treatment. This is due to the vancomycin pre-treatment, which depleted most of the Gram-positive bacteria, including the Clostridium Cluster XIVa. We compared differentially abundant taxa within each treatment group before and after treatment (Fig. S22). However, the comparison between PBS and ButM treatment post-treatment is more important here to demonstrate the effect of butyrate treatment on the abundance of Clostridium clusters on these vancomycin pre-treated mice (Fig. 4n-q, and NEW Fig. S23).

9. Mechanistically, how does butyrate delivery increase Clostridium XIVa? Also, they haven't shown whether the overall protection is mediated via increased Tregs, reduced mMCP-1, or reduced PN-IgE.

Please see response to Reviewer #2 for the proposed mechanism by which butyrate induced hypoxia expands Clostridia Cluster XIVa (Reviewer Fig. 1).

We have measured decrease in core body temperature and elevated levels of mMCP-1, histamine and PN-specific IgE and IgG1 as indicators of anaphylaxis and found all to be reduced in ButM treated mice, compared to PBS or sodium butyrate treated mice (Fig. 4). We have evaluated the regulatory T cell populations in the mesenteric LNs and spleen and did not observe any differences among treatment groups, suggesting that ButM did not affect these Treg populations in this model (NEW Fig. S24). We further evaluated if ButM had any effect on the myeloid cells in peanut allergic mice. ButM treatment significantly down-regulated the expression of MHC Class II and the co-stimulatory marker CD86 on the dendritic cells and macrophages in the colon-draining and ileal-draining LNs (NEW Fig. S26) suggesting downmodulation of antigen presentation. This new result has been added to the Result section line 328-335, and Discussion section line 398-406.

10. Authors should also dose animals with the equivalent dose of butyrate as in ButM (full dose) and directly compare their efficacy in the peanut allergy challenge model.

As mentioned in the response to Reviewer #1 we have conducted another experiment with the peanut allergy model testing free sodium butyrate in a head-to-head comparison with PBS and ButM treatment. The data clearly shows that free sodium butyrate, at the same butyrate dose as ButM, does not protect peanut allergic mice from an anaphylactic response (NEW Fig. 4i-m). The mice treated with sodium butyrate experienced a similar drop in core body temperature compared to the PBS-treated

group after peanut challenge. This new result has also been discussed in the Result section line 287-291.

11. In Figure 5, can the authors also provide the PN-specific IgG1?

We have added the PN-specific IgG1 data (Fig. 4h,m and Fig. S21h)

12. Can the authors show the efficacy of ButM in a food allergy model other than the peanut challenge model?

We are not aware of another reliable model of peanut allergy. We have instead included data from a second colitis model, the CD45RB^{hi}T cell transfer model of colitis (NEW Fig. 5, line 337-366).

Reference

1. Furusawa, Y., *et al.* Commensal microbe-derived butyrate induces the differentiation of colonic regulatory T cells. *Nature* **504**, 446 (2013).
2. Annon, G., Illman, R.J. & Topping, D.L. Acetylated, Propionylated or Butyrylated Starches Raise Large Bowel Short-Chain Fatty Acids Preferentially When Fed to Rats. *The Journal of Nutrition* **133**, 3523-3528 (2003).
3. Bajka, B.H., *et al.* Butyrylated starch increases large bowel butyrate levels and lowers colonic smooth muscle contractility in rats. *Nutrition Research* **30**, 427-434 (2010).
4. Nielsen, T.S., *et al.* High-Amylose Maize, Potato, and Butyrylated Starch Modulate Large Intestinal Fermentation, Microbial Composition, and Oncogenic miRNA Expression in Rats Fed A High-Protein Meat Diet. in *International journal of molecular sciences*, Vol. 20 E2137 (2019).
5. Tan, J., *et al.* Dietary Fiber and Bacterial SCFA Enhance Oral Tolerance and Protect against Food Allergy through Diverse Cellular Pathways. *Cell Reports* **15**, 2809-2824 (2016).
6. Sun, M., *et al.* Microbiota-derived short-chain fatty acids promote Th1 cell IL-10 production to maintain intestinal homeostasis. *Nature Communications* **9**, 3555 (2018).
7. Smith, P.M., *et al.* The microbial metabolites, short-chain fatty acids, regulate colonic Treg cell homeostasis. *Science* **341**, 569-573 (2013).
8. Arpaia, N., *et al.* Metabolites produced by commensal bacteria promote peripheral regulatory T-cell generation. *Nature* **504**, 451-455 (2013).
9. Cait, A., *et al.* Microbiome-driven allergic lung inflammation is ameliorated by short-chain fatty acids. *Mucosal Immunology* **11**, 785-795 (2018).
10. Chen, G., *et al.* Sodium Butyrate Inhibits Inflammation and Maintains Epithelium Barrier Integrity in a TNBS-induced Inflammatory Bowel Disease Mice Model. *EBioMedicine* **30**, 317-325 (2018).
11. Sorbara, M.T., *et al.* Inhibiting antibiotic-resistant Enterobacteriaceae by microbiota-mediated intracellular acidification. *Journal of Experimental Medicine* **216**, 84-98 (2018).
12. Keith, J.W., *et al.* Impact of Antibiotic-Resistant Bacteria on Immune Activation and *Clostridioides difficile* Infection in the Mouse Intestine. *Infect Immun* **88**(2020).
13. Sorbara, M.T., *et al.* Functional and Genomic Variation between Human-Derived Isolates of Lachnospiraceae Reveals Inter- and Intra-Species Diversity. *Cell Host Microbe* **28**, 134-146.e134 (2020).

Rebuttal 2

Point by Point Reply

Reviewer #3 (Report for the authors (Required)):

The authors addressed the previous reviews, and they also presented a new dataset showing the efficacy of ButM in a new colitis model. The new dataset raises some questions.

1. The authors stated that “Finally, we are not aware of another reliable peanut allergy model that does not use antibiotic treatment.” However, there are many papers showing peanut allergy models that do not use antibiotic treatment. The authors used antibiotics to deplete butyrate-producing microbes so that they can test their butyrate-polymers. It remains to be seen whether this approach works in other peanut allergy models without pretreatment of antibiotics.

We should have answered this question more clearly. There are indeed other peanut allergy models that do not use antibiotics. Most of these are models of allergic diarrhea in which the mice are sensitized and then challenged with antigen (usually OVA) plus alum. As we stated in our cover letter to the editor, dysbiosis is central to the proposed mechanism of action of our drug. We therefore did not test our drug in models of food allergy that do not exhibit dysbiosis, since this is not the condition for which it was designed. Based on our communication with the editor, he seems to find this acceptable.

2. Figure 4: there is a typo. I think they used 10 microgram of cholera toxin (not 10 mg).

The reviewer is quite correct. Thank you for catching this error.

3. Figure 5: It took me a while to understand their study design and dataset. Panels d-i) seem to show the dataset after vancomycin treatment. But they should show the dataset (panels d-i) before or after vancomycin treatment in separate experiments so that readers can understand the impact of ButM treatment (before vancomycin-mediated reversal). Also, does ButM work in this model in the absence of Treg transfer? In the peanut allergy model, they showed ButM doesn't affect Tregs. So in this colitis model, it is not clear whether ButM works by directly affecting Tregs or the epithelial barrier integrity.

After considering the reviewer's comments we agree that the presentation of this data was confusing. We have therefore included a new schema for the experimental design (Fig. 5a) and a new Table of Treatment Groups (Fig. 5b). We also reorganized some of the data display and clarified the text. Our proposed mechanism of action does not involve induction of Tregs, so we did not include additional arms of this study without Treg transfer. Our objective in including this model in the revision was to show generality in a second model of gut dysfunction, rather than delve into the mechanistic details of the model, which indeed may be model specific. Based on our communication with the editor, he seems to find this acceptable.

4. Figure 5: panel c and d show “33%” and “22%” watermark in their graphs and it's not clear what they are referring to.

We have removed these watermarks – thank you for pointing this out.